

# Bridging classical data assimilation and optimal transport

Marc Bocquet[1], Pierre J. Vanderbecken[1], Alban Farchi[1], Joffrey Dumont Le Brazidec[1], and
Yelva Roustan[1]

[1]CEREA, École des Ponts and EDF R&D, Île-de-France, France

**Correspondence:** Marc Bocquet (marc.bocquet@enpc.fr)

**Abstract.** Because optimal transport acts as displacement interpolation in physical space rather than as interpolation in value space, it can potentially avoid double penalty errors. As such it provides a very attractive metric for non-negative physical fields comparison — the Wasserstein distance — which could further be used in data assimilation for the geosciences. The algorithmic and numerical implementations of such distance are however not straightforward. Moreover, its theoretical formulation within

typical data assimilation problems face conceptual challenges, resulting in scarce contributions on the topic in the literature.

We formulate the problem in a way that offers a unified view on both classical data assimilation and optimal transport. The resulting *OTDA* framework accounts for both the classical source of prior errors, background and observation, together with a Wasserstein barycentre in between states that stand for these background and observation. We show that the hybrid OTDA analysis can be decomposed as a simpler OTDA problem involving a single Wasserstein distance, followed by a Wasserstein

barycentre problem which ignores the prior errors and can be seen as a *McCann interpolant*. We also propose a less enlightening but straightforward solution to the full OTDA problem, which includes the derivation of its analysis error covariance matrix. Thanks to these theoretical developments, we are able to extend the classical 3D-Var/BLUE paradigm at the core of most classical data assimilation schemes. The resulting formalism is very flexible and can account for sparse, noisy observations and non-Gaussian error statistics. It is illustrated by simple one– and two–dimensional examples that show the richness of the new

types of analysis offered by this unification.

## 1 Introduction

### 1.1 Weakness of classical data assimilation

Geophysical data assimilation is a set of methods and algorithms at the intersection of Earth sciences, mathematics, and computer science, designed to enhance our understanding and predictive capabilities of the complex systems that govern our

planet (Carrassi et al., 2018). These systems encompass the atmosphere, ocean, atmospheric chemistry and biogeochemistry, land surfaces, glaciology, hydrology, etc., and as a whole the climate system. Data assimilation is meant to optimally combine all sources of quantitative information, typically past and present observations, and numerical and statistical models of the system under consideration. Data assimilation (DA) is critical in forecasting chaotic geofluids by resetting the initial conditions of the flow, estimating physical and statistical parameters of the models, and providing a quantitative re-analysis of the past

history of the climate system over decades. Because classical DA is applied to complex and high-dimensional dynamics, the





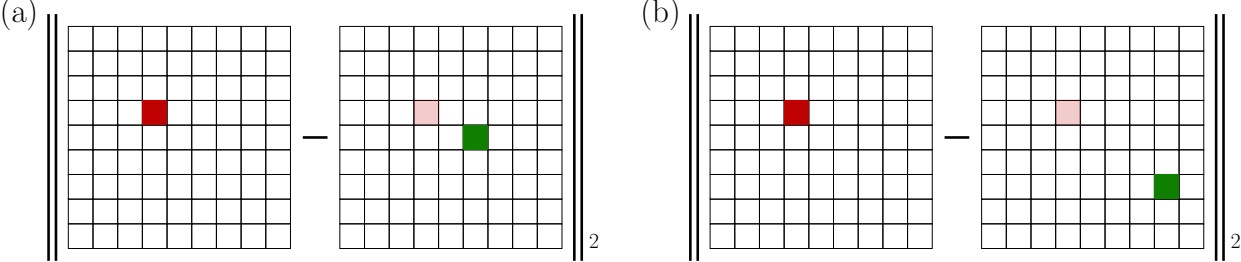

**Figure 1.** These two panels schematise the computation of the RMSE of two analysis increments. Those increments are the difference between the truth (left mesh within both Euclidean norm), concentrated here in the red grid-cell and the analysis located in the green grid-cells (right mesh within both Euclidean norm). The increment on panel (a) is the outcome of a better analysis spatially closer to the truth, as compared to the one in panel (b), and yet both increments yield the same RMSE. This verification metrics is hence impacted by the double-penalty error and does not help in discriminating location errors.

DA algorithms often result from a compromise between the sophistication of the employed mathematical techniques and their numerical scalability and efficiency (Kalnay, 2003; Asch et al., 2016; Evensen et al., 2022). For instance, it is well-known that most DA methods are built around or from an update step – the analysis – where observations and background states are combined, an operation which often relies on Gaussian statistical assumptions.

Here we would like to focus on two other important weaknesses of classical DA. Firstly, the *double-penalty error* in geosciences, which refers to the over-penalisation of errors in both the model and observational data (e.g. Amodei and Stein, 2009), compromises the balance required for effective DA. A typical example is given by a slight mislocation of a plume of pollutant resulting in high predicted concentration values at positions where no pollutant is observed while the model misses the observed peaks of concentration (Farchi et al., 2016). This mismatch is heavily penalised because of the use, over the same

discretised space, of the root-mean square error (RMSE) utilised for a point-by-point comparison. Figure 1 shows an exemplar for such double-penalty error resulting in the inability to properly evaluate a model and learn from an analysis increment. This double-penalty error, a very common contribution to the *representation error* (Janjić et al., 2018), is ubiquitous in the geosciences: in numerical weather prediction and in particular for water vapour, in atmospheric chemistry, in eddy resolving ocean forecasting, etc.

A second weakness is the requirement in classical DA for prior fields, which could be the *first guess* or *background state* and the observations, with substantial overlap in both space and time. For instance, failing to do so is a known pathway for the degeneracy of particle filters in high-dimension, also known as the curse of dimensionality (e.g. Farchi and Bocquet, 2018, and references therein). But, even with Gaussian-based methods less prone to the curse of dimensionality, this can be seen as a drawback. Indeed, the update of classical DA would essentially interpolate in the space of the values of the fields, yielding an

analysis still confined within the support of the background state and that of the observation. This can be seen as a severe flaw if the mismatch in the observations and background state is due to an error in the location of the fields, or more generally when





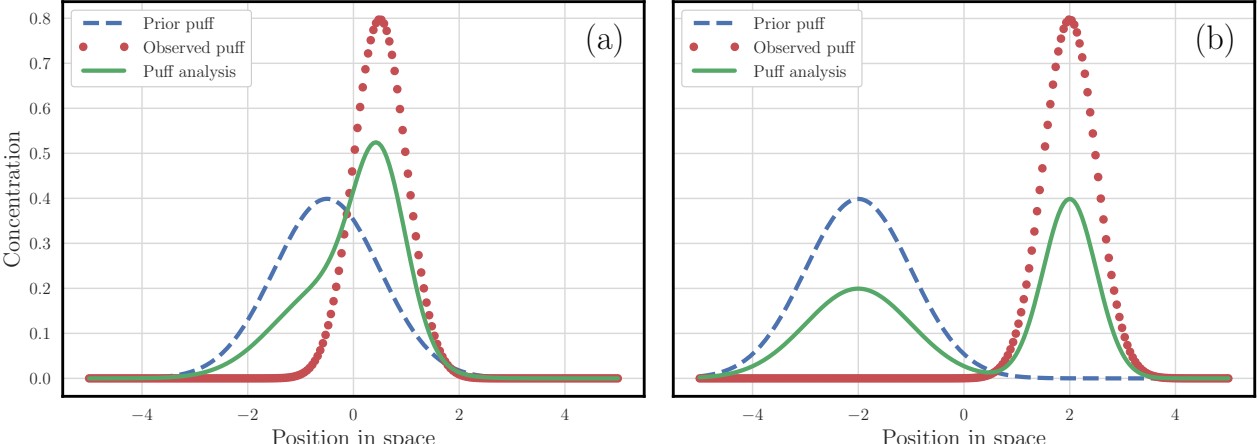

**Figure 2.** These two panels illustrate a reasonably beneficial and a probably useless classical DA update. It is assumed that the observations (red dots) and first guess (dashed blue curve), which represent one-dimensional puffs of pollutant, are subject to a location mismatch. With a significant overlap of these fields, panel (a) displays a consistent analysis. By contrast, in panel (b), the analysis state (green curve) fails to propose a state with a significant presence in between the observations and first guess, just as expected since classical DA is based on an interpolation in field values, but neither in space, nor in time.

these fields are misspecified. Figure 2 illustrates of couple of classical DA experiments with a beneficial analysis and a useless analysis.

### 1.2 Nonlocal verification metrics

In both cases, the issue can be ascribed to the use of a *local* verification metric, meaning that it compares, through the RMSE, values at the same site, of the same grid-cell. Thus, this issue goes beyond DA, and pertains to the use of local metrics.

To avoid being impacted by the double-penalty issue stemming from the use of local verification metrics, smarter *nonlocal* or *multiscale* metrics have been proposed. A typical metric of this kind consists in the combination of a displacement map followed by the use of classical norm such as the RMSE (Hoffman et al., 1995; Keil and Craig, 2009). In this vein, effective verification

metrics can be based on *optical flow-based warping*, or on *deformed meshes*, prior to using classical norms (Gilleland et al., 2010a, b). These metrics can also be designed as *scale-dependent* and possibly *multiscale*, based on an empirical separation of scales, such as with *fuzzy* metrics (Ebert, 2008; Amodei and Stein, 2009), or e.g., *wavelets* (Briggs and Levine, 1997). They can be designed to grasp and quantify objects and features, such as lows and highs (Davis et al., 2006a, b; Lack et al., 2010). Metrics with similar capabilities but not necessary based on a displacement concept, have been introduced in computer vision

such as the *structural similarity index* (Zhou et al., 2004), or in the verification of precipitations (Skok, 2023; Necker et al., 2023).

One of the most elegant approach is based on the theory of *optimal transport* (OT), and the associated *Wasserstein distance*, which sits on solid mathematical foundations and significant developments, which are the main reasons why we will focus





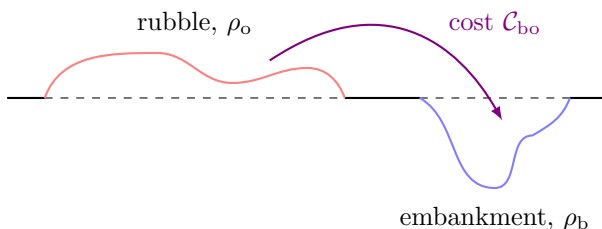

**Figure 3.** Illustration of the earth mover problem introduced by Monge in 1781 (see bulk of paper).

on OT in the following. Examples of application of OT to the verification of tracer and greenhouse gases models are given in
Farchi et al. (2016); Vanderbecken et al. (2023).

### 1.3   Optimal transport and the Wasserstein distance

Before mentioning applications of the Wasserstein distance in the field of geoscience, let us first give a very brief introduction
to the concept and mathematical formulation of OT.

The OT concept stemmed from an engineering but rather universal problem. Gaspard Monge (Monge, 1781) considered the
*earth mover* problem where the goal is to efficiently move rubble to an embankment of about the same volume (see Fig. 3).
Each displacement of a bit of earth has a known cost, so that the goal is to find the cheapest deterministic map that completely
moves the rubble to the embankment. In mathematical terms, the goal is to find the map of minimal cost that transports the
origin measure $\rho_{\mathrm{o}}$ to the target measure $\rho_{\mathrm{b}}$, where *measure* here means that both of them are non-negative, and are integrable
of integral 1. Note that the value 1 is arbitrary here and can be changed to $m > 0$, provided this is the mass of both $\rho_{\mathrm{o}}$ and $\rho_{\mathrm{b}}$.
The cost is defined by a non-negative function $\mathcal{C}_{\mathrm{bo}}$ of two variables (one for the origin space and the other for the target space).
Let us assume a quadratic cost, defined for any couple of points $(x, y)$ of a geometric domain $\Omega$:

$$\mathcal{C}_{\mathrm{bo}}(x, y) = \|x - y\|_2^2, \tag{1}$$

where $\|\cdot\|_2$ is the Euclidean norm. Let us define the set of all admissible differentiable maps $T$ that transport $\rho_{\mathrm{o}}$ to $\rho_{\mathrm{b}}$:

$$\mathcal{U}_{\mathrm{bo}} = \{T : \Omega \mapsto \Omega, \quad \rho_{\mathrm{o}} = |\partial_x T| \rho_{\mathrm{b}} \circ T\}, \tag{2}$$

where $|\partial_x T|$ is the absolute value of the determinant of the Jacobian of $T$, a factor which accounts for the deformation of the
measure by the globally mass-conserving $T$. The *square* of the Wasserstein distance $\mathcal{W}_{\mathcal{C}_{\mathrm{bo}}}$ is then defined by

$$\mathcal{W}_{\mathcal{C}_{\mathrm{bo}}}^2(\rho_{\mathrm{o}}, \rho_{\mathrm{b}}) = \min_{T \in \mathcal{U}_{\mathrm{bo}}} \int_{\Omega} \mathcal{C}_{\mathrm{bo}}(x, T(x)) \rho_{\mathrm{o}}(x) \mathrm{d}x, \tag{3}$$

whose purpose is to minimise the total transport cost between $\rho_{\mathrm{o}}$ and $\rho_{\mathrm{b}}$. It can be shown that $\mathcal{W}_{\mathcal{C}_{\mathrm{bo}}}$ is indeed a proper
mathematical distance. The mathematical formulation is deceptively simple since it is elegant, compact and easy to grasp, but
its theoretical and numerical solutions are far from trivial.



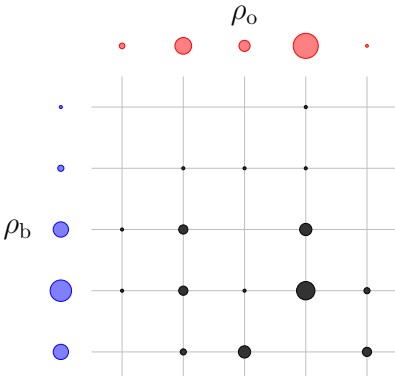

**Figure 4.** A representation of a discrete transport plan between two discrete origin (blue) and target (red) measures. The black dots represent the value of the transference plan. The radius of the dots are proportional to the values of these measures. This transference plan is checked to be admissible but is not necessarily optimal.

A breakthrough was made in the 20th century by Leonid Kantorovich who promoted the Monge problem to a probabilistic formulation. In his point of view, a bit of earth can be split and moved to many sites of the target measure support. The deterministic map $T$ is hence replaced with a probabilistic measure $\pi$ defined over $\Omega \times \Omega$. Such a $\pi$ is called hereafter a *transference plan*. An admissible transference plan is integrable and have $\rho_{\mathrm{o}}$ and $\rho_{\mathrm{b}}$ as one-variable marginals; hence the definition of the admissible set:

$$\mathcal{V}_{\mathrm{bo}} = \left\{ \pi : \Omega \times \Omega \mapsto \mathbb{R}_{+}, \quad \rho_{\mathrm{o}}(x) = \int_{\Omega} \pi(x,y)\mathrm{d}y, \quad \rho_{\mathrm{b}}(y) = \int_{\Omega} \pi(x,y)\mathrm{d}x \right\}. \tag{4}$$

As opposed to the deterministic Monge maps, the transference plans offer a symmetrical view on the origin and target space and their measures. An illustration of a discrete transference plan is given by Fig. 4. In this view the squared Wasserstein distance can be reformulated as

$$\mathcal{W}^2_{\mathcal{C}_{\mathrm{bo}}}(\rho_{\mathrm{o}}, \rho_{\mathrm{b}}) = \min_{\pi \in \mathcal{V}_{\mathrm{bo}}} \int_{\Omega \times \Omega} \mathcal{C}_{\mathrm{bo}}(x,y)\pi(x,y)\mathrm{d}x\mathrm{d}y. \tag{5}$$

Equations (3,5) are the consecrated continuous formulations of OT. In the rest of the paper, we will deal instead with *discrete* related formulations, more tangible, and amenable to algorithmic and numerical implementations.

The field has attracted a lot of attention from pure and applied mathematicians, and computer scientists. A complete introduction to the topic by its experts can be found in the stimulating text books by Vilani (2003, 2009); Peyré and Cuturi (2019). Peyré and Cuturi (2019) nicely provide concrete examples, numerical methods and a broad coverage of the topic from the perspective of applied mathematicians and computer scientists. Hence, it will be referred to quite often in the rest of the paper.



## 1.4 Nonlocal, multiscale metrics and data assimilation

Let us now go back to DA and narrow our focus to the use of advanced metrics in DA. Accounting for displacement error in DA and hence relying on nonlocal verification metrics has been advocated by Ravela et al. (2007); Plu (2013). Metrics built on a multiscale analysis of the fields to achieve a similar goal have been proposed by Ying (2019); Ying et al. (2023).

Wasserstein distance and closely related formulations, have been advocated in the flow formulation of the analysis (DA update) to seamlessly transport the prior to the posterior (El Moselhy and Marzouk, 2012; Oliver, 2014; Marzouk et al., 2017; Farchi and Bocquet, 2018; Tamang et al., 2020). It can for instance be used to adjust the posterior discrete probability density functions (pdf) in the particle filter. It has similarly been used to assist ensemble DA (Tamang et al., 2021, 2022). Finally, it has also very recently been used to compare forecast ensembles for sub-seasonal prediction (Le Coz et al., 2023; Lledó et al., 2023), or precipitation (Duc and Sawada, 2023).

In the context of this paper, it is *critical* to be aware that the use of OT in practical DA focused so far on applying OT to the pdf of a single variable. Quite often, OT is applied to the pdf of a single random variable because

- OT in one dimension (the space of the values taken by this random variable) together with the quadratic cost has a very simple solution that only relies on the cumulative distribution functions of the origin and target measures (see e.g., Remark 2.30 in Peyré and Cuturi, 2019).

- increasing the number of random variables is subject to the curse of dimensionality, necessitating an exponential increase in computational resources.

This is very different from our context and objective where the objects dealt with by OT are field states, not the pdf of one of their variables. In particular, although the computations are very demanding when the physical space that supports the fields scales to dimension 2 and 3, our problem is not subject to the curse of dimensionality.

The present paper stands more in the wake of the seminal proposal of Feyeux (2016); Feyeux et al. (2018). Their idea is to replace the local metrics of classical variational DA, typically the square of the Euclidean distance (hence related to the $L_2$ norm) by the squared Wasserstein distance. This is intuitively what we are after in order to cope with the two weaknesses mentioned in Sect. 1.1 in the context of DA. This should redefine the nature of the DA update step. Let us formalise this idea.

We will seize this opportunity to introduce some of our notation, in the context of discrete DA which is a widely adopted standpoint in the geosciences. Let us focus on a classical DA 3D-Var cost function (Daley, 1991):

$$G_{\mathrm{cl}}(\mathbf{x}^{\mathrm{a}}) = \left\| \mathbf{y}^{\mathrm{b}} - \mathbf{x}^{\mathrm{a}} \right\|_2^2 + \left\| \mathbf{y}^{\mathrm{o}} - \mathbf{H}\mathbf{x}^{\mathrm{a}} \right\|_2^2, \tag{6}$$

where $\|\mathbf{x}\|_2 = \sqrt{\sum_{i=1}^{N} x_i^2}$ is the Euclidean norm, $\mathbf{y}^{\mathrm{b}} \in \mathbb{R}^{N_{\mathrm{b}}}$ is the first guess, $\mathbf{y}^{\mathrm{o}} \in \mathbb{R}^{N_{\mathrm{o}}}$ is the vector of observations, and $\mathbf{H}$ is the observation operator. $\mathbf{x}^{\mathrm{a}} \in \mathbb{R}^{N_{\mathrm{a}}}$ is the dummy variable of this optimisation problem whose optimal value corresponds to the DA analysis. Now, the substitution of the Euclidean norm yields the new 3D-Var cost function:

$$G_{\mathrm{w}}(\mathbf{x}^{\mathrm{a}}) = \mathcal{W}_2^2(\mathbf{y}^{\mathrm{b}}, \mathbf{x}^{\mathrm{a}}) + \mathcal{W}_2^2(\mathbf{y}^{\mathrm{o}}, \mathbf{H}\mathbf{x}^{\mathrm{a}}), \tag{7}$$



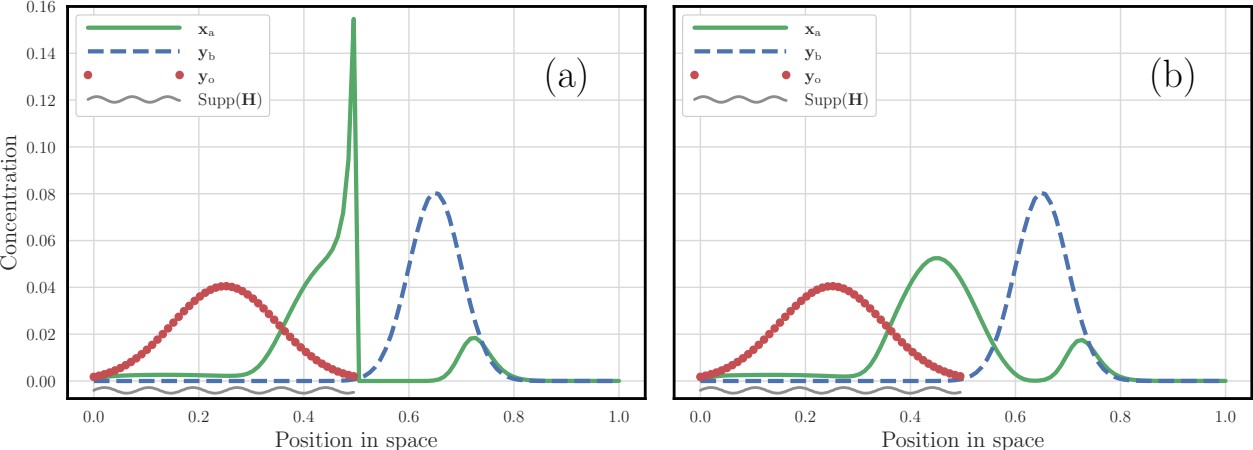

**Figure 5.** These panels illustrate the analysis of a 3D-Var that relies on the Wasserstein distance rather than a local metric. The red dots represent the observations, while the dashed blue curve represents the background state. The observations are only focused on the left half of the domain. The solution of the optimisation problem Eq. (7), is displayed on panel (a) as a solid green curve. The solution of the optimisation problem we will propose in this paper is displayed on panel (b) as a solid green curve. The support of the observation is suggested with a wavy grey segment. These states are typically one-dimensional puff pollutant concentrations. They should not be confused with pdfs of a single random variable.

where $\mathcal{W}_2$ is some discretisation of the Wasserstein distance based on the cost defined by the square of the Euclidean distance. Note that this 3D-Var requires balancing two instances of a Wasserstein-based metric. The analysis state is known as a

*Wasserstein barycentre*, abridged W-barycentre in the following.

Feyeux (2016); Feyeux et al. (2018) explored the optimisation aspects of this DA problem. However, Feyeux (2016) ultimately pointed to a possible inconsistency in the definition of the DA problem formulated in Eq. (7). In the case where the system is fully observed, typically when $\mathbf{H}$ is the identity operator, the outcome of the optimisation problem, i.e. the analysis, matches our expectations. However, when the system is partially observed, inconsistencies are observed. Let us see why.

Panel (a) of Fig. 5 considers the DA problem based on Eq. (7), assuming that only half of the domain is observed. We have solved the corresponding mathematical and numerical problem and displayed its solution. However, most of the mass of the solution concentrates on the observed subdomain, and neglects the rest of the domain where the prior mainly concentrates. Instead, we would have anticipated a solution close to the one offered by panel (b) of Fig. 5, whose formulation and numerical solution differ and follow the new approach developed in the present paper.

The main caveat comes from the implicit assumption that OT is *balanced*, i.e. the origin and target densities must have the same mass. This applies to both OTs in Eq. (7), between $\mathbf{y}^{\mathrm{b}}$ and $\mathbf{x}^{\mathrm{a}}$ and between $\mathbf{y}^{\mathrm{o}}$ and $\mathbf{H}\mathbf{x}^{\mathrm{a}}$:

$$\mathfrak{m}(\mathbf{x}^{\mathrm{a}}) = \mathfrak{m}(\mathbf{y}^{\mathrm{b}}), \qquad \mathfrak{m}(\mathbf{H}\mathbf{x}^{\mathrm{a}}) = \mathfrak{m}(\mathbf{y}^{\mathrm{o}}), \tag{8}$$





where the mass of the vector $\mathbf{x} \in \mathbb{R}^N$ is defined by

$$\mathfrak{m}(\mathbf{x}) = \mathbf{1}^\mathsf{T} \mathbf{x} = \sum_{i=1}^{N} x_i \tag{9}$$

with $\mathbf{1} \in \mathbb{R}^N$ hereafter defined as the vector of entries 1.

Now, if we further assume that $\mathbf{y}^\mathrm{b}$ and $\mathbf{y}^\mathrm{o}$ have the same mass, then

$$\mathfrak{m}(\mathbf{H}\mathbf{x}^\mathrm{a}) = \mathfrak{m}(\mathbf{y}^\mathrm{o}) = \mathfrak{m}(\mathbf{y}^\mathrm{b}) = \mathfrak{m}(\mathbf{x}^\mathrm{a}). \tag{10}$$

As a result, the mass equality $\mathfrak{m}(\mathbf{H}\mathbf{x}^\mathrm{a}) = \mathfrak{m}(\mathbf{x}^\mathrm{a})$ compels to find part of $\mathbf{x}^\mathrm{a}$ in $\mathbb{R}^{N_\mathrm{a}} \setminus \ker(\mathbf{H})$, in order to fully account for $\mathfrak{m}(\mathbf{x}^\mathrm{a})$. Hence this approach amounts to *finding the truth under the streetlight*, unless the system is fully observed.

To overcome this caveat and find a proper alternative to Eq. (7), we need to resort to *unbalanced OT*, i.e. we need to be able to accommodate states of distinct masses. In the computer science context of pure OT, such possibility has been discussed by Chizat et al. (2018). But our solution differs formally and will be DA-centric.

## 1.5    Objective and outline

The objective of this paper is to lift the objection of Feyeux (2016), and propose a DA framework based on the Wasserstein
distance, and hence to offer a consistent way to bridge OT and classical DA. The new formalism will be referred to as *hybrid OTDA* for *hybrid optimal transport data assimilation* in the rest of this paper, and often for the sake of brevity *OTDA*. We will mainly focus on the definition of a 3D-Var DA problem and how to obtain its analysis state and the associated analysis error covariance matrix.

At least within the perimeter of this paper, some restrictions apply compared to traditional DA. Firstly, the physical fields
considered in the DA problem are *non-negative* (concentration of tracer, pollutants, water vapour, hydrometeors, chemical and biogeochemical species, etc.). However, as opposed to Feyeux (2016), the methods of this paper do not require the (possibly noisy) background state $\mathbf{y}^\mathrm{b}$ and observation $\mathbf{y}^\mathrm{o}$ to be non-negative. Secondly, the observation operator $\mathbf{H}$ is assumed to be *linear*. This is only meant for convenience and to obtain a rigorous correspondence between the primal and dual cost functions of the 3D-Var. Finally, we stress once again that the states of our DA problem are physical fields onto which OT is applied and
are not meant to be pdf of a random variable.

The outline of the paper follows. After the present introduction, Sect. 2 discloses our main idea, and discusses two mathematical paths to solve the underlying optimisation problem, a first one which is enlightening but not necessarily practical and an alternative which is direct and robust but hides some of the concepts behind it. Section 3 provides one– and two– dimensional illustrations of a 3D-Var analysis based on the new hybrid OTDA formalism. These illustrations will show the
possibilities and flexibility of the new framework. This section will also depict classical DA as limit case of the formalism. In Sect. 4, the second-order analysis, i.e. the uncertainty quantification of the OTDA 3D-Var, is derived, discussed, and illustrated. Conclusions and perspectives are given in Sect. 5.



## 2 The main proposal

### 2.1 Notation and conventions

Non-negative vectors $\mathbf{x}$ of size $N$ are called *discrete measures*; they lie in the orthant $\mathcal{O}_N^+ \triangleq \mathbb{R}_+^N$. Although most mathematical OT theories work on *normalised* discrete measures, yielding *probability vectors*, this assumption won't be needed in this paper. The open subset of $\mathcal{O}_N^+$ of all the positive discrete measures will be denoted $\mathcal{O}_N^{+,\star} \triangleq \mathbb{R}_{+,\star}^N$.

We will distinguish the *observations* $\mathbf{y}^{\mathrm{b}} \in \mathbb{R}^{\mathfrak{N}_{\mathrm{b}}}$, $\mathbf{y}^{\mathrm{o}} \in \mathbb{R}^{\mathfrak{N}_{\mathrm{o}}}$ from the observable states $\mathbf{x}^{\mathrm{b}} \in \mathcal{O}_{N_{\mathrm{b}}}^+$ and $\mathbf{x}^{\mathrm{o}} \in \mathcal{O}_{N_{\mathrm{o}}}^+$. $\mathbf{y}^{\mathrm{b}}$, which corresponds to the first guess of conventional DA, and $\mathbf{y}^{\mathrm{o}}$, which corresponds to the traditional observation vector, are known before solving the 3D-Var problem. These vectors embody all the information processed in the analysis. By contrast, the
observables $\mathbf{x}^{\mathrm{b}}$ and $\mathbf{x}^{\mathrm{o}}$, which are related to $\mathbf{y}^{\mathrm{b}}$ and $\mathbf{y}^{\mathrm{o}}$, respectively, through an observation operator (the identity for $\mathbf{y}^{\mathrm{b}}$ and $\mathbf{x}^{\mathrm{b}}$), are not known a priori. They will be estimated together with the analysis state $\mathbf{x}^{\mathrm{a}} \in \mathcal{O}_{N_{\mathrm{a}}}^+$. Note that these vectors may well lie in distinct vector spaces of different dimensions, hence the introduction of as many dimensions $N_{\mathrm{b}}, N_{\mathrm{o}}, \mathfrak{N}_{\mathrm{b}}, \mathfrak{N}_{\mathrm{o}}$. $x_i^\star$ can be seen as the value taken by $\mathbf{x}^\star$ at site $\mathbf{r}_i^\star$, for $\star = \mathrm{b}, \mathrm{o}, \mathrm{a}$ and $i \in [\![1, N_\star]\!]$. Mind that the distinction between $\mathbf{y}^{\mathrm{b}}$ and $\mathbf{x}^{\mathrm{b}}$, and the
introduction of $\mathbf{x}^{\mathrm{o}}$ is a novelty of OTDA compared to classical DA.

Like in classical DA, the vectors $\mathbf{y}^{\mathrm{b}}$ and $\mathbf{y}^{\mathrm{o}}$ are subject to (prior) errors whose statistics are specified by the likelihoods $p(\mathbf{y}^{\mathrm{b}}|\mathbf{x}^{\mathrm{b}})$ and $p(\mathbf{y}^{\mathrm{o}}|\mathbf{x}^{\mathrm{o}})$, respectively. Up to constants that do not depend on $\mathbf{x}^{\mathrm{b}}, \mathbf{x}^{\mathrm{o}}, \mathbf{y}^{\mathrm{b}}, \mathbf{y}^{\mathrm{o}}$, we assume the existence of $\zeta_{\mathrm{b}}$ and $\zeta_{\mathrm{o}}$ such that

$$\ln p(\mathbf{y}^{\mathrm{b}}|\mathbf{x}^{\mathrm{b}}) \triangleq -\zeta_{\mathrm{b}}(\mathbf{y}^{\mathrm{b}} - \mathbf{x}^{\mathrm{b}}) + \mathrm{cst}, \qquad \ln p(\mathbf{y}^{\mathrm{o}}|\mathbf{x}^{\mathrm{o}}) \triangleq -\zeta_{\mathrm{o}}(\mathbf{y}^{\mathrm{o}} - \mathbf{H}\mathbf{x}^{\mathrm{o}}) + \mathrm{cst}, \tag{11}$$

so that various error statistics can be considered. These errors are hypothesised to be mutually independent. The observation operator $\mathbf{H}: \mathcal{O}_{N_{\mathrm{o}}}^+ \mapsto \mathbb{R}^{\mathfrak{N}_{\mathrm{o}}}$ used in the definition of $\zeta_{\mathrm{o}}$ is assumed to be linear. This qualification is for convenience and could be lifted if necessary. It is further assumed that $\zeta_{\mathrm{b}}$ and $\zeta_{\mathrm{o}}$ are strictly convex functions. This is for instance the case if we choose Gaussian error statistics yielding

$$\zeta_{\mathrm{b}}(\mathbf{e}_{\mathrm{b}}) = \frac{1}{2}\|\mathbf{e}_{\mathrm{b}}\|_{\mathbf{B}^{-1}}^2, \qquad \zeta_{\mathrm{o}}(\mathbf{e}_{\mathrm{o}}) = \frac{1}{2}\|\mathbf{e}_{\mathrm{o}}\|_{\mathbf{R}^{-1}}^2, \tag{12}$$

where $\|\mathbf{e}\|_{\mathbf{A}} = \sqrt{\mathbf{e}^{\mathsf{T}}\mathbf{A}\mathbf{e}}$. $\mathbf{B}$ is the positive definite background error covariance matrix, and $\mathbf{R}$ is the positive definite observation error covariance matrix. Finally, the $\mathfrak{m}(\star)$ operator will act in the following on not only vectors but, more generally, on any tensor and will return the sum of all of its entries.

### 2.2 Formalism of discrete optimal transport

To discretise and solve the continuous Kantorovich optimisation problem introduced in Sect. 1.3, we will need two elementary
pieces about OT. These are not the only techniques we will leverage, but both represent cornerstones towards a numerical solution to our proposal, and hence they need a proper introduction.



### 2.2.1 The primal cost function

Let us consider two discrete measures $\mathbf{x}^{\mathrm{b}} \in \mathcal{O}_{N_{\mathrm{b}}}^{+}$ and $\mathbf{x}^{\mathrm{o}} \in \mathcal{O}_{N_{\mathrm{o}}}^{+}$ having the same mass

$$m \overset{\Delta}{=} \mathfrak{m}(\mathbf{x}^{\mathrm{b}}) = \mathfrak{m}(\mathbf{x}^{\mathrm{o}}). \tag{13}$$

For convenience, $\mathcal{O}_{\mathrm{b,o}}^{+}$ will be used as an alias for the set $\mathcal{O}_{N_{\mathrm{b}} \times N_{\mathrm{o}}}^{+}$. A cost matrix $\mathbf{C}_{\mathrm{bo}} \in \mathcal{O}_{\mathrm{b,o}}^{+}$ is given. The optimisation problem will be formulated using discrete Kantorovich transference plans $\mathbf{P}^{\mathrm{bo}} \in \mathcal{O}_{\mathrm{b,o}}^{+}$. The optimal discrete transference plan is given by the minimiser of the following optimisation problem:

$$W_{\mathbf{C}_{\mathrm{bo}}}(\mathbf{x}^{\mathrm{b}}, \mathbf{x}^{\mathrm{o}}) \overset{\Delta}{=} \min_{\mathbf{P}^{\mathrm{bo}} \in \mathcal{U}_{\mathrm{bo}}(\mathbf{x}^{\mathrm{b}}, \mathbf{x}^{\mathrm{o}})} \mathrm{Tr}(\mathbf{C}_{\mathrm{bo}}^{\mathsf{T}} \mathbf{P}^{\mathrm{bo}}), \tag{14a}$$

where the trace sums up the costs attached to each path, and the set of admissible transference plans is defined by

$$\mathcal{U}_{\mathrm{bo}} \overset{\Delta}{=} \left\{ \mathbf{P} \in \mathcal{O}_{\mathrm{b,o}}^{+} : \quad \mathbf{P}\mathbf{1}_{\mathrm{o}} = \mathbf{x}^{\mathrm{b}}, \quad \mathbf{P}^{\mathsf{T}}\mathbf{1}_{\mathrm{b}} = \mathbf{x}^{\mathrm{o}} \right\}, \tag{14b}$$

which selects the discrete transference plans with the proper marginals. $W_{\mathbf{C}_{\mathrm{bo}}}$ could be viewed as a discrete equivalent to *the square of* the Wasserstein distance $\mathcal{W}_{\mathcal{C}_{\mathrm{bo}}}^{2}$ introduced in Eq. (5).

### 2.2.2 Entropic regularisation

Adding to the fact that the optimisation problem Eq. (14a) is constrained, it may neither be convex, nor exhibit a single
minimum. Hence, *entropic regularisation* is used to render the problem strictly convex. A comprehensive justification is given by Peyré and Cuturi (2019). Note however, that we will use a *Kullback-Leibler divergence* (KL) regularisation term to be inserted in Eq. (14a),

$$\mathrm{Tr}(\mathbf{C}_{\mathrm{bo}}^{\mathsf{T}} \mathbf{P}^{\mathrm{bo}}) \rightarrow \mathrm{Tr}(\mathbf{C}_{\mathrm{bo}}^{\mathsf{T}} \mathbf{P}^{\mathrm{bo}}) + \varepsilon \mathcal{K}(\mathbf{P}^{\mathrm{bo}} | \boldsymbol{\nu}^{\mathrm{bo}}), \tag{15}$$

which incorporates some prior transference plan $\boldsymbol{\nu}^{\mathrm{bo}}$ and does not require $\mathfrak{m}(\mathbf{P}^{\mathrm{bo}}) = 1$, whereas Peyré and Cuturi (2019) opted
for a basic entropy term. The KL term (Boyd and Vandenberghe, 2004) is defined by

$$\mathcal{K}(\mathbf{p} | \mathbf{q}) \overset{\Delta}{=} \sum_{i} p_{i} \ln \frac{p_{i}}{q_{i}} - p_{i} + q_{i}. \tag{16}$$

We choose, e.g., $\boldsymbol{\nu}^{\mathrm{bo}} = \mathbf{x}^{\mathrm{b}}(\mathbf{x}^{\mathrm{o}})^{\mathsf{T}}/m$, and $\varepsilon > 0$ which is the regularisation scalar parameter. Note that this particular $\boldsymbol{\nu}^{\mathrm{bo}}$ is an admissible transference plan, i.e. it belongs to $\mathcal{U}_{\mathrm{bo}}$, and can be interpreted as a complete statistical decoupling of the transference plan with respect to the origin and target discrete measures. In the limit $\varepsilon \rightarrow 0^{+}$ of vanishing regularisation,
the solution should not depend on the choice of $\boldsymbol{\nu}^{\mathrm{bo}}$. However, the convergence to the solution at finite $\varepsilon$ may depend on this choice. The primal cost function augmented with such an entropic regularisation is usually solved numerically using the iterative *Sinkhorn algorithm* (Sinkhorn, 1964). However, this is not the path followed in this paper, although we have used it as well.

Finally note that the technique to convexify such an optimisation problem with a KL term has been introduced in DA by
Bocquet (2009); Bocquet et al. (2011) following principles of statistical physics.



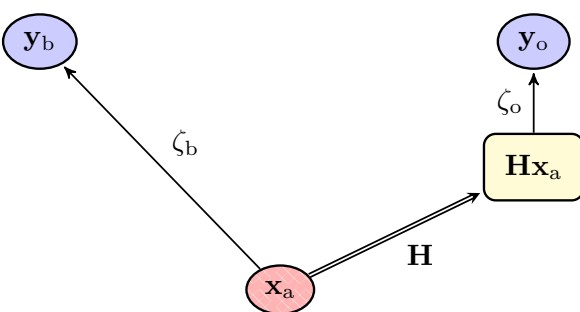

**Figure 6.** A diagrammatic representation of the classical 3D-Var update, with the observations $\mathbf{y}^{\mathrm{b}}$ (the first guess) and $\mathbf{y}^{\mathrm{o}}$ (the observation vector), the analysis state $\mathbf{x}^{\mathrm{a}}$, and the observed analysis $\mathbf{Hx}^{\mathrm{a}}$. A double line arrow represents a deterministic map, whereas a single line arrow represents a statistical binding between the origin and the target.

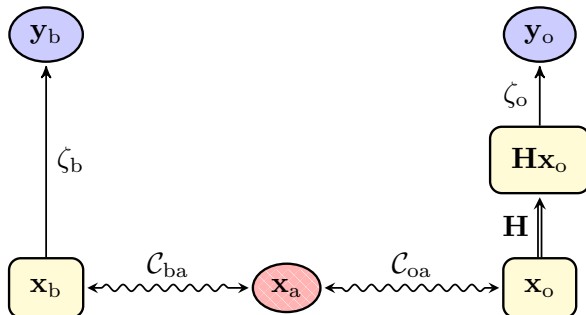

**Figure 7.** A diagrammatic representation of the hybrid OTDA 3D-Var update, with the observations $\mathbf{y}^{\mathrm{b}}$ (the first guess) and $\mathbf{y}^{\mathrm{o}}$ (the observation vector), the observables $\mathbf{x}^{\mathrm{b}}$, $\mathbf{x}^{\mathrm{o}}$, and $\mathbf{x}^{\mathrm{a}}$ which is the W-barycentre. A double line arrow represents a deterministic map, a single line arrow represents a statistical binding between the origin and the target, and a waving line represents the *weaker* bindings of $\mathbf{x}^{\mathrm{b}}$ and $\mathbf{x}^{\mathrm{a}}$ and $\mathbf{x}^{\mathrm{o}}$ and $\mathbf{x}^{\mathrm{a}}$ through OTs. This diagram can be seen as an unfolding of that of Fig. 6.

## 2.3 From classical data assimilation to hybrid optimal transport data assimilation

Figure 6 is a schematic representation of the flow of information in a classical DA update, and in particular 3D-Var, using the notation introduced above. In this case, the observables $\mathbf{x}^{\mathrm{b}}$, $\mathbf{x}^{\mathrm{o}}$, and the analysis state $\mathbf{x}^{\mathrm{a}}$ coincide by construction, hence $\mathbf{x}^{\mathrm{b}}$ and $\mathbf{x}^{\mathrm{o}}$ are not needed. This diagram, which could also be seen as a Bayesian network, corresponds to the cost function

$$L_{\mathrm{cl}}(\mathbf{x}^{\mathrm{a}}) = \zeta_{\mathrm{b}}(\mathbf{y}^{\mathrm{b}} - \mathbf{x}^{\mathrm{a}}) + \zeta_{\mathrm{o}}(\mathbf{y}^{\mathrm{o}} - \mathbf{Hx}^{\mathrm{a}}), \tag{17}$$

to be minimised over $\mathbf{x}^{\mathrm{a}}$. Now let us make use of the observables $\mathbf{x}^{\mathrm{b}}$ and $\mathbf{x}^{\mathrm{o}}$ as new degrees of freedom but bind them by OTs to $\mathbf{x}^{\mathrm{a}}$, using the cost matrices $\mathbf{C}_{\mathrm{ba}}$ and $\mathbf{C}_{\mathrm{oa}}$, respectively. This yields the diagram in Fig. 7, which corresponds to the cost



function

$$L_{\mathrm{w}}(\mathbf{x}^{\mathrm{a}}) = \min_{\mathbf{x}^{\mathrm{b}} \in \mathcal{O}_{N_{\mathrm{b}}}^{+},\, \mathbf{x}^{\mathrm{o}} \in \mathcal{O}_{N_{\mathrm{o}}}^{+}} \left\{ \zeta_{\mathrm{b}}(\mathbf{y}^{\mathrm{b}} - \mathbf{x}^{\mathrm{b}}) + \zeta_{\mathrm{o}}(\mathbf{y}^{\mathrm{o}} - \mathbf{H}\mathbf{x}^{\mathrm{o}}) + W_{\mathbf{C}_{\mathrm{ba}}}(\mathbf{x}^{\mathrm{b}}, \mathbf{x}^{\mathrm{a}}) + W_{\mathbf{C}_{\mathrm{oa}}}(\mathbf{x}^{\mathrm{o}}, \mathbf{x}^{\mathrm{a}}) \right\}. \tag{18}$$

It must be minimised over $\mathbf{x}^{\mathrm{a}}$, yielding an analysis state $\mathbf{x}^{\mathrm{a}}$ which can also be seen as the W-barycentre between $\mathbf{x}^{\mathrm{b}}$ and $\mathbf{x}^{\mathrm{o}}$. Note that $\mathbf{x}^{\mathrm{b}}$ and $\mathbf{x}^{\mathrm{o}}$ are discrete measures of unknown mass. For the optimisation problem, they lie in $\mathcal{O}_{N_{\mathrm{b}}}^{+}$ and $\mathcal{O}_{N_{\mathrm{o}}}^{+}$, respectively. We call Eq. (18) a *high-level* primal cost function because the metrics $W_{\mathbf{C}_{\mathrm{ba}}}$ and $W_{\mathbf{C}_{\mathrm{oa}}}$ have not yet been replaced by their transference plan expression as opposed to, e.g., Eq. (14a). Passing to a lower level primal cost function would require to expand Eq. (18) using Eq. (14a) twice.

In the subsequent two subsections, we will investigate two pathways to solve the optimisation problem Eq. (18). The first path, Sect. 2.4, unveils some of the key concepts behind its solution, and partially disentangle the classical DA part from the W-barycentre part of the full analysis. This approach is enlightening but not necessarily practical. The second path is an alternative which is direct and robust but hides some of the fundamental principles underlying the solution. The busy reader could skip directly to the latter, i.e. Sect. 2.5.

## 2.4 Decomposition of the optimisation problem and effective cost metric

In this section, key ideas behind the minimisation of Eq. (18) are sketched and discussed. The level of mathematical rigour of this section is that of casual methodological DA in the geoscience literature, not at the level of applied mathematics. However, we stress that all the algorithms discussed here have been tested numerically successfully on various configurations. The solution of Eq. (18) presented in this section is not necessarily robust, but it is enlightening and hence worth discussing.

Repeated contravariant indices – meaning the same tensor index present as upper and lower index – in tensor expressions will be understood as summed over, following Einstein's convention.

### 2.4.1 Dual formulation of the full primal problem

One way, though not the only one, to write the explicit primal problem associated to Eq. (18) is through the use of a *gluing* transference plan $\mathbf{P}^{\mathrm{boa}} \in \mathcal{O}_{\mathrm{b,o,a}}^{+}$ where $\mathcal{O}_{\mathrm{b,o,a}}^{+} = \mathbb{R}_{+}^{N_{\mathrm{b}} N_{\mathrm{o}} N_{\mathrm{a}}}$ (see p.11-12 of Vilani, 2009), a 3–tensor whose marginals are $\mathbf{x}^{\mathrm{b}}$,

$\mathbf{x}^{\mathrm{o}}$, and $\mathbf{x}^{\mathrm{a}}$ and that glues the transference plans $\mathbf{P}^{\mathrm{ba}}$ between $\mathbf{x}^{\mathrm{b}}$ and $\mathbf{x}^{\mathrm{a}}$ and $\mathbf{P}^{\mathrm{oa}}$ between $\mathbf{x}^{\mathrm{o}}$ and $\mathbf{x}^{\mathrm{a}}$:

$$\mathcal{L} = \min_{\mathbf{x}^{\mathrm{a}} \in \mathcal{O}_{\mathrm{a}}^{+}} L_{\mathrm{w}}(\mathbf{x}^{\mathrm{a}}), \tag{19a}$$

$$= \min_{\mathbf{x}^{\mathrm{b}} \in \mathcal{O}_{\mathrm{b}}^{+},\, \mathbf{x}^{\mathrm{o}} \in \mathcal{O}_{\mathrm{o}}^{+},\, \mathbf{x}^{\mathrm{a}} \in \mathcal{O}_{\mathrm{a}}^{+}} \left[ \zeta_{\mathrm{b}}(\mathbf{y}^{\mathrm{b}} - \mathbf{x}^{\mathrm{b}}) + \zeta_{\mathrm{o}}(\mathbf{y}^{\mathrm{o}} - \mathbf{H}\mathbf{x}^{\mathrm{o}}) + \min_{\mathbf{P} \in \mathcal{U}_{\mathrm{boa}}} \left\{ P_{ijk} C_{\mathrm{ba}}^{ik} + P_{ijk} C_{\mathrm{oa}}^{jk} \right\} \right]. \tag{19b}$$

where the admissible set of (glued) transference plans, the set of all 3–tensors of non negative entries whose marginals are $\mathbf{x}^{\mathrm{b}}$, $\mathbf{x}^{\mathrm{o}}$ and $\mathbf{x}^{\mathrm{a}}$, is defined by

$$\mathcal{U}_{\mathrm{boa}} \triangleq \left\{ \mathbf{P} \in \mathcal{O}_{\mathrm{b,o,a}}^{+} : \quad \forall i,\, P_{ijk} 1_{\mathrm{o}}^{j} 1_{\mathrm{a}}^{k} = x_{i}^{\mathrm{b}}, \quad \forall j,\, P_{ijk} 1_{\mathrm{b}}^{i} 1_{\mathrm{a}}^{k} = x_{j}^{\mathrm{o}}, \quad \forall k,\, P_{ijk} 1_{\mathrm{b}}^{i} 1_{\mathrm{o}}^{j} = x_{k}^{\mathrm{a}} \right\}. \tag{19c}$$



Because of the hardly scalable dimensionality of the primal problem, based on either a 3–tensor, or a couple of 2–tensors, we wish to derive a dual problem equivalent to the primal one, using Lagrange multipliers to lift the constraints with, as will be checked later, a significantly smaller dimensionality. This leads to the following series of transformation of the problem, from a Lagrangian to a dual cost function:

$$\mathcal{L} = \min_{\mathbf{x}^{\mathrm{b}}\,\mathbf{x}^{\mathrm{o}}\,\mathbf{x}^{\mathrm{a}}}\left[\max_{\mathbf{f}_{\mathrm{b}}}\{(\mathbf{y}^{\mathrm{b}}-\mathbf{x}^{\mathrm{b}})^{\mathsf{T}}\mathbf{f}_{\mathrm{b}}-\zeta_{\mathrm{b}}^{*}(\mathbf{f}_{\mathrm{b}})\}+\max_{\mathbf{f}_{\mathrm{o}}}\{(\mathbf{y}^{\mathrm{o}}-\mathbf{H}\mathbf{x}^{\mathrm{o}})^{\mathsf{T}}\mathbf{f}_{\mathrm{o}}-\zeta_{\mathrm{o}}^{*}(\mathbf{f}_{\mathrm{o}})\}\right.$$

$$+\min_{\mathbf{P}\in\mathcal{O}_{\mathrm{b,o,a}}^{+}}\left\{P_{ijk}C_{\mathrm{ba}}^{ik}+P_{ijk}C_{\mathrm{oa}}^{jk}\right.$$

$$\left.\left.+\max_{\mathbf{h}_{\mathrm{b}}\,\mathbf{h}_{\mathrm{o}}\,\mathbf{f}_{\mathrm{a}}}\left\{h_{i}^{\mathrm{b}}\big(x_{i}^{\mathrm{b}}-P_{ijk}1_{\mathrm{o}}^{j}1_{\mathrm{a}}^{k}\big)+h_{j}^{\mathrm{o}}\big(x_{j}^{\mathrm{o}}-P_{ijk}1_{\mathrm{b}}^{i}1_{\mathrm{a}}^{k}\big)+f_{\mathrm{a}}^{k}\big(x_{k}^{\mathrm{a}}-P_{ijk}1_{\mathrm{b}}^{i}1_{\mathrm{o}}^{j}\big)\right\}\right\}\right], \tag{20a}$$

$$= \min_{\mathbf{x}^{\mathrm{a}}\,\mathbf{x}^{\mathrm{b}}\,\mathbf{x}^{\mathrm{o}}}\left[\max_{\mathbf{f}_{\mathrm{b}},\mathbf{f}_{\mathrm{o}}}\{(\mathbf{y}^{\mathrm{b}}-\mathbf{x}^{\mathrm{b}})^{\mathsf{T}}\mathbf{f}_{\mathrm{b}}-\zeta_{\mathrm{b}}^{*}(\mathbf{f}_{\mathrm{b}})+(\mathbf{y}^{\mathrm{o}}-\mathbf{H}\mathbf{x}^{\mathrm{o}})^{\mathsf{T}}\mathbf{f}_{\mathrm{o}}-\zeta_{\mathrm{o}}^{*}(\mathbf{f}_{\mathrm{o}})\}\right.$$

$$+\max_{\mathbf{h}_{\mathrm{b}}\,\mathbf{h}_{\mathrm{o}}\,\mathbf{f}_{\mathrm{a}}}\min_{\mathbf{P}\in\mathcal{O}_{\mathrm{b,o,a}}^{+}}\left\{P_{ijk}C_{\mathrm{ba}}^{ik}+P_{ijk}C_{\mathrm{oa}}^{jk}\right.$$

$$\left.\left.+h_{i}^{\mathrm{b}}\big(x_{i}^{\mathrm{b}}-P_{ijk}1_{\mathrm{o}}^{j}1_{\mathrm{a}}^{k}\big)+h_{j}^{\mathrm{o}}\big(x_{j}^{\mathrm{o}}-P_{ijk}1_{\mathrm{b}}^{i}1_{\mathrm{a}}^{k}\big)+f_{\mathrm{a}}^{k}\big(x_{k}^{\mathrm{a}}-P_{ijk}1_{\mathrm{b}}^{i}1_{\mathrm{o}}^{j}\big)\right\}\right], \tag{20b}$$

$$= \max_{\substack{\mathbf{h}_{\mathrm{b}}\,\mathbf{h}_{\mathrm{o}}\,\mathbf{f}_{\mathrm{a}}\\ \mathbf{f}_{\mathrm{b}}\,\mathbf{f}_{\mathrm{o}}}}\min_{\substack{\mathbf{x}^{\mathrm{a}}\,\mathbf{x}^{\mathrm{b}}\,\mathbf{x}^{\mathrm{o}}\\ \mathbf{P}\in\mathcal{O}_{\mathrm{b,o,a}}^{+}}}\left[(\mathbf{y}^{\mathrm{b}}-\mathbf{x}^{\mathrm{b}})^{\mathsf{T}}\mathbf{f}_{\mathrm{b}}-\zeta_{\mathrm{b}}^{*}(\mathbf{f}_{\mathrm{b}})+(\mathbf{y}^{\mathrm{o}}-\mathbf{H}\mathbf{x}^{\mathrm{o}})^{\mathsf{T}}\mathbf{f}_{\mathrm{o}}-\zeta_{\mathrm{o}}^{*}(\mathbf{f}_{\mathrm{o}})\right.$$

$$\left.+P_{ijk}C_{\mathrm{ba}}^{ik}+P_{ijk}C_{\mathrm{oa}}^{jk}+h_{i}^{\mathrm{b}}\big(x_{i}^{\mathrm{b}}-P_{ijk}1_{\mathrm{o}}^{j}1_{\mathrm{a}}^{k}\big)+h_{j}^{\mathrm{o}}\big(x_{j}^{\mathrm{o}}-P_{ijk}1_{\mathrm{b}}^{i}1_{\mathrm{a}}^{k}\big)+f_{\mathrm{a}}^{k}\big(x_{k}^{\mathrm{a}}-P_{ijk}1_{\mathrm{b}}^{i}1_{\mathrm{o}}^{j}\big)\right], \tag{20c}$$

$$= \max_{\mathbf{f}_{\mathrm{b}}\,\mathbf{f}_{\mathrm{o}}}\{\mathbf{f}_{\mathrm{b}}^{\mathsf{T}}\mathbf{y}^{\mathrm{b}}+\mathbf{f}_{\mathrm{o}}^{\mathsf{T}}\mathbf{y}^{\mathrm{o}}-\zeta_{\mathrm{b}}^{*}(\mathbf{f}_{\mathrm{b}})-\zeta_{\mathrm{o}}^{*}(\mathbf{f}_{\mathrm{o}})$$

$$+\min_{\substack{\mathbf{h}_{\mathrm{b}}\,\mathbf{h}_{\mathrm{o}}\,\mathbf{f}_{\mathrm{a}}\\ \mathbf{x}^{\mathrm{a}}\,\mathbf{x}^{\mathrm{b}}\,\mathbf{x}^{\mathrm{o}}\,\mathbf{P}\in\mathcal{O}_{\mathrm{b,o,a}}^{+}}}\left[(\mathbf{h}_{\mathrm{b}}-\mathbf{f}_{\mathrm{b}})^{\mathsf{T}}\mathbf{x}^{\mathrm{b}}+(\mathbf{h}_{\mathrm{o}}-\mathbf{H}^{\mathsf{T}}\mathbf{f}_{\mathrm{o}})^{\mathsf{T}}\mathbf{x}^{\mathrm{o}}+\mathbf{f}_{\mathrm{a}}^{\mathsf{T}}\mathbf{x}^{\mathrm{a}}\right.$$

$$\left.+P_{ijk}\big(C_{\mathrm{ba}}^{ik}+C_{\mathrm{oa}}^{jk}-h_{i}^{\mathrm{b}}1_{\mathrm{o}}^{j}1_{\mathrm{a}}^{k}-h_{j}^{\mathrm{o}}1_{\mathrm{b}}^{i}1_{\mathrm{a}}^{k}-f_{\mathrm{a}}^{k}1_{\mathrm{b}}^{i}1_{\mathrm{o}}^{j}\big)\big\}\right], \tag{20d}$$

$$= \max_{\mathbf{f}_{\mathrm{b}}\,\mathbf{f}_{\mathrm{o}}}\left[\mathbf{f}_{\mathrm{b}}^{\mathsf{T}}\mathbf{y}^{\mathrm{b}}+\mathbf{f}_{\mathrm{o}}^{\mathsf{T}}\mathbf{y}^{\mathrm{o}}-\zeta_{\mathrm{b}}^{*}(\mathbf{f}_{\mathrm{b}})-\zeta_{\mathrm{o}}^{*}(\mathbf{f}_{\mathrm{o}})\right.$$

$$\left.+\min_{\mathbf{P}\in\mathcal{O}_{\mathrm{b,o,a}}^{+}}P_{ijk}\Big(C_{\mathrm{ba}}^{ik}+C_{\mathrm{oa}}^{jk}-f_{\mathrm{b}}^{i}1_{\mathrm{o}}^{j}1_{\mathrm{a}}^{k}-H_{l}^{j}f_{\mathrm{o}}^{l}1_{\mathrm{b}}^{i}1_{\mathrm{a}}^{k}\Big)\right]. \tag{20e}$$

In Eq. (20a), the maps $\zeta_{\mathrm{b}}^{*}$ and $\zeta_{\mathrm{o}}^{*}$ are the Legendre-Fenchel transforms of the maps $\zeta_{\mathrm{b}}$ and $\zeta_{\mathrm{o}}$, respectively. Let us recall that the Legendre-Fenchel transform $\mathbf{f}\mapsto\zeta^{*}(\mathbf{f})$ of the map $\mathbf{e}\mapsto\zeta(\mathbf{e})$ is defined by $\zeta^{*}(\mathbf{f})=\sup_{\mathbf{e}}\{\mathbf{f}^{\mathsf{T}}\mathbf{e}-\zeta(\mathbf{e})\}$. For instance, in the

case of Gaussian error statistics as in Eq. (12), these transforms are given by

$$\zeta_{\mathrm{b}}^{*}(\mathbf{f}_{\mathrm{b}})=\frac{1}{2}\|\mathbf{f}_{\mathrm{b}}\|_{\mathbf{B}}^{2}, \qquad \zeta_{\mathrm{o}}^{*}(\mathbf{f}_{\mathrm{o}})=\frac{1}{2}\|\mathbf{f}_{\mathrm{o}}\|_{\mathbf{R}}^{2}. \tag{21}$$

From Eq. (20d) to Eq. (20e), taking the minimum over the observables $\mathbf{x}^{\mathrm{b}}$, $\mathbf{x}^{\mathrm{o}}$, and $\mathbf{x}^{\mathrm{a}}$ implies to enforce $\mathbf{h}_{\mathrm{b}}=\mathbf{f}_{\mathrm{b}}$, $\mathbf{h}_{\mathrm{o}}=\mathbf{H}^{\mathsf{T}}\mathbf{f}_{\mathrm{o}}$, and $\mathbf{f}_{\mathrm{a}}=\mathbf{0}$. Hence, we finally obtain the dual problem which only depends on the Lagrange multipliers:

$$\mathcal{L}^{*} = \max_{(\mathbf{f}_{\mathrm{b}},\mathbf{f}_{\mathrm{o}})\in\mathcal{U}_{\mathrm{bo}}^{*}(\mathbf{C}_{\mathrm{ba}},\mathbf{C}_{\mathrm{oa}},\mathbf{H})}\{\mathbf{f}_{\mathrm{b}}^{\mathsf{T}}\mathbf{y}^{\mathrm{b}}+\mathbf{f}_{\mathrm{o}}^{\mathsf{T}}\mathbf{y}^{\mathrm{o}}-\zeta_{\mathrm{b}}^{*}(\mathbf{f}_{\mathrm{b}})-\zeta_{\mathrm{o}}^{*}(\mathbf{f}_{\mathrm{o}})\}, \tag{22a}$$



where the $*$ symbol refers to *dual* and where the polyhedron $\mathcal{U}_{\mathrm{bo}}^*(\mathbf{C}_{\mathrm{ba}}, \mathbf{C}_{\mathrm{oa}}, \mathbf{H})$ is defined by

$$\mathcal{U}_{\mathrm{bo}}^*(\mathbf{C}_{\mathrm{ba}}, \mathbf{C}_{\mathrm{oa}}, \mathbf{H}) \triangleq \left\{ \mathbf{f}_{\mathrm{b}} \in \mathbb{R}^{\mathfrak{N}_{\mathrm{b}}}, \mathbf{f}_{\mathrm{o}} \in \mathbb{R}^{\mathfrak{N}_{\mathrm{o}}} : \quad \forall i,j,k, \quad f_{\mathrm{b}}^i + f_{\mathrm{o}}^l H_l^j \leq C_{\mathrm{ba}}^{ik} + C_{\mathrm{oa}}^{jk} \right\}. \tag{22b}$$

The inequality constraints of the polyhedron $\mathcal{U}_{\mathrm{bo}}^*$ stem from the positivity constraint $P_{ijk} \geq 0$ in Eq. (20e). Very importantly, we have the coincidence of the minimum of the primal problem with the maximum of the dual problem $\mathcal{L} = \mathcal{L}^*$, a property called *strong duality* (see Sect. 5.2 in Boyd and Vandenberghe, 2004). Strong duality can for instance be achieved if both the primal and dual cost functions are convex, which is the case here. This allows us to trade the primal for the dual problem. Since for each pair $\mathbf{f}_{\mathrm{b}}, \mathbf{f}_{\mathrm{o}}$ in $\mathcal{U}_{\mathrm{bo}}^*$, there are $N_{\mathrm{a}}$ constraints indexed by $k \in [\![1, N_{\mathrm{a}}]\!]$, and that the tightest of these constraints can account for the others, the problem Eq. (22) should be equivalent to

$$\mathcal{L}^* = \max_{(\mathbf{f}_{\mathrm{b}}, \mathbf{f}_{\mathrm{o}}) \in \mathcal{U}_{\mathrm{bo}}^*(\mathbf{C}_{\mathrm{bo}}, \mathbf{H})} \left\{ \mathbf{f}_{\mathrm{b}}^\mathsf{T} \mathbf{y}^{\mathrm{b}} + \mathbf{f}_{\mathrm{o}}^\mathsf{T} \mathbf{y}^{\mathrm{o}} - \zeta_{\mathrm{b}}^*(\mathbf{f}_{\mathrm{b}}) - \zeta_{\mathrm{o}}^*(\mathbf{f}_{\mathrm{o}}) \right\}, \tag{23a}$$

where the polyhedron $\mathcal{U}_{\mathrm{bo}}^*(\mathbf{C}_{\mathrm{bo}}, \mathbf{H})$ is defined by

$$\mathcal{U}_{\mathrm{bo}}^*(\mathbf{C}_{\mathrm{bo}}, \mathbf{H}) \triangleq \left\{ \mathbf{f}_{\mathrm{b}} \in \mathbb{R}^{\mathfrak{N}_{\mathrm{b}}}, \mathbf{f}_{\mathrm{o}} \in \mathbb{R}^{\mathfrak{N}_{\mathrm{o}}} : \quad \forall i,j, \quad f_{\mathrm{b}}^i + f_{\mathrm{o}}^l H_l^j \leq C_{\mathrm{bo}}^{ij} \right\}, \tag{23b}$$

and where the *effective cost metric* $\mathbf{C}_{\mathrm{bo}}$ is given by (in the absence of entropic regularisation)

$$[\mathbf{C}_{\mathrm{bo}}]_{ij} \triangleq \min_k \{ [\mathbf{C}_{\mathrm{ba}}]_{ik} + [\mathbf{C}_{\mathrm{oa}}]_{jk} \}. \tag{23c}$$

According to Eq. (23c), this effective cost is given by the cost of the cheapest path(s), which is intuitive. The optimal transference glued plan, $\mathbf{P}$, can be connected to the optimal transference plan $\mathbf{P}^{\mathrm{bo}}$ between $\mathbf{x}^{\mathrm{b}}$ and $\mathbf{x}^{\mathrm{o}}$ with the cost $\mathbf{C}_{\mathrm{bo}}$ in Eq. (23c), by marginalising on the intermediate density, i.e the W-barycentre

$$P_{ij}^{\mathrm{bo}} = P_{ijk} 1_{\mathrm{a}}^k. \tag{24}$$

The solution for the analysis state $\mathbf{x}^{\mathrm{a}}$ is given by

$$x_k^{\mathrm{a}} = P_{ijk} 1_{\mathrm{b}}^i 1_{\mathrm{o}}^j, \tag{25}$$

by the definition of the marginals of the gluing transference plan $\mathbf{P}$, Eq. (19c). Yet, we do not have a direct access to the optimal gluing $\mathbf{P}$ from the dual problem Eq. (23). This will be made simpler later on when adding the entropic regularisation to the problem.

For now, let us find an alternative solution bypassing the need for the gluing $\mathbf{P}$ and define the map

$$\kappa^{\mathrm{bo}} : [\![1, N_{\mathrm{b}}]\!] \times [\![1, N_{\mathrm{o}}]\!] \mapsto \mathcal{P}([\![1, N_{\mathrm{a}}]\!])$$
$$(i, j) \mapsto \kappa_{ij}^{\mathrm{bo}} = \arg\min_k \left( C_{\mathrm{ba}}^{ik} + C_{\mathrm{oa}}^{jk} \right), \tag{26}$$

where $\mathcal{P}(S)$ is defined as the set of all subsets of $S$. The set $\kappa_{ij}^{\mathrm{bo}}$ lists all the indices $k$ that are *relays* to the transport in between the sites corresponding to index $i$ and index $j$. That is why the W-barycentre can be obtained from $\mathbf{P}^{\mathrm{bo}}$:

$$x_k^{\mathrm{a}} = P_{ijk} 1_{\mathrm{b}}^i 1_{\mathrm{o}}^j = \sum_{ij} P_{ij}^{\mathrm{bo}} \delta_{k \in \kappa_{ij}^{\mathrm{bo}}}. \tag{27}$$





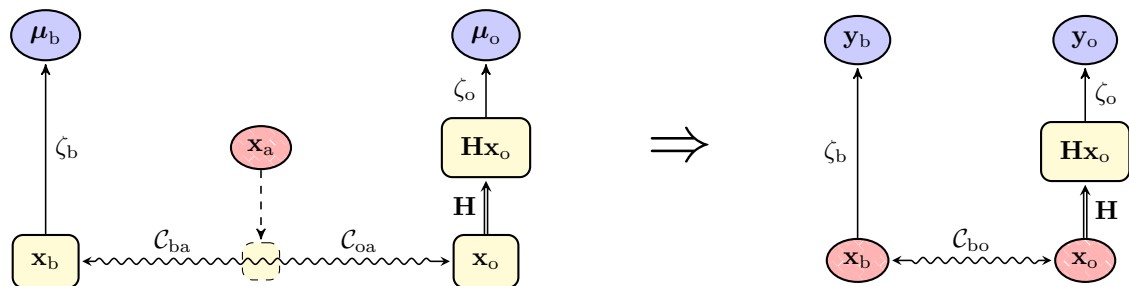

**Figure 8.** Trading a full hybrid OTDA problem characterised by a W-barycentre defined by the cost metrics $\mathbf{C}_{\mathrm{ba}}$ and $\mathbf{C}_{\mathrm{oa}}$, with a simplified hybrid OTDA problem characterised by a single OT problem defined by an effective cost metric $\mathbf{C}_{\mathrm{bo}}$.

We will show in the next section how to estimate $\mathbf{P}^{\mathrm{bo}}$ using entropic regularisation and hence leverage Eq. (27) to compute $\mathbf{x}^{\mathrm{a}}$. $\kappa^{\mathrm{bo}}$ is reminiscent of the so-called *McCann interpolant* in OT theory because it is only related to the OT between $\mathbf{x}^{\mathrm{b}}$ and $\mathbf{x}^{\mathrm{o}}$,

bypassing $\mathbf{x}^{\mathrm{a}}$, and hence the transference plan $\mathbf{P}^{\mathrm{bo}}$. Please refer to Remark 7.1 by Peyré and Cuturi (2019) and to Gangbo and McCann (1996) for a description of the McCann interpolant, even when there is no Monge map. This suggests that the analysis $\mathbf{x}^{\mathrm{a}}$ is not an interpolation of $\mathbf{x}^{\mathrm{b}}$ and $\mathbf{x}^{\mathrm{o}}$ in the space of values as for classical DA, but along a geodesic in a Riemannian space built on a metric derived from the Wasserstein distance.

Nonetheless, the above derivation shows that we can trade a W-barycentre problem characterised by a couple of OT problems

for a single OT problem defined by an effective metric $\mathbf{C}_{\mathrm{bo}}$. This principle is schematically illustrated by Fig. 8.

This suggests a simpler two-step algorithm, where the steps consist of (i) solving a hybrid OTDA problem but with a single OT problem under an effective cost metric, which yields the analysed observables $\mathbf{x}^{\mathrm{b}}$ and $\mathbf{x}^{\mathrm{o}}$ and then (ii) computing the W-barycentre of these $\mathbf{x}^{\mathrm{b}}$ and $\mathbf{x}^{\mathrm{o}}$. Let us now go through these two steps, while adding entropic regularisation to the problem.

### 2.4.2 First step: simplified hybrid optimal transport data assimilation problem

The first step of the full OTDA algorithm is hence a simplified OTDA problem based on a single OT problem driven by the cost $\mathbf{C}_{\mathrm{bo}}$. The corresponding high-level primal cost function is

$$\mathcal{L} = \min_{\mathbf{x}^{\mathrm{b}} \in \mathcal{O}_{\mathrm{b}}^+ \, \mathbf{x}^{\mathrm{o}} \in \mathcal{O}_{\mathrm{o}}^+} \left\{ \zeta_{\mathrm{b}}(\mathbf{y}^{\mathrm{b}} - \mathbf{x}^{\mathrm{b}}) + \zeta_{\mathrm{o}}(\mathbf{y}^{\mathrm{o}} - \mathbf{H}\mathbf{x}^{\mathrm{o}}) + W_{\mathbf{C}_{\mathrm{bo}}}(\mathbf{x}^{\mathrm{b}}, \mathbf{x}^{\mathrm{o}}) \right\}. \tag{28}$$

The associated (lower level) primal cost function, but adding entropic regularisation ($\varepsilon > 0$), is then

$$\mathcal{L}_\varepsilon = \min_{\mathbf{x}^{\mathrm{b}} \in \mathcal{O}_{\mathrm{b}}^+ \, \mathbf{x}^{\mathrm{o}} \in \mathcal{O}_{\mathrm{o}}^+} \left[ \zeta_{\mathrm{b}}(\mathbf{y}^{\mathrm{b}} - \mathbf{x}^{\mathrm{b}}) + \zeta_{\mathrm{o}}(\mathbf{y}^{\mathrm{o}} - \mathbf{H}\mathbf{x}^{\mathrm{o}}) + \min_{\mathbf{P} \in \mathcal{U}_{\mathrm{bo}}} \left( \varepsilon \mathcal{K}(\mathbf{P}|\boldsymbol{\nu}) + P_{ij} C_{\mathrm{bo}}^{ij} \right) \right]. \tag{29a}$$

In this optimisation problem, the admissible set of transference plans, i.e. the set of all 2–tensors of non negative entries whose marginals are $\mathbf{x}^{\mathrm{b}}$ and $\mathbf{x}^{\mathrm{o}}$, is defined by

$$\mathcal{U}_{\mathrm{bo}} \triangleq \left\{ \mathbf{P} \in \mathcal{O}_{\mathrm{b,o}}^+ : \quad \mathbf{P}\mathbf{1}_{\mathrm{o}} = \mathbf{x}^{\mathrm{b}}, \quad \mathbf{P}^{\mathsf{T}}\mathbf{1}_{\mathrm{b}} = \mathbf{x}^{\mathrm{o}} \right\}. \tag{29b}$$



Since $\mathbf{x}^{\mathrm{b}}$ and $\mathbf{x}^{\mathrm{o}}$ are not predetermined, the prior transference plan $\boldsymbol{\nu}$ cannot be selected from $\mathcal{U}_{\mathrm{bo}}$ a priori. The simplest choice, which we decided to implement, is hence to set $\nu_{ij}$ to a constant, which assumes some statistical prior independence of $\mathbf{x}^{\mathrm{b}}$ and

$\mathbf{x}^{\mathrm{o}}$. A derivation of the dual problem equivalent to $\mathcal{L}_{\varepsilon}$ can be obtained in the exact same way as in the previous subsection, although it is now less cluttered since there is only one OT to account for, instead of two. The associated Lagrangian is

$$
\begin{aligned}
\mathcal{L}_{\varepsilon} = \max_{\mathbf{f}_{\mathrm{b}} \in \mathbb{R}^{\mathfrak{N}_{\mathrm{b}}} \mathbf{f}_{\mathrm{o}} \in \mathbb{R}^{\mathfrak{N}_{\mathrm{o}}}} & \Big[ \mathbf{f}_{\mathrm{b}}^{\mathsf{T}} \mathbf{y}^{\mathrm{b}} + \mathbf{f}_{\mathrm{o}}^{\mathsf{T}} \mathbf{y}^{\mathrm{o}} - \zeta_{\mathrm{b}}^{*}(\mathbf{f}_{\mathrm{b}}) - \zeta_{\mathrm{o}}^{*}(\mathbf{f}_{\mathrm{o}}) \\
& + \min_{\mathbf{P} \in \mathcal{O}_{\mathrm{b,o}}^{+}} \left( \varepsilon \sum_{ij} \left\{ P_{ij} \ln \frac{P_{ij}}{\nu_{ij}} - P_{ij} + \nu_{ij} \right\} + P_{ij} \left\{ C_{\mathrm{bo}}^{ij} - f_{\mathrm{b}}^{i} 1_{\mathrm{o}}^{j} - H_{l}^{j} f_{\mathrm{o}}^{l} 1_{\mathrm{b}}^{i} \right\} \right) \Big].
\end{aligned}
\tag{30}
$$

Again, the maps $\zeta_{\mathrm{b}}^{*}$ and $\zeta_{\mathrm{o}}^{*}$ are the Legendre-Fenchel transforms of the maps $\zeta_{\mathrm{b}}$ and $\zeta_{\mathrm{o}}$. The variables $\mathbf{f}_{\mathrm{b}}$ and $\mathbf{f}_{\mathrm{o}}$ are Lagrange vectors; they are used to enforce the marginals of the transference plan associated to $W_{\mathbf{C}_{\mathrm{bo}}}$. The unconstrained minimisation

over $\mathbf{P}$, i.e. the inner minimisation problem in Eq. (30), is obtained by cancelling the gradient with respect to $\mathbf{P}$, which yields

$$
P_{ij} = \nu_{ij} e^{\left( f_{\mathrm{b}}^{i} + f_{\mathrm{o}}^{l} H_{l}^{j} - C_{\mathrm{bo}}^{ij} \right)/\varepsilon}.
\tag{31}
$$

Substituting this solution into minus the Lagrangian $-\mathcal{L}_{\varepsilon}$ gives the regularised dual problem

$$
\mathcal{J}_{\varepsilon}^{*} = \min_{\mathbf{f}_{\mathrm{b}} \in \mathbb{R}^{\mathfrak{N}_{\mathrm{b}}} \mathbf{f}_{\mathrm{o}} \in \mathbb{R}^{\mathfrak{N}_{\mathrm{o}}}} J_{\varepsilon}^{*}(\mathbf{f}_{\mathrm{b}}, \mathbf{f}_{\mathrm{o}}),
\tag{32a}
$$

with the associated Lagrangian

$$
J_{\varepsilon}^{*}(\mathbf{f}_{\mathrm{b}}, \mathbf{f}_{\mathrm{o}}) = \varepsilon(Z_{\varepsilon} - \mathfrak{m}(\boldsymbol{\nu})) + \zeta_{\mathrm{b}}^{*}(\mathbf{f}_{\mathrm{b}}) + \zeta_{\mathrm{o}}^{*}(\mathbf{f}_{\mathrm{o}}) - \mathbf{f}_{\mathrm{b}}^{\mathsf{T}} \mathbf{y}^{\mathrm{b}} - \mathbf{f}_{\mathrm{o}}^{\mathsf{T}} \mathbf{y}^{\mathrm{o}},
\tag{32b}
$$

which relies on the *partition function*

$$
Z_{\varepsilon} = \sum_{ij} P_{ij}.
\tag{32c}
$$

The notation $\mathcal{J}_{\varepsilon}^{*}$ and $J_{\varepsilon}^{*}$, rather than $\mathcal{L}_{\varepsilon}^{*}$ and $L_{\varepsilon}^{*}$, signifies that we work on the opposite of $\mathcal{L}_{\varepsilon}^{*}$ and $L_{\varepsilon}^{*}$ so as to obtain a dual problem to be minimised rather than maximised. Most importantly, we have, under conditions that will be satisfied in the

following, the coincidence of the two minima $\mathcal{J}_{\varepsilon}^{*} = -\mathcal{L}_{\varepsilon}$, i.e. strong duality. Assuming one can obtain a proper correspondence between the optimal $\mathbf{f}_{\mathrm{b}}$, $\mathbf{f}_{\mathrm{o}}$ of the dual problem and $\mathbf{x}^{\mathrm{b}}$, $\mathbf{x}^{\mathrm{o}}$ of the primal problem, this implies, once again, that the primal problem can be traded for the dual problem.

Even though the regularised optimisation problem is slightly different from the unregularised one, a difference which is controlled by the value of $\varepsilon$, the new dual optimisation problem is free, i.e. without constraints. It can be solved as it is, using

for instance the L-BFGS-B minimiser (Liu and Nocedal, 1989). The advantage of the regularised dual formulation is two-fold: the dual cost function is unconstrained (free optimisation) and we will trade a minimisation over $N_{\mathrm{b}} \times N_{\mathrm{o}}$ variables for a minimisation over $N_{\mathrm{b}} + N_{\mathrm{o}}$ variables. This dual formulation can be viewed as a generalised *Physical–space Statistical Analysis System* (PSAS) formalism (Courtier, 1997), an approach where classical DA algebra is mostly carried out in observation space.





Once the optimal values for $\mathbf{f}_b$ and $\mathbf{f}_o$ are obtained, the optimal discrete Kantorovich transference plan $\mathbf{P}$ can be computed using Eq. (31). As a result, as marginals of this transference plan, the solutions for the observables are

$$x_i^b = P_{ij} 1_o^j = \sum_j P_{ij}, \qquad x_j^o = P_{ij} 1_b^i = \sum_i P_{ij}. \tag{33}$$

### 2.4.3 Second step: Wasserstein barycentre

Now that we have obtained the observables $\mathbf{x}^b$ and $\mathbf{x}^o$ via Eq. (33), we would like to compute their W-barycentre. The joint mass $m$ of these observables can be computed:

$$m = \mathfrak{m}(\mathbf{x}^b) = \mathfrak{m}(\mathbf{x}^o). \tag{34}$$

The high-level primal cost function of this W-barycentre problem is

$$J_w = \min_{\mathbf{x}^a \in \mathcal{O}_{N_a}^+} \left\{ W_{\mathbf{C}_{ba}}(\mathbf{x}^b, \mathbf{x}^a) + W_{\mathbf{C}_{oa}}(\mathbf{x}^o, \mathbf{x}^a) \right\}. \tag{35}$$

We have found and practised several ways to solve this problem. One way is to compute the McCann interpolant. This is theoretically elegant but Eq. (27) did not leverage regularisation of the W-barycentre problem. Instead, the approach reported here is to use the dual optimisation problem, in conjunction with the entropic regularisation at finite $\varepsilon > 0$. We leverage our knowledge of the mass $m$ resulting from the first step of the algorithm by enforcing the mass in the cost function, $\mathfrak{m}(\mathbf{P}) = m$. This seems redundant but it actually yields *by construction* and very naturally a numerical efficient algorithm comparable to the *ad hoc* log-domain scheme proposed in Sect. 4.4 of Peyré and Cuturi (2019).

Again, one way, though not the only one, to write the primal problem goes trough the use of a gluing transference plan, a 3–tensor whose marginals are $\mathbf{x}^b$, $\mathbf{x}^o$, and $\mathbf{x}^a$:

$$L_\varepsilon = \min_{\substack{\mathbf{x}^a \in \mathcal{O}_{N_a}^+ \\ \mathbf{P} \in \mathcal{U}_{boa}(\mathbf{x}^a)}} \left\{ \mathbf{P} \cdot \mathbf{C}_{boa} + \varepsilon \mathcal{K}(\mathbf{P}|\boldsymbol{\nu}) + \mathbf{f}_b^\mathsf{T} \mathbf{x}^b + \mathbf{f}_o^\mathsf{T} \mathbf{x}^o \right\}, \tag{36a}$$

where $[\mathbf{C}_{boa}]_{ijk} = C_{ba}^{ik} + C_{oa}^{jk}$, the binary operator $\cdot$ denotes the contraction of tensors, and

$$\mathcal{U}_{boa}(\mathbf{x}^a) \triangleq \left\{ \mathbf{P} \in \mathcal{O}_{b,o,a}^+ : \quad \forall i,\, P_{ijk} 1_o^j 1_a^k = x_i^b, \quad \forall j,\, P_{ijk} 1_b^i 1_a^k = x_j^o, \quad \forall k,\, P_{ijk} 1_b^i 1_o^j = x_k^a \right\}. \tag{36b}$$

The 3–tensor $\boldsymbol{\nu}$ is chosen to be $\nu_{ijk} = x_i^b x_j^o / (m N_a)$, which is uniform in $k$ and for which $\mathfrak{m}(\boldsymbol{\nu}) = m$. The resulting dual problem is

$$\mathcal{J}^* = \min_{\mathbf{f}_b \in \mathbb{R}^{\mathfrak{N}_b} \, \mathbf{f}_o \in \mathbb{R}^{\mathfrak{N}_o}} J^*(\mathbf{f}_b, \mathbf{f}_o), \tag{37a}$$

where the associated Lagrangian is

$$J^*(\mathbf{f}_b, \mathbf{f}_o) = \varepsilon \left( m \ln \frac{Z_\varepsilon}{m} + m - \mathfrak{m}(\nu) \right) - \mathbf{f}_b^\mathsf{T} \mathbf{x}^b - \mathbf{f}_o^\mathsf{T} \mathbf{x}^o, \tag{37b}$$




with the partition function

$$Z_\varepsilon = \sum_{ijk} \nu_{ijk} e^{\left(f_b^i + f_o^l H_l^j - C_{ba}^{ik} - C_{oa}^{jk}\right)/\varepsilon}. \tag{37c}$$

This partition function is elegant but impractical since in high dimension a 3–tensor might be too large to store and compute with. However the partition function Eq. (37c) can be simplified by noticing that

$$Z_\varepsilon = \sum_{ij} \nu_{ij} e^{\left(f_b^i + f_o^l H_l^j - C_{bo}^{ij}\right)/\varepsilon}, \tag{38}$$

where we introduced the effective cost metric

$$[\mathbf{C}_{bo}]_{ij} = -\varepsilon \ln\left(\sum_k \frac{\nu_{ijk}}{\nu_{ij}} e^{-\left(C_{ba}^{ik} + C_{oa}^{jk}\right)/\varepsilon}\right), \tag{39}$$

which is the regularised cost – known in statistics and machine learning as a *soft-plus* transform – of Eq. (23c). The 2–tensor $\nu_{ij}$ plays the same role as that of the first step of the algorithm; we choose it as $\nu_{ij} = x_i^b x_j^o / m$, for which $\mathfrak{m}(\boldsymbol{\nu}) = m$. The dual problem now only involves 2–tensors and becomes numerically more efficient. Given the optimal $\mathbf{f}_b$ and $\mathbf{f}_o$, the (glued) optimal transference plan $\mathbf{P}^{boa}$ is formally given by

$$P_{ijk}^{boa} = \frac{\nu_{ijk}}{Z_\varepsilon} e^{\left(f_b^i + f_o^l H_l^j - C_{ba}^{ik} - C_{oa}^{jk}\right)/\varepsilon}. \tag{40}$$

The W-barycentre $\mathbf{x}^a$ is then given as a marginal of $\mathbf{P}^{boa}$:

$$x_k^a = P_{ijk} 1_b^i 1_o^j = \frac{1}{Z_\varepsilon} \sum_{ij} \nu_{ijk} e^{\left(f_b^i + f_o^l H_l^j - C_{ba}^{ik} - C_{oa}^{jk}\right)/\varepsilon}. \tag{41}$$

Because of the normalisation of the transference plan to $m$, the entropic regularisation exhibits a $\varepsilon m \ln Z_\varepsilon$ instead of $\varepsilon Z_\varepsilon$. This systematically enforces normalisation in the computations of the gradients, as well as in the course of the numerical optimisation of the dual cost function, *de facto* working in log-domain. We experienced more stable computations and the ability to reach smaller $\varepsilon$, as compared to the case without normalisation. This completes the solution through the 2-step OTDA algorithm.

### 2.4.4 Classical data assimilation as a particular case

The primal problem Eq. (17) of classical DA reads

$$\mathcal{L}_{cl} = \min_{\mathbf{x}^a \in \mathcal{O}_{N_a}^+} \left\{ \zeta_b(\mathbf{y}^b - \mathbf{x}^a) + \zeta_o(\mathbf{y}^o - \mathbf{H}\mathbf{x}^a) \right\}. \tag{42}$$

Let us see how the OTDA formalism Eq. (23) can account for classical DA. In the context of classical DA, the observable spaces for $\mathbf{x}^b$, $\mathbf{x}^o$, and $\mathbf{x}^a$ are assumed to coincide. Let us then define the cost matrices

$$[\mathbf{C}_{ba}^\infty]_{ij} \triangleq [\mathbf{C}_{oa}^\infty]_{ij} \triangleq \begin{cases} 0 & \text{if } i = j \\ +\infty & \text{if } i \neq j \end{cases}, \tag{43}$$




i.e. it is assumed that the cost of moving masses is as large as can be. Looking back at Eq. (19) but with these costs, it is clear
that in order to avoid the primal cost function to be $+\infty$, the transference plan $P_{ijk}$ must always be 0 unless $i = j = k$. But this
implies from the definition of $\mathcal{U}_{\mathrm{boa}}$ that the observables coincide: $\mathbf{x}^{\mathrm{b}} = \mathbf{x}^{\mathrm{o}} = \mathbf{x}^{\mathrm{a}}$ and that their mass is given by $\mathfrak{m}(\mathbf{P})$. Hence,
the OTDA primal problem is mathematically equivalent in this limit to the classical DA primal problem. And classical DA can
be seen as a limit case of the hybrid OTDA problem with the specific cost matrices $\mathbf{C}_{\mathrm{ba}}^{\infty}$ and $\mathbf{C}_{\mathrm{oa}}^{\infty}$. Note that from its definition
Eq. (23c), the effective cost $\mathbf{C}_{\mathrm{bo}}$ obtained from $\mathbf{C}_{\mathrm{ba}}^{\infty}$ and $\mathbf{C}_{\mathrm{oa}}^{\infty}$ coincide with $\mathbf{C}_{\mathrm{bo}}^{\infty} \overset{\Delta}{=} \mathbf{C}_{\mathrm{ba}}^{\infty} = \mathbf{C}_{\mathrm{oa}}^{\infty}$.

## 425 2.5 A direct solution

The two-step approach of Sect. 2.4 has the merit to connect to the traditional W-barycentre problem, by first estimating $\mathbf{x}^{\mathrm{b}}$ and
$\mathbf{x}^{\mathrm{o}}$, and later computing the W-barycentre in between both states. It also suggests the existence of the effective cost metric of
the problem. However, going through its consecutive steps may not be necessary for pure computational purposes. Here we
describe a direct approach that yields the analysis of the OTDA problem. It is less enlightening but is practical and will be used
in the subsequent illustrations of the present paper.

An alternative formulation to the primal problem Eq. (19) relies on two transference plans $\mathbf{P}^{\mathrm{ba}}$ and $\mathbf{P}^{\mathrm{oa}}$ corresponding to
the two transports of the underlying W-barycentre problem, instead of the gluing one. Moreover, entropic regularisation is
enforced via $\mathcal{K}(\mathbf{P}^{\mathrm{ba}}|\boldsymbol{\nu}^{\mathrm{ba}})$ and $\mathcal{K}(\mathbf{P}^{\mathrm{oa}}|\boldsymbol{\nu}^{\mathrm{oa}})$. The corresponding optimisation problem reads

$$\mathcal{L} = \min_{\mathbf{x}^{\mathrm{b}} \in \mathcal{O}_{\mathrm{b}}^{+}\, \mathbf{x}^{\mathrm{o}} \in \mathcal{O}_{\mathrm{o}}^{+}\, \mathbf{x}^{\mathrm{a}} \in \mathcal{O}_{\mathrm{a}}^{+}} \left[ \zeta_{\mathrm{b}}(\mathbf{y}^{\mathrm{b}} - \mathbf{x}^{\mathrm{b}}) + \zeta_{\mathrm{o}}(\mathbf{y}^{\mathrm{o}} - \mathbf{H}\mathbf{x}^{\mathrm{o}}) \right.$$

$$\left. + \min_{\mathbf{P}^{\mathrm{ba}} \in \mathcal{U}_{\mathrm{ba}}\, \mathbf{P}^{\mathrm{oa}} \in \mathcal{U}_{\mathrm{oa}}} \left\{ \varepsilon \mathcal{K}(\mathbf{P}^{\mathrm{ba}}|\boldsymbol{\nu}^{\mathrm{ba}}) + \varepsilon \mathcal{K}(\mathbf{P}^{\mathrm{oa}}|\boldsymbol{\nu}^{\mathrm{oa}}) + P_{ik}^{\mathrm{ba}} C_{\mathrm{ba}}^{ik} + P_{jk}^{\mathrm{oa}} C_{\mathrm{oa}}^{jk} \right\} \right], \tag{44a}$$

where the admissible sets of transference plans $\mathbf{P}^{\mathrm{ba}}$ and $\mathbf{P}^{\mathrm{oa}}$ are defined by

$$\mathcal{U}_{\mathrm{ba}} \overset{\Delta}{=} \left\{ \mathbf{P} \in \mathcal{O}_{\mathrm{b,a}}^{+} : \quad \mathbf{P}\mathbf{1}_{\mathrm{a}} = \mathbf{x}^{\mathrm{b}}, \quad \mathbf{P}^{\mathsf{T}}\mathbf{1}_{\mathrm{b}} = \mathbf{x}^{\mathrm{a}} \right\}, \tag{44b}$$

$$\mathcal{U}_{\mathrm{oa}} \overset{\Delta}{=} \left\{ \mathbf{P} \in \mathcal{O}_{\mathrm{o,a}}^{+} : \quad \mathbf{P}\mathbf{1}_{\mathrm{a}} = \mathbf{x}^{\mathrm{o}}, \quad \mathbf{P}^{\mathsf{T}}\mathbf{1}_{\mathrm{b}} = \mathbf{x}^{\mathrm{a}} \right\}. \tag{44c}$$

Following the same type of derivation as reported in the previous sections, the corresponding dual problem to be minimised is
obtained as

$$\mathcal{J}_{\varepsilon}^{*} = \min_{\mathbf{f}_{\mathrm{b}} \in \mathbb{R}^{N_{\mathrm{b}}}\, \mathbf{f}_{\mathrm{o}} \in \mathbb{R}^{N_{\mathrm{o}}}\, \mathbf{f}_{\mathrm{a}} \in \mathbb{R}^{N_{\mathrm{a}}}} J_{\varepsilon}^{*}(\mathbf{f}_{\mathrm{b}}, \mathbf{f}_{\mathrm{o}}, \mathbf{f}_{\mathrm{a}}), \tag{45a}$$

where, discarding the constant $-\varepsilon \mathfrak{m}(\boldsymbol{\nu}^{\mathrm{ba}}) - \varepsilon \mathfrak{m}(\boldsymbol{\nu}^{\mathrm{oa}})$, the associated regularised Lagrangian is

$$J_{\varepsilon}^{*}(\mathbf{f}_{\mathrm{b}}, \mathbf{f}_{\mathrm{o}}, \mathbf{f}_{\mathrm{a}}) = \varepsilon Z_{\varepsilon}^{\mathrm{ba}}(\mathbf{f}_{\mathrm{b}}, \mathbf{f}_{\mathrm{a}}) + \varepsilon Z_{\varepsilon}^{\mathrm{oa}}(\mathbf{f}_{\mathrm{o}}, \mathbf{f}_{\mathrm{a}}) + \zeta_{\mathrm{b}}^{*}(\mathbf{f}_{\mathrm{b}}) + \zeta_{\mathrm{o}}^{*}(\mathbf{f}_{\mathrm{o}}) - \mathbf{f}_{\mathrm{b}}^{\mathsf{T}}\mathbf{y}^{\mathrm{b}} - \mathbf{f}_{\mathrm{o}}^{\mathsf{T}}\mathbf{y}^{\mathrm{o}}, \tag{45b}$$

with a partition function associated to each transport:

$$Z_{\varepsilon}^{\mathrm{ba}} \overset{\Delta}{=} \sum_{ik} P_{ik}^{\mathrm{ba}}, \qquad Z_{\varepsilon}^{\mathrm{oa}} \overset{\Delta}{=} \sum_{jk} P_{jk}^{\mathrm{oa}} \tag{45c}$$





where

$$P_{ik}^{\mathrm{ba}} = \nu_{ik}^{\mathrm{ba}} e^{\left(f_{\mathrm{b}}^i + f_{\mathrm{a}}^k - C_{\mathrm{ba}}^{ik}\right)/\varepsilon}, \qquad P_{jk}^{\mathrm{oa}} = \nu_{jk}^{\mathrm{oa}} e^{\left(f_{\mathrm{o}}^l H_l^j - f_{\mathrm{a}}^k - C_{\mathrm{oa}}^{jk}\right)/\varepsilon}. \tag{45d}$$

It turns out that the optimal $\mathbf{f}_{\mathrm{a}}$ can be obtained analytically as a function of $\mathbf{f}_{\mathrm{b}}$ and $\mathbf{f}_{\mathrm{o}}$, which we checked makes the optimisation numerically more efficient and robust. Indeed, let us introduce $\psi_k \triangleq e^{f_{\mathrm{a}}^k/\varepsilon}$. We could optimise $J_\varepsilon^*(\mathbf{f}_{\mathrm{b}}, \mathbf{f}_{\mathrm{o}}, \mathbf{f}_{\mathrm{a}} = \varepsilon \ln \boldsymbol{\psi})$ on $\boldsymbol{\psi}$:

$$0 = \partial_{\psi_k} J_\varepsilon^*(\mathbf{f}_{\mathrm{b}}, \mathbf{f}_{\mathrm{o}}, \mathbf{f}_{\mathrm{a}}) = \sum_i \nu_{ik}^{\mathrm{ba}} e^{\left(f_{\mathrm{b}}^i - C_{\mathrm{ba}}^{ik}\right)/\varepsilon} - \frac{1}{\psi_k^2} \sum_j \nu_{jk}^{\mathrm{oa}} e^{\left(f_{\mathrm{o}}^l H_l^j - C_{\mathrm{oa}}^{jk}\right)/\varepsilon}, \tag{46}$$

yielding the solution

$$\psi_k^2 = \frac{Z_{\varepsilon,k}^{\mathrm{oa}}}{Z_{\varepsilon,k}^{\mathrm{ba}}}, \qquad Z_{\varepsilon,k}^{\mathrm{oa}} \triangleq \sum_j \nu_{jk}^{\mathrm{oa}} e^{\left(f_{\mathrm{o}}^l H_l^j - C_{\mathrm{oa}}^{jk}\right)/\varepsilon}, \qquad Z_{\varepsilon,k}^{\mathrm{ba}} \triangleq \sum_i \nu_{ik}^{\mathrm{ba}} e^{\left(f_{\mathrm{b}}^i - C_{\mathrm{ba}}^{ik}\right)/\varepsilon}. \tag{47}$$

Up to irrelevant constants, the resulting effective cost function using the optimal $\psi_k$ is

$$J_\varepsilon^*(\mathbf{f}_{\mathrm{b}}, \mathbf{f}_{\mathrm{o}}) = 2\varepsilon \sum_k \sqrt{Z_{\varepsilon,k}^{\mathrm{ba}} Z_{\varepsilon,k}^{\mathrm{oa}}} + \zeta_{\mathrm{b}}^*(\mathbf{f}_{\mathrm{b}}) + \zeta_{\mathrm{o}}^*(\mathbf{f}_{\mathrm{o}}) - \mathbf{f}_{\mathrm{b}}^{\mathsf{T}} \mathbf{y}^{\mathrm{b}} - \mathbf{f}_{\mathrm{o}}^{\mathsf{T}} \mathbf{y}^{\mathrm{o}}. \tag{48}$$

Now, the optimal W-barycentre $\mathbf{x}^{\mathrm{a}}$ is given by either $x_k^{\mathrm{a}} = P_{ik}^{\mathrm{ba}} 1_{\mathrm{b}}^i$ or $x_k^{\mathrm{a}} = P_{jk}^{\mathrm{oa}} 1_{\mathrm{o}}^j$, i.e.

$$x_k^{\mathrm{a}} = \psi_k Z_{\varepsilon,k}^{\mathrm{ba}} = \frac{1}{\psi_k} Z_{\varepsilon,k}^{\mathrm{oa}}, \tag{49}$$

from which we can infer the $\psi_k$-free expression

$$x_k^{\mathrm{a}} = \sqrt{Z_{\varepsilon,k}^{\mathrm{ba}} Z_{\varepsilon,k}^{\mathrm{oa}}}. \tag{50}$$

It is also useful to retrieve the optimal value of $\mathbf{f}_{\mathrm{a}}$ and obtain

$$f_{\mathrm{a}}^k = \varepsilon \ln \psi_k = \frac{\varepsilon}{2} \ln\left(\frac{Z_{\varepsilon,k}^{\mathrm{oa}}}{Z_{\varepsilon,k}^{\mathrm{ba}}}\right), \tag{51}$$

so that we can compute the other two analysed observables, $\mathbf{x}^{\mathrm{b}}$ and $\mathbf{x}^{\mathrm{o}}$, using

$$x_i^{\mathrm{b}} = P_{ik}^{\mathrm{ba}} 1_{\mathrm{a}}^k = \sum_k \psi_k \nu_{ik}^{\mathrm{ba}} e^{\left(f_{\mathrm{b}}^i - C_{\mathrm{ba}}^{ik}\right)/\varepsilon} = e^{f_{\mathrm{b}}^i/\varepsilon} \sum_k \nu_{ik}^{\mathrm{ba}} e^{\left(f_{\mathrm{a}}^k - C_{\mathrm{ba}}^{ik}\right)/\varepsilon}, \tag{52a}$$

$$x_j^{\mathrm{o}} = P_{jk}^{\mathrm{oa}} 1_{\mathrm{a}}^k = \sum_k \frac{1}{\psi_k} \nu_{jk}^{\mathrm{oa}} e^{\left(f_{\mathrm{o}}^l H_l^j - C_{\mathrm{oa}}^{jk}\right)/\varepsilon} = e^{\left(f_{\mathrm{o}}^l H_l^j\right)/\varepsilon} \sum_k \nu_{jk}^{\mathrm{oa}} e^{\left(-f_{\mathrm{a}}^k - C_{\mathrm{oa}}^{jk}\right)/\varepsilon}. \tag{52b}$$

Note that most of these expressions can be assessed in a robust way in the log-domain. For instance we use in practice, 
equivalently to Eqs. (50,52):

$$\varepsilon \ln x_k^{\mathrm{a}} = \frac{\varepsilon}{2} \ln \sum_i \nu_{ik}^{\mathrm{ba}} e^{\left(f_{\mathrm{b}}^i - C_{\mathrm{ba}}^{ik}\right)/\varepsilon} + \frac{\varepsilon}{2} \ln \sum_j \nu_{jk}^{\mathrm{oa}} e^{\left(f_{\mathrm{o}}^l H_l^j - C_{\mathrm{oa}}^{jk}\right)/\varepsilon}, \tag{53a}$$

$$\varepsilon \ln x_i^{\mathrm{b}} = f_{\mathrm{b}}^i + \varepsilon \ln \sum_k \nu_{ik}^{\mathrm{ba}} e^{\left(f_{\mathrm{a}}^k - C_{\mathrm{ba}}^{ik}\right)/\varepsilon}, \tag{53b}$$

$$\varepsilon \ln x_j^{\mathrm{o}} = f_{\mathrm{o}}^l H_l^j + \varepsilon \ln \sum_k \nu_{jk}^{\mathrm{oa}} e^{\left(-f_{\mathrm{a}}^k - C_{\mathrm{oa}}^{jk}\right)/\varepsilon}. \tag{53c}$$





## 3 Numerical illustrations

In this section, we showcase a selection of OTDA 3D-Var analyses. These are meant to stress the versatility of the formalism and the diverse solutions it offers, with significantly more degrees of freedom than in classical DA. The OTDA state analysis is carried out using Sect. 2.5 and its formulas. Unless specifically discussed, entropic regularisation is used with $\varepsilon = 10^{-3}$. The dual cost function Eq. (48) is minimised using the quasi-Newton method L-BFGS-B (Liu and Nocedal, 1989), which yields the optimal $\mathbf{f}_b, \mathbf{f}_o$. Then Eq. (53) is employed to compute $\mathbf{x}^b$, $\mathbf{x}^o$, and $\mathbf{x}^a$.

### 3.1 One-dimensional examples

Considering the case where the physical space of the fields is one-dimensional, we build bell-shaped observations $\mathbf{y}^b$ and $\mathbf{y}^o$, related to an observable space of size $N_b = N_o = N_a = 10^2$ shared by $\mathbf{x}^b$, $\mathbf{x}^o$ and $\mathbf{x}^a$. Since $\mathbf{y}^b$ is a fully observed instance of $\mathbf{x}^b$, we have $\mathfrak{N}_b = N_b = 10^2$, while $\mathfrak{N}_o$ may differ from $N_o$ depending on the definition of the observation operator $\mathbf{H}$. We choose (Gaussian statistics)

$$\zeta_b(\mathbf{e}_b) = \frac{1}{2\sigma_b^2}\|\mathbf{e}_b\|^2, \qquad \zeta_b(\mathbf{e}_o) = \frac{1}{2\sigma_o^2}\|\mathbf{e}_o\|^2, \tag{54}$$

with $\sigma_b = \sigma_o = 10^{-2}$. The states are discretised over the interval $[0,1]$ at sites/grid cells $r_i^\star = (i - \frac{1}{2})/N_\star$ for $i \in [\![1, N_\star]\!]$, with $\star = b, o, a$. Unless otherwise specified, the cost metric has a quadratic dependence with the distance between sites, i.e. $[\mathbf{C}_{ba}]_{ik} = |r_i^b - r_k^a|^2$ and $[\mathbf{C}_{oa}]_{jk} = |r_j^o - r_k^a|^2$. This is our reference setup. The observation operator and the mass of the observations $\mathbf{y}^b$ and $\mathbf{y}^o$ will be described for each experiment.

We consider four experiments where we choose to vary key parameters in the OTDA setup.

### 3.1.1 Varying the imbalance of the observation states

In the first experiment, the system is fully observed with $\mathbf{H} = \mathbf{I}$. We choose $\mathfrak{m}(\mathbf{y}^b) = 1$ and the mass of $\mathbf{y}^o$ to be in the set $\mathfrak{m}(\mathbf{y}^o) \in \{0.5, 1, 1.5\}$, all the other parameters being fixed to the reference. The results are displayed in Fig. 9. Panel (a) corresponds to the case $\mathfrak{m}(\mathbf{y}^o) = 0.5$. The resulting mass of the analysed observables is then $\mathfrak{m}(\mathbf{x}^a) = \mathfrak{m}(\mathbf{x}^b) = \mathfrak{m}(\mathbf{x}^o) = 0.79$. 490 The adjustment of $\mathbf{x}^b$ compared to $\mathbf{y}^b$, and the adjustment of $\mathbf{x}^o$ compared to $\mathbf{y}^o$, which are required to balance $\mathbf{x}^b$, $\mathbf{x}^o$ are patent. Panel (b) corresponds to the case $\mathfrak{m}(\mathbf{y}^o) = 1$. The resulting mass of the analysed observables is then $\mathfrak{m}(\mathbf{x}^a) = \mathfrak{m}(\mathbf{x}^b) = \mathfrak{m}(\mathbf{x}^o) = 1$. No adjustment is required here since $\mathfrak{m}(\mathbf{y}^o) = \mathfrak{m}(\mathbf{y}^b)$, and $\mathbf{x}^o$ and $\mathbf{y}^o$, as well as $\mathbf{x}^b$ and $\mathbf{y}^b$ coincide. Finally, the mass of $\mathbf{y}^o$ is set to $\mathfrak{m}(\mathbf{y}^o) = 1.5$ in panel (c). The resulting mass of the analysed observables is then $\mathfrak{m}(\mathbf{x}^a) = \mathfrak{m}(\mathbf{x}^b) = \mathfrak{m}(\mathbf{x}^o) = 1.34$. The adjustment of $\mathbf{x}^b$ compared to $\mathbf{y}^b$, and the adjustment of $\mathbf{x}^o$ compared to $\mathbf{y}^o$, which are required to balance 495 $\mathbf{x}^b$, $\mathbf{x}^o$ are visually obvious, but the balancing goes in the opposite direction compared to panel (a), as expected.





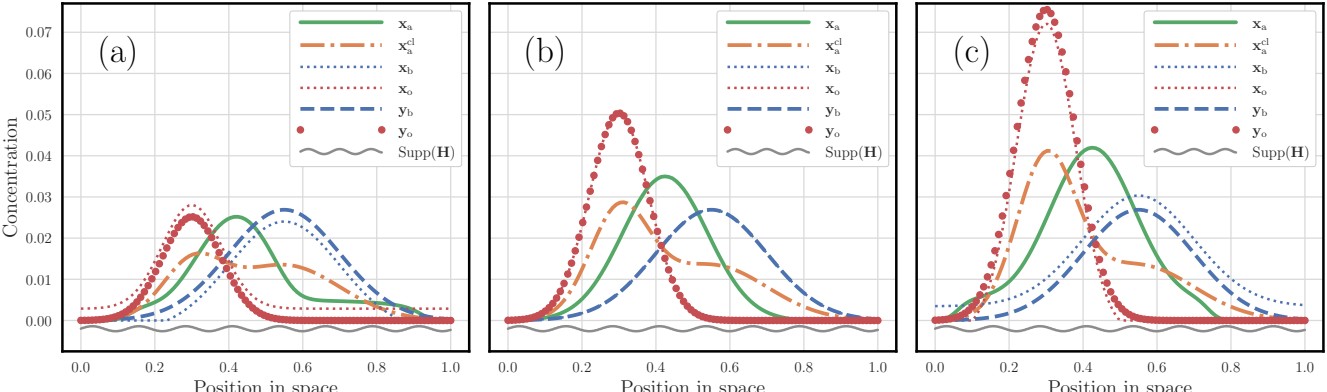

**Figure 9.**

A hybrid OTDA 3D-Var analysis with one-dimensional physical states, where only the mass of $\mathbf{y}^\circ$ is varied. Its mass is $\mathfrak{m}(\mathbf{y}^\circ) = 0.5$ in panel (a), $\mathfrak{m}(\mathbf{y}^\circ) = 1$ in panel (b), and $\mathfrak{m}(\mathbf{y}^\circ) = 1.5$ in panel (c). The dashed blue curve corresponds to the first guess $\mathbf{y}^\mathrm{b}$, the red dots correspond to the observations $\mathbf{y}^\circ$, the analysis state $\mathbf{x}^\mathrm{a}$ is the solid green curve, the analysed observables $\mathbf{x}^\mathrm{b}$ and $\mathbf{x}^\circ$ are blue and red dotted curves, respectively. The support of the observation is underlined by a wavy grey segment. The corresponding classical analysis is also plotted with a dashed-dotted orange curve. The x-axis corresponds to the position in space; the y-axis corresponds to the concentration value of the fields.

### 3.1.2 Varying the sparseness of the observation operator

In this second experiment, all the other parameters being fixed to their reference value, only a fraction of the domain is observed, over $\left[0, \frac{1}{4}\right]$, $\left[0, \frac{1}{2}\right]$ and $\left[0, \frac{3}{4}\right]$, where $\mathbf{H} \in \mathcal{O}^+_{\mathfrak{N}_\mathrm{o} \times N_\mathrm{o}}$ with $\mathfrak{N}_\mathrm{o} = N_\mathrm{o}/4, N_\mathrm{o}/2, 3N_\mathrm{o}/4$, and $H^j_l = \delta_{l,j}$ for $l \in [\![1, \mathfrak{N}_\mathrm{o}]\!]$ and $j \in [\![1, N_\mathrm{o}]\!]$.

The masses of the states that are built to generate $\mathbf{y}^\mathrm{b}$ and $\mathbf{y}^\circ$, before applying any observation operator, are set to 1 and 1.5, respectively. As a result, we have $\mathfrak{m}(\mathbf{y}^\mathrm{b}) = 1$ but $\mathfrak{m}(\mathbf{y}^\circ)$ may depart from 1.5 depending on $\mathbf{H}$. The fully observed case corresponds to panel (c) of Fig. 9. The results are displayed in Fig. 10. It shows how smooth the OTDA solution can be compared to that of classical DA. Yet, as in panel (a), OTDA can also handle obviously diverging sources of information as in the case where the support of $\mathbf{H}$ is $\left[0, \frac{1}{4}\right]$ and where $\mathbf{y}^\circ$ and $\mathbf{y}^\mathrm{b}$ can be seen are barely consistent. In that case, the OTDA
solution is smooth but bimodal.

### 3.1.3 Changing the nature of the cost metric

In this third experiment, we choose the cost metric to be of the form $[\mathbf{C}_\mathrm{ba}]_{ik} = |r^\mathrm{b}_i - r^\mathrm{a}_k|^\alpha$ and $[\mathbf{C}_\mathrm{oa}]_{jk} = |r^\circ_j - r^\mathrm{a}_k|^\alpha$. Only half of the domain is observed over $\left[0, \frac{1}{2}\right]$, as in the case of Fig. 10, panel (b). Since the mass of the state used to produce $\mathbf{y}^\circ$ is 1.5, we have a slightly different $\mathfrak{m}(\mathbf{y}^\circ) = 1.49$. All of the other parameters follow the reference setup. The results are displayed in
Fig. 11. For panel (a), $\alpha$ is set to 0.5. For panel (b), $\alpha$ is set to 1. For panel (c), the cost metric is piecewise; it is quadratic, i.e.



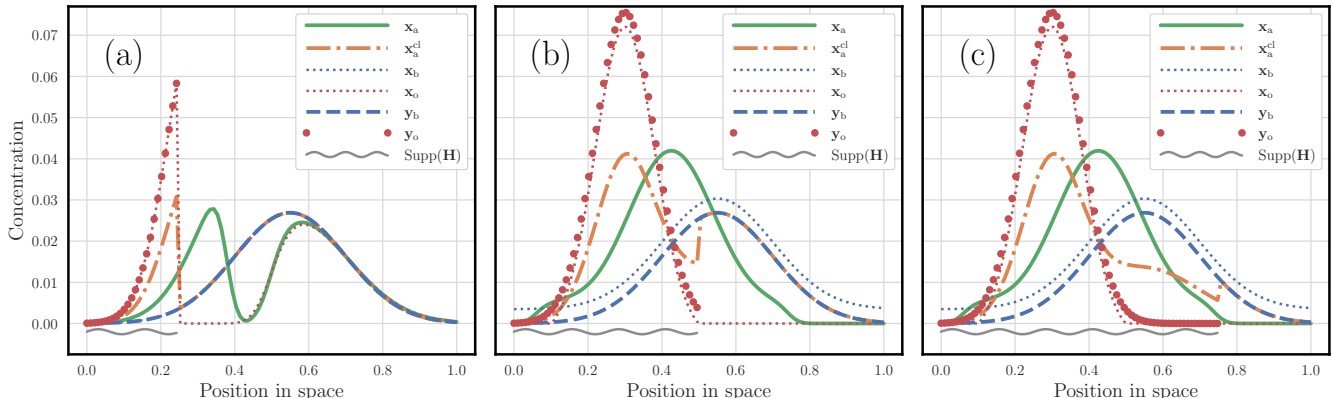

**Figure 10.**

A hybrid OTDA 3D-Var analysis with one-dimensional physical states, where the observation operator is increasingly sparse. The support of $\mathbf{H}$ is $\left[0, \frac{1}{4}\right]$ for panel (a), $\left[0, \frac{1}{2}\right]$ for panel (b), and $\left[0, \frac{3}{4}\right]$ for panel (c). See Fig. 9 for the description of the legend.

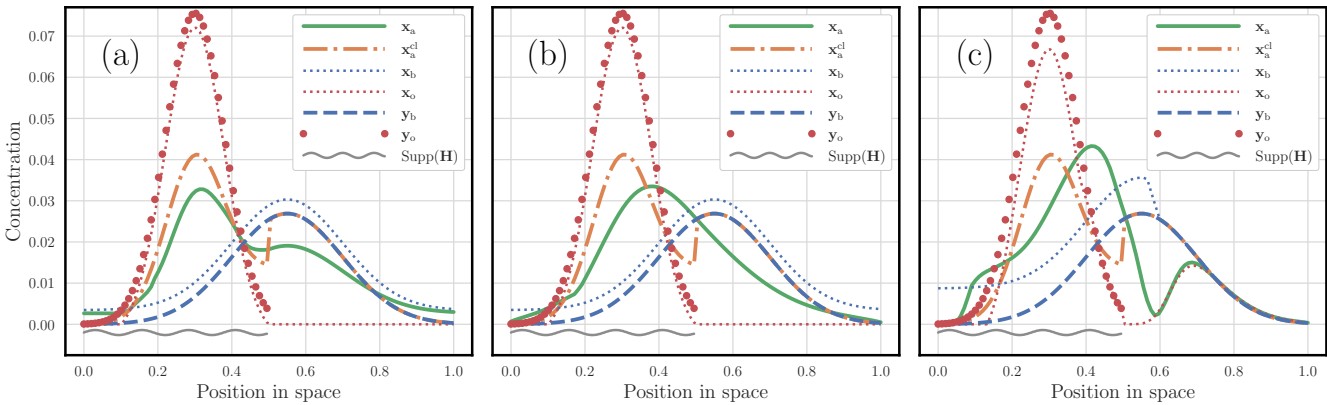

**Figure 11.**

A hybrid OTDA 3D-Var analysis with one-dimensional physical states, where the cost metrics are changed. See the bulk of the text for a definition of those three cost metrics. See Fig. 9 for the description of the legend.

$\alpha = 2$, for pairs of sites separated by less than $10^{-1}$, i.e. $|r_i^{\mathrm{b}} - r_k^{\mathrm{a}}| = |r_j^{\mathrm{o}} - r_k^{\mathrm{a}}| \leq 10^{-1}$, whereas for pairs of sites beyond this range, the costs are chosen to be infinite. Hence transport is prohibited beyond a distance of $10^{-1}$. The case of a pure quadratic cost corresponds to panel (b) of Fig. 10. The impact on the shape of the OTDA analysis is very significant, and suggests that one could easily tailor their own cost to suit their specific DA problem.





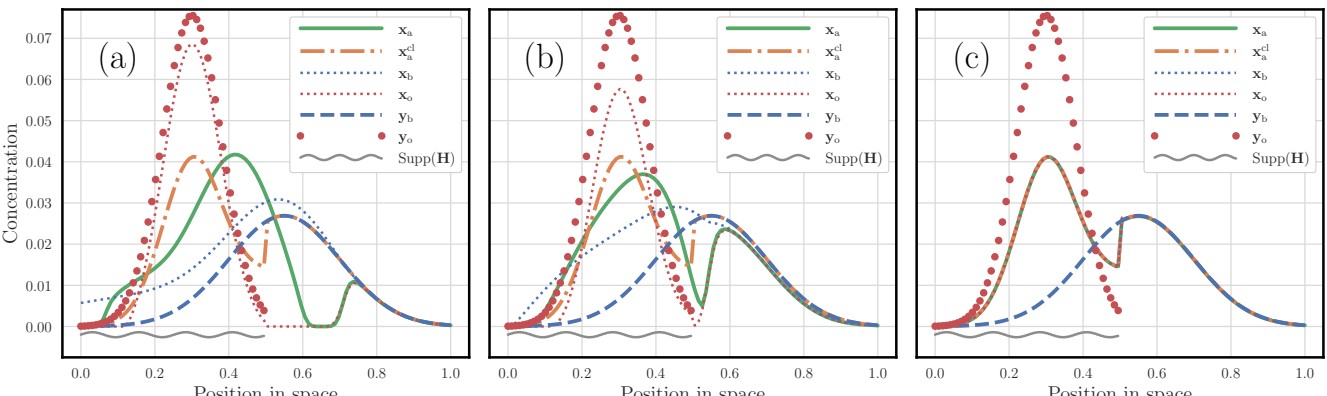

**Figure 12.** Scaling up the cost metrics $\lambda\mathbf{C}_{\mathrm{ba}}$ and $\lambda\mathbf{C}_{\mathrm{oa}}$ with increasing $\lambda$, the OTDA analysis converges to the classical DA analysis. Panels (a), (b), and (c) correspond to the scaling values $\lambda = 10^3, 10^4, 10^6$, respectively. See Fig. 9 for the description of the legend.

### 3.1.4   Classical data assimilation as a subcase of the hybrid optimal transport data assimilation

In the fourth experiment, we would like to numerically check the theoretical prediction of Sect. 2.4.4. Consider again the reference configuration. But only half of the domain, over $\left[0, \frac{1}{2}\right]$, is observed, $\mathbf{H} \in \mathcal{O}^+_{\mathfrak{N}_{\mathrm{o}} \times N_{\mathrm{o}}}$ with $\mathfrak{N}_{\mathrm{o}} = N_{\mathrm{o}}/2$ and $H^j_l = \delta_{l,j}$ for $l \in [\![1, \mathfrak{N}_{\mathrm{o}}]\!]$ and $j \in [\![1, N_{\mathrm{o}}]\!]$. Most importantly, the cost metric has a quadratic dependence with the distance between sites, i.e. $[\mathbf{C}_{\mathrm{ba}}]_{ik} = \lambda|r^{\mathrm{b}}_i - r^{\mathrm{a}}_k|^2$ and $[\mathbf{C}_{\mathrm{oa}}]_{jk} = \lambda|r^{\mathrm{o}}_j - r^{\mathrm{a}}_k|^2$. The case $\lambda = 1$ corresponds to panel (b) of Fig. 10. Figure 12 shows the results corresponding to: $\lambda = 10^3$ for panel (a), $\lambda = 10^4$ for panel (b), and $\lambda = 10^6$ for panel (c). When $\lambda$ is increased, the OTDA analysis should tend to the classical DA solution. This is indeed corroborated by Fig. 12 and comforts the claim of Sect. 2.4.4. Note that, as opposed to the three earlier experiments, we had here to tune $\varepsilon$ since the wide range of $\lambda$ has a significant impact in the balance of the key terms of the cost function (transport cost, discrepancy errors, and regularisation).

### 3.2   Two-dimensional examples

Considering the case where the physical space of the fields is two-dimensional, we perform a couple of 3D-Var analysis on concentration fields (puffs of a pollutant). The states are discretised in the domain $[0,1]^2$ at sites/grid cells $r^\star_{i,j} = \left((i - \frac{1}{2})/N^x_\star, (j - \frac{1}{2})/N^y_\star\right)$ for $(i,j) \in [\![1, N^x_\star]\!] \times [\![1, N^y_\star]\!]$, with $\star = \mathrm{b}, \mathrm{o}, \mathrm{a}$. We choose $N^x_{\mathrm{b}} = N^y_{\mathrm{b}} = N^x_{\mathrm{o}} = N^y_{\mathrm{o}} = N^x_{\mathrm{a}} = N^y_{\mathrm{a}} = 10^2$, such that $N_{\mathrm{b}} = N_{\mathrm{o}} = N_{\mathrm{a}} = 10^4$. The observation vectors are $\mathbf{y}^{\mathrm{b}}$ and $\mathbf{y}^{\mathrm{o}}$. Since $\mathbf{y}^{\mathrm{b}}$ is a fully observed instance of $\mathbf{x}^{\mathrm{b}}$, we have $\mathfrak{N}_{\mathrm{b}} = N_{\mathrm{b}}$, while $\mathfrak{N}_{\mathrm{o}}$ may differ from $N_{\mathrm{o}}$ depending on the definition of the observation operator $\mathbf{H}$. Moreover, we choose (Gaussian statistics)

$$\zeta_{\mathrm{b}}(\mathbf{e}_{\mathrm{b}}) = \frac{1}{2\sigma^2_{\mathrm{b}}}\|\mathbf{e}_{\mathrm{b}}\|^2, \qquad \zeta_{\mathrm{b}}(\mathbf{e}_{\mathrm{o}}) = \frac{1}{2\sigma^2_{\mathrm{o}}}\|\mathbf{e}_{\mathrm{o}}\|^2, \tag{55}$$

with $\sigma_{\mathrm{b}} = \sigma_{\mathrm{o}} = 10^{-2}$. The entropic regularisation parameter is set to $\varepsilon = 10^{-3}$.





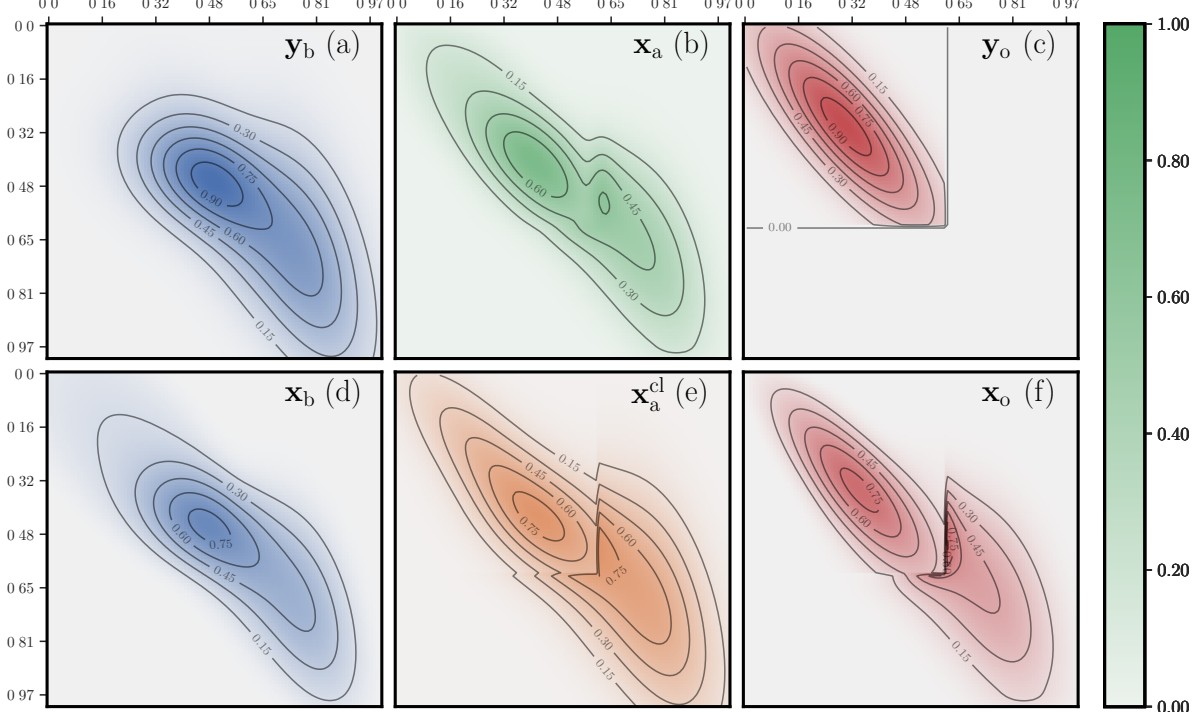

**Figure 13.** Two-dimensions concentration maps (plumes) of a hybrid OTDA analysis for the first configuration. The observations $\mathbf{y}^{\mathrm{b}}$ and $\mathbf{y}^{\mathrm{o}}$, the analysed observables $\mathbf{x}^{\mathrm{b}}$, $\mathbf{x}^{\mathrm{a}}$ i.e. the state analysis, $\mathbf{x}^{\mathrm{o}}$, and the corresponding classical DA analysis $\mathbf{x}^{\mathrm{a}}_{\mathrm{cl}}$ are displayed. All fields are rescaled so their joint maximum is 1. All heatmaps use the same scale. The colour bar represents a unified contrast scale for the diverse field concentrations.

The first analysis is displayed in Fig. 13. The observation operator $\mathbf{H}$ is the identity but its support is restricted to the subdomain $[0, 0.6]^2$. The plumes of pollutants $\mathbf{y}^{\mathrm{b}}$ and $\mathbf{y}^{\mathrm{o}}$ are generated from states formed as combinations of bell-like puffs.
The system is unbalanced with $\mathfrak{m}(\mathbf{y}^{\mathrm{b}}) = 1.35$, $\mathfrak{m}(\mathbf{y}^{\mathrm{o}}) = 0.73$. The cost metric has a quadratic dependence with the distance between sites, i.e. $[\mathbf{C}_{\mathrm{ba}}]_{ik} = \left\| \mathbf{r}^{\mathrm{b}}_i - \mathbf{r}^{\mathrm{a}}_k \right\|^2_2$ and $[\mathbf{C}_{\mathrm{oa}}]_{jk} = \left\| \mathbf{r}^{\mathrm{o}}_j - \mathbf{r}^{\mathrm{a}}_k \right\|^2_2$. The OTDA analysis is clearly smoother than the classical solution. The classical solution does not cope very well with the seemingly disagreeing sources of information $\mathbf{y}^{\mathrm{b}}$ and $\mathbf{y}^{\mathrm{o}}$, which generates sharp transitions in the classical analysis. If $\mathbf{y}^{\mathrm{b}}$ and $\mathbf{y}^{\mathrm{o}}$ were consistently obtained from a truth perturbed with errors with short-range correlation, then the classical analysis would be as good as can be, while the OTDA solution may be
too safe. However, if one believes that structural errors and in particular location errors can impact $\mathbf{y}^{\mathrm{b}}$ and $\mathbf{y}^{\mathrm{o}}$, then the classical solution is improper and the OTDA analysis preferable.

The second analysis is displayed in Fig. 14. The support of the observation operator $\mathbf{H}$ is again contained within the subdomain $[0, 0.6]^2$ but only one of four grid cells are actually observed in this area. The observation states $\mathbf{y}^{\mathrm{b}}$ and $\mathbf{y}^{\mathrm{o}}$ are generated from the same states as for Fig. 13. The system is unbalanced with $\mathfrak{m}(\mathbf{y}^{\mathrm{b}}) = 1.35$, $\mathfrak{m}(\mathbf{y}^{\mathrm{o}}) = 0.18$. The cost metric is defined



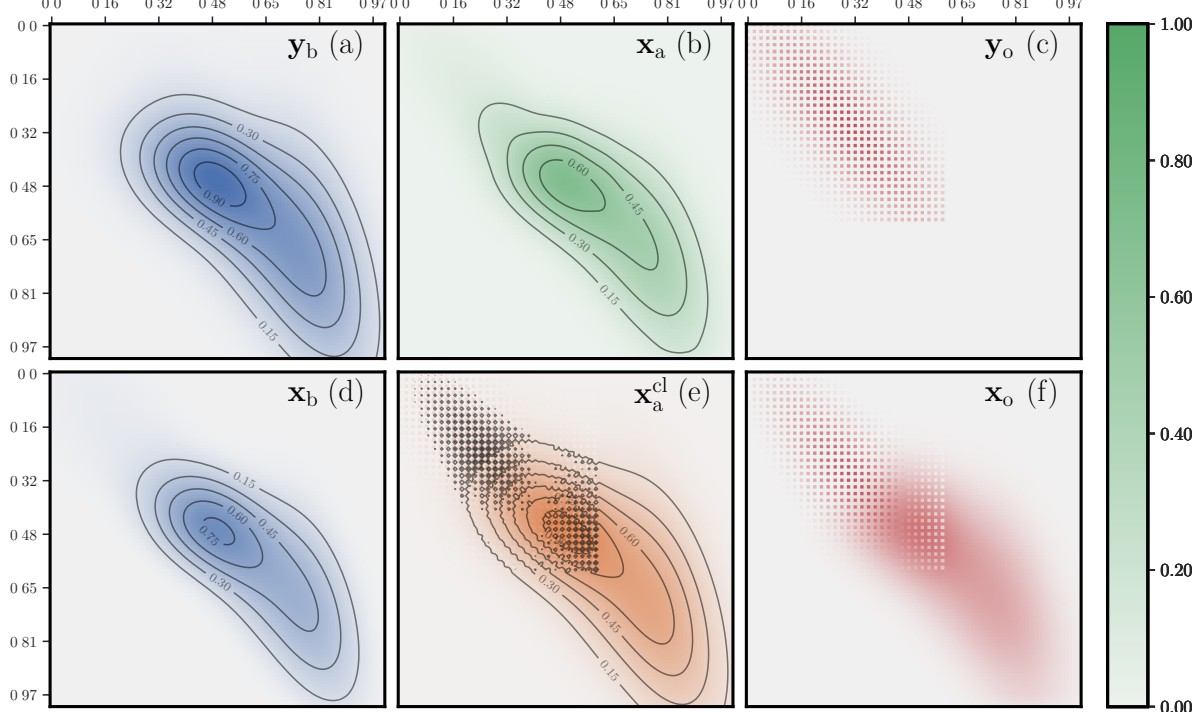

**Figure 14.** Two-dimensions concentrations maps of a hybrid OTDA analysis for the second configuration. The observations $\mathbf{y}^{\mathrm{b}}$ and $\mathbf{y}^{\mathrm{o}}$, the analysed observables $\mathbf{x}^{\mathrm{b}}$, $\mathbf{x}^{\mathrm{a}}$ i.e. the state analysis, $\mathbf{x}^{\mathrm{o}}$, and the corresponding classical DA analysis. Compared to Fig. 13, only $\mathbf{H}$ has changed. The level sets in panels (c) and (f) are omitted since they are driven by the staggered observation operator.

to be the same as in Fig. 13. The OTDA analysis is even smoother in this case as compared to the classical DA analysis. It is much less impacted by the sparseness of the observation operator. The classical solution has to account for the staggered observations in the top left corner of the domain because the first guess in that region is weak. By contrast, the OTDA solution assumes that location errors are possible and it hence moves around the mass corresponding to these observations, so that the structure of the observation operator is not as impactful on the analysis.

**4  Uncertainty quantification**

In this section, we compute the posterior error covariance matrix $\mathbf{P}^{\mathrm{a}}$ associated to the state analysis $\mathbf{x}^{\mathrm{a}}$, in order to complete the OTDA 3D-Var analysis description. There are many ways to proceed depending on the chosen regularisation and on the targeted degree of generality. Here, for the sake of consistency, we report on the way to derive $\mathbf{P}^{\mathrm{a}}$ following the computation of the analysis state $\mathbf{x}^{\mathrm{a}}$ proposed in Sect. 2.5.



## 4.1 Mathematical results

Let us denote the compounded vectors of the observations, of the Lagrange multipliers, and of the observables, as well as the compounded observation operator by

$$
\mathbf{y} \triangleq \begin{bmatrix} \mathbf{y}^{\mathrm{b}} \\ \mathbf{y}^{\mathrm{o}} \end{bmatrix}, \qquad \mathbf{f} \triangleq \begin{bmatrix} \mathbf{f}_{\mathrm{b}} \\ \mathbf{f}_{\mathrm{o}} \end{bmatrix}, \qquad \mathbf{x} \triangleq \begin{bmatrix} \mathbf{x}^{\mathrm{b}} \\ \mathbf{x}^{\mathrm{o}} \end{bmatrix}, \qquad \mathcal{H} \triangleq \begin{bmatrix} \mathbf{I}_{\mathrm{b}} & \mathbf{0} \\ \mathbf{0} & \mathbf{H} \end{bmatrix}, \tag{56}
$$

of size $\mathfrak{N}_{\mathrm{b}} + \mathfrak{N}_{\mathrm{o}}$, $\mathfrak{N}_{\mathrm{b}} + \mathfrak{N}_{\mathrm{o}}$, $N_{\mathrm{b}} + N_{\mathrm{o}}$, and $(\mathfrak{N}_{\mathrm{b}} + \mathfrak{N}_{\mathrm{o}}) \times (N_{\mathrm{b}} + N_{\mathrm{o}})$, respectively. Similarly, we define the sum of the error statistics by $\zeta(\mathbf{f}) \triangleq \zeta_{\mathrm{b}}(\mathbf{f}_{\mathrm{b}}) + \zeta_{\mathrm{o}}(\mathbf{f}_{\mathrm{o}})$, whose Legendre-Fenchel transform is $\zeta^*(\mathbf{f}) = \zeta_{\mathrm{b}}^*(\mathbf{f}_{\mathrm{b}}) + \zeta_{\mathrm{o}}^*(\mathbf{f}_{\mathrm{o}})$. Using this notation, we can recapitulate the key results of Sect. 2.5: the effective dual cost function is

$$
J_{\varepsilon}^*(\mathbf{f}) \triangleq \varepsilon Z_{\varepsilon}(\mathbf{f}) + \zeta^*(\mathbf{f}) - \mathbf{f}^{\mathsf{T}} \mathbf{y}, \qquad Z_{\varepsilon}(\mathbf{f}) \triangleq 2 \sum_{k} \sqrt{Z_{\varepsilon,k}^{\mathrm{ba}}(\mathbf{f}_{\mathrm{b}}) Z_{\varepsilon,k}^{\mathrm{oa}}(\mathbf{f}_{\mathrm{o}})}, \tag{57}
$$

and the analysis state reads

$$
x_k^{\mathrm{a}}(\mathbf{f}) = \sqrt{Z_{\varepsilon,k}^{\mathrm{ba}}(\mathbf{f}_{\mathrm{b}}) Z_{\varepsilon,k}^{\mathrm{oa}}(\mathbf{f}_{\mathrm{o}})}, \tag{58}
$$

where the dependence of the analysis state and the partition functions on $\mathbf{f}$, $\mathbf{f}_{\mathrm{b}}$ and $\mathbf{f}_{\mathrm{o}}$ is now emphasised and made explicit.

Any prior source of error in the system stems from $\mathbf{y}$, and hence drives the posterior error in the analysis $\mathbf{x}^{\mathrm{a}}$. That is why we are interested in the sensitivity of $\mathbf{x}^{\mathrm{a}}$ with respect to $\mathbf{y}$, i.e. $\delta \mathbf{x}^{\mathrm{a}} = \partial_{\mathbf{y}} \mathbf{x}^{\mathrm{a}} \delta \mathbf{y}$. Denoting the expectation operator by $\mathbb{E}$, the error covariance matrix is then defined by

$$
\mathbf{P}^{\mathrm{a}} = \mathbb{E}\left[\delta \mathbf{x}^{\mathrm{a}} (\delta \mathbf{x}^{\mathrm{a}})^{\mathsf{T}}\right] = (\partial_{\mathbf{y}} \mathbf{x}^{\mathrm{a}}) \mathbb{E}\left[\delta \mathbf{y} \delta \mathbf{y}^{\mathsf{T}}\right] (\partial_{\mathbf{y}} \mathbf{x}^{\mathrm{a}})^{\mathsf{T}} = (\partial_{\mathbf{y}} \mathbf{x}^{\mathrm{a}}) (\partial_{\mathbf{f}}^2 \zeta^*) (\partial_{\mathbf{y}} \mathbf{x}^{\mathrm{a}})^{\mathsf{T}}, \tag{59}
$$

from which a matrix factor $\mathbf{X}^{\mathrm{a}}$ of $\mathbf{P}^{\mathrm{a}}$, i.e. which satisfies $\mathbf{P}^{\mathrm{a}} = \mathbf{X}^{\mathrm{a}} (\mathbf{X}^{\mathrm{a}})^{\mathsf{T}}$ and whose expressions are usually much shorter than those of $\mathbf{P}^{\mathrm{a}}$, can be extracted, up to the multiplication by an orthogonal matrix on the right:

$$
\mathbf{X}^{\mathrm{a}} = \partial_{\mathbf{y}} \mathbf{x}^{\mathrm{a}} (\partial_{\mathbf{f}}^2 \zeta^*)^{\frac{1}{2}}. \tag{60}
$$

To compute the sensitivity matrix $\partial_{\mathbf{y}} \mathbf{x}^{\mathrm{a}}$, we leverage the stationarity of the dual cost function at the minimum:

$$
\partial_{\mathbf{f}} J_{\varepsilon}^*(\mathbf{f}(\mathbf{y}), \mathbf{y}) = \mathbf{0}, \tag{61}
$$

and resort to the implicit function theorem:

$$
\mathbf{0} = \mathrm{d}_{\mathbf{y}} \partial_{\mathbf{f}} J_{\varepsilon}^*(\mathbf{f}(\mathbf{y}), \mathbf{y}) = \partial_{\mathbf{f}}^2 J_{\varepsilon}^* \partial_{\mathbf{y}} \mathbf{f} + \partial_{\mathbf{f}} \partial_{\mathbf{y}} J_{\varepsilon}^*, \tag{62}
$$

which yields

$$
\partial_{\mathbf{y}} \mathbf{f} = -\left[\partial_{\mathbf{f}}^2 J_{\varepsilon}^*\right]^{-1} \partial_{\mathbf{f}} \partial_{\mathbf{y}} J_{\varepsilon}^* = \left[\partial_{\mathbf{f}}^2 J_{\varepsilon}^*\right]^{-1}, \tag{63}
$$





since $\partial_{\mathbf{f}}\partial_{\mathbf{y}}J_{\varepsilon}^{*} = -\mathbf{I}_{\mathfrak{b}\mathfrak{o}}$, where $\mathbf{I}_{\mathfrak{b}\mathfrak{o}}$ is the identity matrix in the compounded observation space $\mathbb{R}^{\mathfrak{N}_{\mathfrak{b}}+\mathfrak{N}_{\mathfrak{o}}}$. The sensitivity $\partial_{\mathbf{y}}\mathbf{x}^{\mathrm{a}}$ can now be computed using the Leibniz chain rule and Eq. (63):

$$\frac{\partial \mathbf{x}^{\mathrm{a}}}{\partial \mathbf{y}} = \frac{\partial \mathbf{x}^{\mathrm{a}}}{\partial \mathbf{f}}\frac{\partial \mathbf{f}}{\partial \mathbf{y}} = \partial_{\mathbf{f}}\mathbf{x}^{\mathrm{a}}\left[\partial_{\mathbf{f}}^{2}J_{\varepsilon}^{*}\right]^{-1}. \tag{64}$$

Let us now compute the Jacobian and Hessian in the right-hand side of Eq. (64). To that end and in order to externalise the observation operator, we introduce $\hat{Z}_{\varepsilon}$ and $\hat{\mathbf{x}}_{\mathrm{a}}$ such that

$$\hat{Z}_{\varepsilon}(\boldsymbol{\eta} = \boldsymbol{\mathcal{H}}^{\mathsf{T}}\mathbf{f}) \stackrel{\Delta}{=} Z_{\varepsilon}(\mathbf{f}), \qquad \hat{\mathbf{x}}_{\mathrm{a}}(\boldsymbol{\eta} = \boldsymbol{\mathcal{H}}^{\mathsf{T}}\mathbf{f}) \stackrel{\Delta}{=} \mathbf{x}^{\mathrm{a}}(\mathbf{f}), \tag{65}$$

and the related Jacobian and Hessians

$$\boldsymbol{\Omega}_{\mathrm{bo},\mathrm{a}} \stackrel{\Delta}{=} \partial_{\boldsymbol{\eta}}\hat{\mathbf{x}}_{\mathrm{a}}, \qquad \boldsymbol{\Omega}_{\mathrm{bo},\mathrm{bo}} \stackrel{\Delta}{=} \varepsilon\partial_{\boldsymbol{\eta}}^{2}\hat{Z}_{\varepsilon}, \qquad \boldsymbol{\Lambda}_{\mathrm{bo},\mathrm{bo}} \stackrel{\Delta}{=} \partial_{\mathbf{f}}^{2}\zeta^{*}. \tag{66}$$

$\hat{Z}_{\varepsilon}$ and $\hat{\mathbf{x}}_{\mathrm{a}}$ can be shown to exist; they can be read off from the explicit expressions of $Z_{\varepsilon}$ and $\mathbf{x}^{\mathrm{a}}$ as functions of $\mathbf{f}$. These Jacobian and Hessians depend on the choice of the regularisation operator and they need to be computed analytically, which is simple but tedious, and not reported here since this is a regularisation-dependent calculation. The Hessian of the dual cost function Eq. (57) can then be written as the sum

$$\partial_{\mathbf{f}}^{2}J_{\varepsilon}^{*} = \boldsymbol{\Lambda}_{\mathrm{bo},\mathrm{bo}} + \boldsymbol{\mathcal{H}}\boldsymbol{\Omega}_{\mathrm{bo},\mathrm{bo}}\boldsymbol{\mathcal{H}}^{\mathsf{T}}, \tag{67}$$

while the sensitivity matrix now reads

$$\partial_{\mathbf{f}}\mathbf{x}^{\mathrm{a}} = \boldsymbol{\Omega}_{\mathrm{bo},\mathrm{a}}^{\mathsf{T}}\boldsymbol{\mathcal{H}}^{\mathsf{T}}. \tag{68}$$

Note that $\boldsymbol{\Omega}_{\mathrm{bo},\mathrm{bo}}$ can be interpreted as the covariance matrix of $\mathbf{x}$, the compounded observable vector as defined in Eq. (56) though seen as a random vector, on the assumption that $\mathbf{x}^{\mathrm{b}}$ and $\mathbf{x}^{\mathrm{o}}$ are connected via the W-barycentre $\mathbf{x}^{\mathrm{a}}$ and the optimal transference plans $\mathbf{P}^{\mathrm{ba}}$ and $\mathbf{P}^{\mathrm{oa}}$, all seen as random vectors. Combining Eqs. (67,68) with Eq. (60), we finally obtain the expression for a factor $\mathbf{X}^{\mathrm{a}}$ of $\mathbf{P}^{\mathrm{a}}$:

$$\mathbf{X}^{\mathrm{a}} = \boldsymbol{\Omega}_{\mathrm{bo},\mathrm{a}}^{\mathsf{T}}\boldsymbol{\mathcal{H}}^{\mathsf{T}}\left[\boldsymbol{\Lambda}_{\mathrm{bo},\mathrm{bo}} + \boldsymbol{\mathcal{H}}\boldsymbol{\Omega}_{\mathrm{bo},\mathrm{bo}}\boldsymbol{\mathcal{H}}^{\mathsf{T}}\right]^{-1}\boldsymbol{\Lambda}_{\mathrm{bo},\mathrm{bo}}^{\frac{1}{2}}, \tag{69}$$

or, alternatively using the Sherman-Morisson-Woodbury transformation, while assuming $\boldsymbol{\Omega}_{\mathrm{bo},\mathrm{bo}}$ to be invertible,

$$\mathbf{X}^{\mathrm{a}} = \boldsymbol{\Omega}_{\mathrm{bo},\mathrm{a}}^{\mathsf{T}}\boldsymbol{\Omega}_{\mathrm{bo},\mathrm{bo}}^{-1}\left[\boldsymbol{\Omega}_{\mathrm{bo},\mathrm{bo}}^{-1} + \boldsymbol{\mathcal{H}}^{\mathsf{T}}\boldsymbol{\Lambda}_{\mathrm{bo},\mathrm{bo}}^{-1}\boldsymbol{\mathcal{H}}\right]^{-1}\boldsymbol{\mathcal{H}}^{\mathsf{T}}\boldsymbol{\Lambda}_{\mathrm{bo},\mathrm{bo}}^{-\frac{1}{2}}. \tag{70}$$

These formulas are similar to the normal equations of classical DA. But mind that, in Eqs. (69,70), all the prior error statistics are encapsulated in $\boldsymbol{\Lambda}_{\mathrm{bo},\mathrm{bo}}$ whereas the impact of OT is encoded in $\boldsymbol{\Omega}_{\mathrm{bo},\mathrm{bo}}$. To be concrete, note that, when using Gaussian statistics Eq. (12), $\boldsymbol{\Lambda}_{\mathrm{bo},\mathrm{bo}}$ would simply read

$$\boldsymbol{\Lambda}_{\mathrm{bo},\mathrm{bo}} = \begin{bmatrix} \boldsymbol{\Lambda}_{\mathrm{bb}} & \mathbf{0} \\ \mathbf{0} & \boldsymbol{\Lambda}_{\mathrm{oo}} \end{bmatrix} = \begin{bmatrix} \mathbf{B} & \mathbf{0} \\ \mathbf{0} & \mathbf{R} \end{bmatrix}. \tag{71}$$





## 4.2 Interpretation

Further, we can perform a block decomposition of $\mathbf{\Omega}$ conformally to the spaces of $\mathbf{x}^{\mathrm{b}}$ and $\mathbf{x}^{\mathrm{o}}$:

$$\mathbf{\Omega}_{\mathrm{bo,bo}} \triangleq \begin{bmatrix} \mathbf{\Omega}_{\mathrm{bb}} & \mathbf{\Omega}_{\mathrm{bo}} \\ \mathbf{\Omega}_{\mathrm{bo}}^{\mathsf{T}} & \mathbf{\Omega}_{\mathrm{oo}} \end{bmatrix}. \tag{72}$$

It can be shown that $\mathbf{\Omega}_{\mathrm{bo}}$ is proportional to the optimal transference plan of the effective transport between $\mathbf{x}^{\mathrm{o}}$ and $\mathbf{x}^{\mathrm{b}}$, and that the blocks of the diagonal are themselves diagonal and depend on the observable states

$$\mathbf{\Omega}_{\mathrm{bo}} = \frac{1}{\varepsilon}\mathbf{P}^{\mathrm{bo}}, \qquad \mathbf{\Omega}_{\mathrm{bb}} = \frac{1}{\varepsilon}\operatorname{diag}(\mathbf{x}^{\mathrm{b}}), \qquad \mathbf{\Omega}_{\mathrm{oo}} = \frac{1}{\varepsilon}\operatorname{diag}(\mathbf{x}^{\mathrm{o}}). \tag{73}$$

For instance, this could be shown by the explicit computation of $\mathbf{\Omega}_{\mathrm{bo,bo}} = \varepsilon\partial_{\boldsymbol{\eta}}^2 \hat{Z}_\varepsilon$.

Let us now examine the impact of OT on the analysis error covariance matrix. We first define

$$\mathbf{\Delta} = \mathbf{\Lambda}_{\mathrm{bo,bo}}^{-\frac{1}{2}} \mathcal{H} \mathbf{\Omega}_{\mathrm{bo,bo}}^{\frac{1}{2}}, \tag{74}$$

whose thin singular value decomposition is $\mathbf{U}\mathbf{\Sigma}\mathbf{V}^{\mathsf{T}}$, where $\mathbf{U}$ is an orthogonal matrix of size $(\mathfrak{N}_{\mathrm{b}} + \mathfrak{N}_{\mathrm{o}}) \times (\mathfrak{N}_{\mathrm{b}} + \mathfrak{N}_{\mathrm{o}})$, $\mathbf{\Sigma}$ is a rectangular and diagonal matrix of size $(\mathfrak{N}_{\mathrm{b}} + \mathfrak{N}_{\mathrm{o}}) \times (N_{\mathrm{b}} + N_{\mathrm{o}})$, and $\mathbf{V}$ is an orthogonal matrix of size $(N_{\mathrm{b}} + N_{\mathrm{o}}) \times (N_{\mathrm{b}} + N_{\mathrm{o}})$. Then, we *standardise* Eq. (69) following, e.g., Sect. 2.4.1 in Rodgers (2000):

$$\mathbf{X}^{\mathrm{a}} = \mathbf{\Omega}_{\mathrm{bo,a}}^{\mathsf{T}} \mathcal{H}^{\mathsf{T}} \left[ \mathbf{\Lambda}_{\mathrm{bo,bo}} + \mathcal{H}\mathbf{\Omega}_{\mathrm{bo,bo}}\mathcal{H}^{\mathsf{T}} \right]^{-1} \mathbf{\Lambda}_{\mathrm{bo,bo}}^{\frac{1}{2}}, \tag{75a}$$

$$= \mathbf{\Omega}_{\mathrm{bo,a}}^{\mathsf{T}} \mathcal{H}^{\mathsf{T}} \mathbf{\Lambda}_{\mathrm{bo,bo}}^{-\frac{1}{2}} \left[ \mathbf{I}_{\mathfrak{bo}} + \mathbf{\Lambda}_{\mathrm{bo,bo}}^{-\frac{1}{2}} \mathcal{H}\mathbf{\Omega}_{\mathrm{bo,bo}}\mathcal{H}^{\mathsf{T}} \mathbf{\Lambda}_{\mathrm{bo,bo}}^{-\frac{1}{2}} \right]^{-1}, \tag{75b}$$

$$= \mathbf{\Omega}_{\mathrm{bo,a}}^{\mathsf{T}} \mathbf{\Omega}_{\mathrm{bo,bo}}^{-\frac{1}{2}} \mathbf{\Delta}^{\mathsf{T}} \left[ \mathbf{I}_{\mathfrak{bo}} + \mathbf{\Delta}\mathbf{\Delta}^{\mathsf{T}} \right]^{-1}, \tag{75c}$$

$$= \mathbf{\Omega}_{\mathrm{bo,a}}^{\mathsf{T}} \mathbf{\Omega}_{\mathrm{bo,bo}}^{-\frac{1}{2}} \mathbf{V}^{\mathsf{T}} \mathbf{\Sigma}^{\mathsf{T}} \left[ \mathbf{I}_{\mathfrak{bo}} + \mathbf{\Sigma}\mathbf{\Sigma}^{\mathsf{T}} \right]^{-1} \mathbf{U}^{\mathsf{T}}. \tag{75d}$$

Defining $\boldsymbol{\sigma} = (\mathbf{\Sigma}\mathbf{\Sigma}^{\mathsf{T}})^{\frac{1}{2}}$, which is square diagonal of size $(\mathfrak{N}_{\mathrm{b}} + \mathfrak{N}_{\mathrm{o}}) \times (\mathfrak{N}_{\mathrm{b}} + \mathfrak{N}_{\mathrm{o}})$, we obtain, up to a multiplication by an irrelevant orthogonal matrix on the right, an equivalent factor for $\mathbf{P}^{\mathrm{a}}$:

$$\mathbf{X}^{\mathrm{a}} = \mathbf{\Omega}_{\mathrm{bo,a}}^{\mathsf{T}} \mathbf{\Omega}_{\mathrm{bo,bo}}^{-\frac{1}{2}} \mathbf{V}^{\mathsf{T}} \frac{\boldsymbol{\sigma}}{\mathbf{I}_{\mathfrak{bo}} + \boldsymbol{\sigma}^2}. \tag{75e}$$

The diagonal values of $\boldsymbol{\sigma}$, denoted $\sigma_i \geq 0$, represent the independent degrees freedom (dof) of information that can be extracted from the observations, which in our case is the first guess $\mathbf{y}^{\mathrm{b}}$ and the traditional observations $\mathbf{y}^{\mathrm{o}}$, in contrast to Rodgers (2000) who only considers the dofs from $\mathbf{y}^{\mathrm{o}}$. The higher the $\sigma_i$, the more information attached to the dof of index $i$, and the more squeezed the corresponding direction in $\mathbf{X}^{\mathrm{a}}$ and $\mathbf{P}^{\mathrm{a}}$. From Eq. (74), and in particular its transpose: $\mathbf{\Delta}^{\mathsf{T}} = \mathbf{\Omega}_{\mathrm{bo,bo}}^{\frac{1}{2}} \mathcal{H}^{\mathsf{T}} \mathbf{\Lambda}_{\mathrm{bo,bo}}^{-\frac{1}{2}}$ we can trace the flow of any piece of information. Such piece of information stems from the observation vectors, and hence its flow starts in $\mathbf{\Delta}^{\mathsf{T}}$ from $\mathbf{\Lambda}_{\mathrm{bo,bo}}^{-\frac{1}{2}}$ the square root of the precision matrix $\mathbf{\Lambda}_{\mathrm{bo,bo}}^{-1}$. It is then transferred from the observation spaces to the observable spaces through $\mathcal{H}^{\mathsf{T}}$. It is finally optimally transported across the space of $\mathbf{x}^{\mathrm{b}}$ and $\mathbf{x}^{\mathrm{o}}$ by $\mathbf{\Omega}_{\mathrm{bo,bo}}$ whose off-diagonal block is proportional to the transference plan $\mathbf{P}^{\mathrm{bo}}$. Hence, OT is not a primary source of uncertainty, as $\mathbf{y}^{\mathrm{b}}$ and $\mathbf{y}^{\mathrm{o}}$ can be, but moves information in between the observable spaces.



Let us now check the OTDA analysis error covariance matrix $\mathbf{P}^{\mathrm{a}}$ in the classical DA limit. To that end, we study Eq. (69) in the classical limit. Similarly to $\boldsymbol{\Omega}_{\mathrm{bb}}$ and $\boldsymbol{\Omega}_{\mathrm{oo}}$ in Eq. (73), $\boldsymbol{\Omega}_{\mathrm{aa}}$ is defined as the covariance matrix of $\mathbf{x}^{\mathrm{a}}$ when only accounting for both OTs, and it can be shown that it reads

$$\boldsymbol{\Omega}_{\mathrm{aa}} = \frac{1}{\varepsilon}\operatorname{diag}(\mathbf{x}^{\mathrm{a}}). \tag{76}$$

When the cost tends to $\mathbf{C}_{\mathrm{bo}}^{\infty}$, following the same arguments as in Sect. 2.4.4, $\mathbf{x}^{\mathrm{b}}$, $\mathbf{x}^{\mathrm{o}}$, and $\mathbf{x}^{\mathrm{a}}$ must merge and, consequently, $\boldsymbol{\Omega}_{\mathrm{bo}} = \boldsymbol{\Omega}_{\mathrm{aa}} = \boldsymbol{\Omega}_{\mathrm{bb}} = \boldsymbol{\Omega}_{\mathrm{oo}}$. Hence, in this limit $\boldsymbol{\Omega}_{\mathrm{bo,bo}} = \mathbf{1}_2\boldsymbol{\Omega}_{\mathrm{aa}}\mathbf{1}_2^{\mathsf{T}}$, and $\boldsymbol{\Omega}_{\mathrm{bo,a}} = \mathbf{1}_2\boldsymbol{\Omega}_{\mathrm{aa}}$, with $\mathbf{1}_2 = \begin{bmatrix} 1 & 1 \end{bmatrix}^{\mathsf{T}}$. Then, substituting these expressions of $\boldsymbol{\Omega}_{\mathrm{bo,bo}}$ and $\boldsymbol{\Omega}_{\mathrm{bo,a}}$ into Eq. (69), we get

$$\mathbf{X}^{\mathrm{a}} = \boldsymbol{\Omega}_{\mathrm{aa}}\mathbf{1}_2^{\mathsf{T}}\boldsymbol{\mathcal{H}}^{\mathsf{T}}\left[\boldsymbol{\Lambda}_{\mathrm{bo,bo}} + \boldsymbol{\mathcal{H}}\mathbf{1}_2\boldsymbol{\Omega}_{\mathrm{aa}}\mathbf{1}_2^{\mathsf{T}}\boldsymbol{\mathcal{H}}^{\mathsf{T}}\right]^{-1}\boldsymbol{\Lambda}_{\mathrm{bo,bo}}^{\frac{1}{2}}, \tag{77a}$$

$$= \boldsymbol{\Omega}_{\mathrm{aa}}\mathbf{1}_2^{\mathsf{T}}\boldsymbol{\mathcal{H}}^{\mathsf{T}}\left[\mathbf{I}_{\mathrm{bo}} + \boldsymbol{\Lambda}_{\mathrm{bo,bo}}^{-1}\boldsymbol{\mathcal{H}}\mathbf{1}_2\boldsymbol{\Omega}_{\mathrm{aa}}\mathbf{1}_2^{\mathsf{T}}\boldsymbol{\mathcal{H}}^{\mathsf{T}}\right]^{-1}\boldsymbol{\Lambda}_{\mathrm{bo,bo}}^{-\frac{1}{2}}, \tag{77b}$$

$$= \boldsymbol{\Omega}_{\mathrm{aa}}\left[\mathbf{I}_{\mathrm{a}} + \mathbf{1}_2^{\mathsf{T}}\boldsymbol{\mathcal{H}}^{\mathsf{T}}\boldsymbol{\Lambda}_{\mathrm{bo,bo}}^{-1}\boldsymbol{\mathcal{H}}\mathbf{1}_2\boldsymbol{\Omega}_{\mathrm{aa}}\right]^{-1}\mathbf{1}_2^{\mathsf{T}}\boldsymbol{\mathcal{H}}^{\mathsf{T}}\boldsymbol{\Lambda}_{\mathrm{bo,bo}}^{-\frac{1}{2}}, \tag{77c}$$

$$= \left[\boldsymbol{\Omega}_{\mathrm{aa}}^{-1} + \mathbf{1}_2^{\mathsf{T}}\boldsymbol{\mathcal{H}}^{\mathsf{T}}\boldsymbol{\Lambda}_{\mathrm{bo,bo}}^{-1}\boldsymbol{\mathcal{H}}\mathbf{1}_2\right]^{-1}\mathbf{1}_2^{\mathsf{T}}\boldsymbol{\mathcal{H}}^{\mathsf{T}}\boldsymbol{\Lambda}_{\mathrm{bo,bo}}^{-\frac{1}{2}}, \tag{77d}$$

where $\mathbf{I}_{\mathrm{a}}$ is the identity matrix of size $N_{\mathrm{a}}$. From Eq. (77b) to Eq. (77c) we relied on the shift matrix lemma (e.g., Asch et al., 2016). For $\boldsymbol{\Omega}_{\mathrm{aa}}^{-1}$ in Eq. (77d) to exist, it must be assumed that $\mathbf{x}^{\mathrm{a}} \in \mathcal{O}_{N_{\mathrm{a}}}^{+,\star}$, i.e. all the entries of $\mathbf{x}^{\mathrm{a}}$ are positive. This is verified when using entropic regularisation with $\varepsilon > 0$, no matter how small the entries of $\mathbf{x}^{\mathrm{a}}$ can be. Moreover, if $\mathbf{x}^{\mathrm{a}}$ has zero entries, $\mathbf{x}^{\mathrm{a}}$ can be represented as the limit of a sequence of positive discrete measures.

Now, since we have

$$\mathbf{A}^{-1} \triangleq \mathbf{1}_2^{\mathsf{T}}\boldsymbol{\mathcal{H}}^{\mathsf{T}}\boldsymbol{\Lambda}_{\mathrm{bo,bo}}^{-1}\boldsymbol{\mathcal{H}}\mathbf{1}_2 = \boldsymbol{\Lambda}_{\mathrm{bb}}^{-1} + \mathbf{H}^{\mathsf{T}}\boldsymbol{\Lambda}_{\mathrm{oo}}^{-1}\mathbf{H}, \tag{78}$$

we conclude from Eq. (77d) that the classical limit of the analysis error covariance matrix is

$$\mathbf{P}^{\mathrm{a}} = \mathbf{X}^{\mathrm{a}}(\mathbf{X}^{\mathrm{a}})^{\mathsf{T}} = \left[\boldsymbol{\Omega}_{\mathrm{aa}}^{-1} + \mathbf{A}^{-1}\right]^{-1}\mathbf{A}^{-1}\left[\boldsymbol{\Omega}_{\mathrm{aa}}^{-1} + \mathbf{A}^{-1}\right]^{-1}. \tag{79}$$

If the limit of $\mathbf{x}^{\mathrm{a}}$ when $\varepsilon \to 0^+$ is in $\mathcal{O}_{N_{\mathrm{a}}}^{+,\star}$, then $\boldsymbol{\Omega}_{\mathrm{aa}}^{-1} = \varepsilon\operatorname{diag}(\mathbf{x}^{\mathrm{a}})^{-1}$ must vanish. In this case:

$$\mathbf{P}^{\mathrm{a}} \xrightarrow[\varepsilon \to 0^+]{} \mathbf{A} = \left(\partial_{\mathbf{f}_{\mathrm{b}}}^2\zeta_{\mathrm{b}} + \mathbf{H}^{\mathsf{T}}\partial_{\mathbf{f}_{\mathrm{o}}}^2\zeta_{\mathrm{o}}\mathbf{H}\right)^{-1}, \tag{80}$$

which, assuming Gaussian errors, would read $\mathbf{P}^{\mathrm{a}} = \left(\mathbf{B}^{-1} + \mathbf{H}^{\mathsf{T}}\mathbf{R}^{-1}\mathbf{H}\right)^{-1}$, as expected from classical DA. However, if some of the entries of $\mathbf{x}^{\mathrm{a}}$ vanish in the limit $\varepsilon \to 0^+$, we suspect that the limit of $\mathbf{P}^{\mathrm{a}}$ will be the classical analysis error covariance matrix $\mathbf{A}$ but with the columns and rows associated to the vanishing entries of $\mathbf{x}^{\mathrm{a}}$ tapered to $0$.

## 4.3 Numerical illustration

We consider the one-dimensional example where one half of the domain is observed, over $\left[0, \frac{1}{2}\right]$, $\mathbf{H} \in \mathcal{O}_{\mathfrak{N}_{\mathrm{o}} \times N_{\mathrm{o}}}^+$ with $\mathfrak{N}_{\mathrm{o}} = N_{\mathrm{o}}/2$ and $H_l^j = \delta_{l,j}$ for $l \in [\![1, \mathfrak{N}_{\mathrm{o}}]\!]$ and $j \in [\![1, N_{\mathrm{o}}]\!]$; the observations are unbalanced, $\mathfrak{m}(\mathbf{y}^{\mathrm{b}}) = 1$ and $\mathfrak{m}(\mathbf{y}^{\mathrm{o}}) = 1.49$; they have





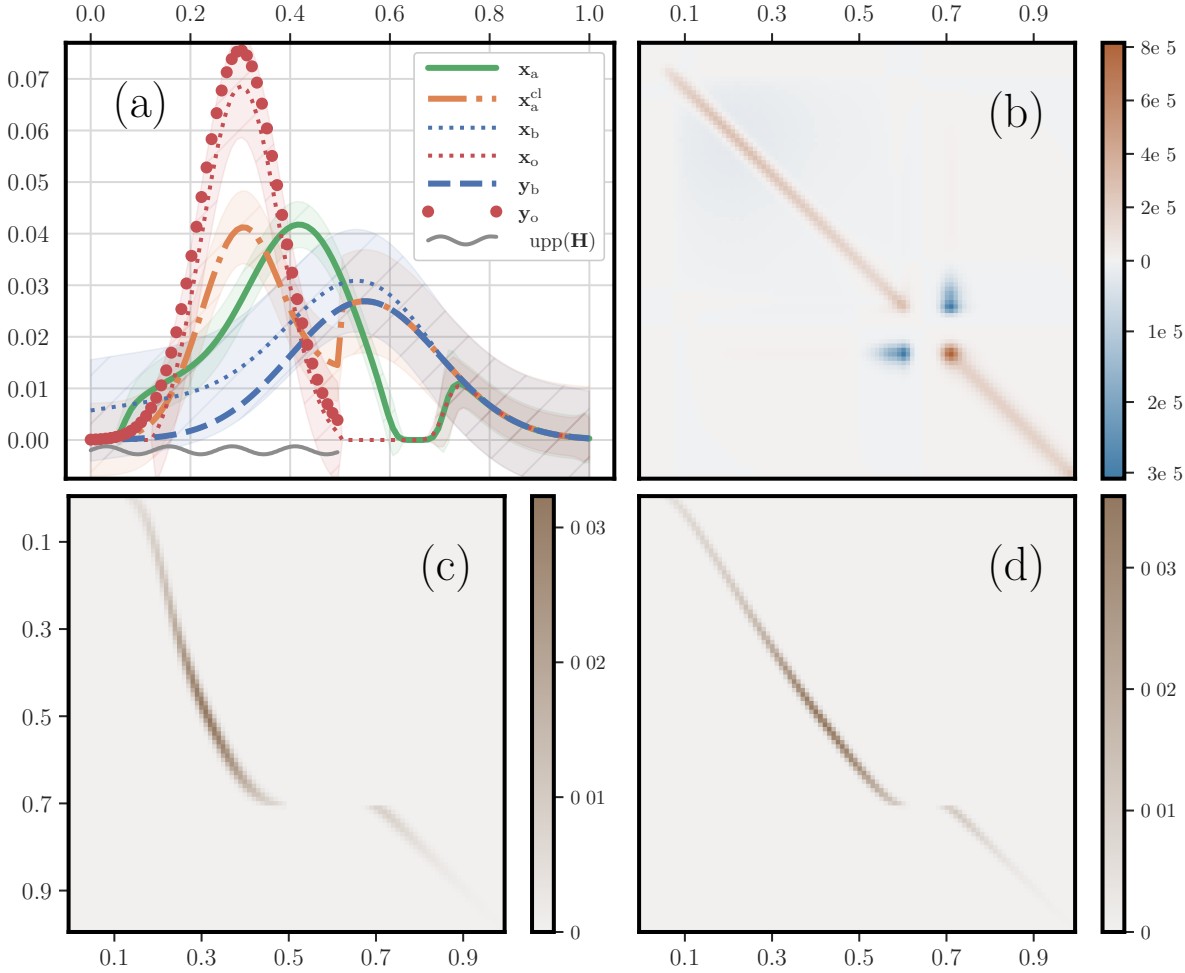

**Figure 15.** Illustration of the second-order analysis of an OTDA 3D-Var. Panel (a) shows the same plot as panel (a) of Fig. 12, but with errors bars (plus/minus the standard deviations) computed from the diagonal of the diagnosed posterior error covariance matrices associated to $\mathbf{x}^\mathrm{a}$, $\mathbf{x}^\mathrm{b}$, and $\mathbf{x}^\mathrm{o}$. Panel (b) displays the analysis error covariance matrix $\mathbf{P}^\mathrm{a}$. Panel (c) shows the optimal transference plan $\mathbf{P}^\mathrm{bo}$. Panel (d) shows the a, b block part of the Jacobian matrix $\boldsymbol{\Omega}_\mathrm{bo,a}$, which is denoted $\boldsymbol{\Omega}_\mathrm{ba}$.

been generated through $\mathcal{H}$ by discrete measures of mass 1 and 1.5, respectively; The cost metric has a quadratic dependence with the distance between sites, i.e. $[\mathbf{C}_\mathrm{ba}]_{ik} = \lambda |r_i^\mathrm{b} - r_k^\mathrm{a}|^2$ and $[\mathbf{C}_\mathrm{oa}]_{jk} = \lambda |r_j^\mathrm{o} - r_k^\mathrm{a}|^2$, where $\lambda = 10^3$. We use the results of Sect. 4.1 to compute the analysis error covariance matrix $\mathbf{P}^\mathrm{a}$, the transference plan $\mathbf{P}^\mathrm{bo}$, and the Jacobian $\boldsymbol{\Omega}_\mathrm{bo,a}$. The numerical results are displayed in Fig. 15. The OTDA analysis state is bimodal, some mass being left over to the right of the domain to account for the long tail of the first guess, which is far from the observation support. Hence, there is a vanishing field region, roughly $[0.6, 0.7]$, which separates the two components of the analysis state. As expected from OT theory, $\mathbf{P}^\mathrm{bo}$ seems





to converge towards a (non-trivial and barely differentiable) Monge map which, in this discrete context, has two branches, separated by the gap created by the vanishing field region. The analysis error covariance matrix $\mathbf{P}^a$ seems to converge to a diagonal matrix, with the exception of the vanishing field region. Indeed, there seems to be an uncertainty as to how much mass should be transferred from the first guess tail $[0.7, 1]$ to the main region $[0, 0.6]$. This is given away by peaks of variances near the edges of the gap, and by negative covariances between the two edge points of the gap.

## 5   Conclusions

In this paper, we have introduced a theoretical framework for integrating nonlocal optimal transport (OT) metrics into data assimilation (DA), which we refer to as *hybrid OTDA*. This framework addresses the inconsistencies initially identified by Feyeux et al. when local metrics in classical DA are replaced with nonlocal ones based on OT.

Our focus has been on defining a 3D-Var approach for hybrid OTDA and deriving the first- and second-order moments of its analysis. The hybrid OTDA 3D-Var method blends classical DA and its background and observation error statistics with a Wasserstein barycentre problem involving the observables associated with the first-guess and the observation vector. Importantly, our work demonstrates that classical DA is encompassed within this theoretical framework.

We have shown that this optimisation problem can be decomposed and simplified into a hybrid OTDA problem with a single OT problem based on an effective cost. This first problem yields the estimated $\mathbf{x}^b$ and $\mathbf{x}^o$, followed by a pure W-barycentre problem involving these states, whose solution is known as the McCann interpolant. This W-barycentre computation serves as the final analysis step.

Our proposed method can be applied to sparsely and noisily observed systems, as expected from a robust DA method. It can also accommodate non-trivial error statistics typical of a 3D-Var approach. Furthermore, we have illustrated the method's flexibility in defining cost metrics through various 1D and 2D numerical examples. We have empirically checked how the OTDA analysis shifts towards the classical DA analysis, within the OTDA framework.

Note that, for now, some limitations apply; mainly the framework is presently meant for non-negative fields.

While we have looked into several other promising developments of our methodology, we have chosen not to report them in this paper. These developments will be the subject of a future publication, including:

– the derivation of a Bayesian and probabilistic standpoint on OTDA,

– a generalised formalism where physical regularisation such as smoothness of the field can be enforced on the analysis state,

– a *stochastic matrix* formalism, which is a substitute to using transference plans, but could offer more robustness in the presence of entropic regularisation,

– employing cost matrices defined across several spaces, which is useful for realistic application where $\mathbf{x}^b$ and $\mathbf{x}^o$ lies in very distinct spaces, such as the space of emission of a pollutant, and the space of the pollutant concentrations, respectively.





While our primary focus in this paper was on the derivation and understanding of key cost functions within the hybrid OTDA framework, we did not delve much into the numerical challenges, algorithmic complexity, or computing acceleration. For this
700 aspect of the developments, we would rather rely on the developments of the experts of OT who are continuously improving on the efficiency of numerical OT (e.g., Flamary et al., 2021).

In addition to strengthening the developments mentioned above, our future research will explore the application of the hybrid OTDA formalism in a sequential DA framework, as this paper concentrated solely on the static analysis. We are also interested in investigating the role played by error statistics and cost metrics $\{\zeta_\mathrm{b}, \zeta_\mathrm{o}, \mathbf{C}_\mathrm{ba}, \mathbf{C}_\mathrm{oa}\}$ and their balancing in the hybrid OTDA
705 analysis, as well as developing their objective tuning.

*Code availability.* The products of this paper are exclusively optimisation problems and methods to solve them; their implementation (code) used in the illustrative sections rely on freely available software to solve the optimisation problems, mainly L-BFGS-B and its implementation in Scipy https://github.com/scipy/scipy and the Python Optimal Transport library and and its implementation https://github.com/PythonOT.

*Author contributions.* MB, PJV and AF developed the methodology. MB implemented the numerics. MB wrote the manuscript. MB, PJV,
710 AF, JDLB and YR revised the manuscript.

*Competing interests.* The contact author has declared that none of the authors has any competing interests.

*Acknowledgements.* This research has been supported by the national research project ANR-ARGONAUT (grant no. ANR-19-CE01-0007, PollutAnt and gReenhouse Gases emissiOns moNitoring from spAce at high ResolUTion). CEREA is a member of Institut Pierre-Simon Laplace (IPSL).





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
