# Peer review of "Bridging classical data assimilation and optimal transport: The 3D–Var case"

_EGUsphere, 2023_

## Author Response (AR1)

**Extended authors' response to Reviewer 1's comments for "Bridging classical data assimilation and optimal transport"**

Preprint egusphere-2023-2755

M. Bocquet , P. J. Vanderbecken, A. Farchi, J. Dumont Le Brazidec, and Y. Roustan

22 March 2024

*The paper discusses a very interesting and pertinent problem and proposes a solution that is certainly worth looking at further. The continued use of quadratic error functional is somewhat anachronistic given that the main motivation for using it are simplicity of analytic computations and an assumption of Gaussian errors. The first argument carries less weight in the age of supercomputers, and the second was always known to be wrong except in (important) special cases. I cannot, however, recommend the publication of the paper in its present form and believe that the paper requires a major revision. This is due to the following major concerns (see below for a few more minor concerns)*

We appreciate the reviewer's comments and suggestions. In the following, we discuss the raised concerns and what we have changed or will revise in the manuscript. The pdf document highlighting the differences between the original and the revised manuscript is provided.

*(1) I find the first section very confusing. The authors are discussing problems related to the so-called double penalty error and the non-overlap of functions (or distributions) that appear in data assimilation. It is not clear however on what level the authors are working. More specifically, it seems first that the authors want to work on the level of probability distributions for state variables. Later however it turns out that they want to work on the level of state variables directly, yet focus on those that represent meteorological fields which essentially have the character of distributions. This, however, has to be clear from the very beginning.*

It has been made clear in the manuscript that the theory applies to fields, not probability distributions. For instance in section 1.4:
*In the context of this paper, it is critical to be aware that the use of OT in practical DA focused so far on applying OT to the pdf of a single variable. Quite often, OT is applied to the pdf of a single random variable because... This is very different from our context and objective where the objects dealt with by OT are field states, not the pdf of one of their variables.*

Or, in section 1.5, Objective and Outline, we state:
*At least within the perimeter of this paper, some restrictions apply compared to traditional DA. Firstly, the physical fields considered in the DA problem are non-negative (concentration of tracer, pollutants, water vapour, hydrometeors, chemical and biogeochemical species, etc.). However, as opposed to Feyeux (2016), the methods of this paper do not require the (possibly noisy) background state $\mathbf{y}^{\mathrm{b}}$ and observation $\mathbf{y}^{\mathrm{o}}$ to be non-negative. Secondly, the observation operator $\mathbf{H}$ is assumed to be linear. This is only meant for convenience and to obtain a rigorous correspondence between the primal and dual cost functions of the 3D-Var. Finally, we stress once again that the states of our DA problem are physical fields onto which OT is applied and are not meant to be pdf of a random variable.*

That said, we fully agree with the reviewer that the second motivational argument (overlapping of fields) could generate a lot of confusion and could even be misleading, especially so early in the manuscript. We explain below with the third point raised how we addressed this issue. We have also made sure that the application of OT to non-negative physical fields was unambiguously stated, at every important step of the abstract and introduction (please see the difference document).

*(2) I understand what the authors label as the first (of two) weaknesses of classical DA which is often termed the double penalty error. This problem however is ultimately an issue resulting from a mismatch between the employed distance and the smoothness of meteorological fields. If the correct metric is selected depending on the smoothness of the meteorological fields, there is no double penalty problem. Labelling this as a problem of "classical DA" however implies that classical DA only uses the mean square error which is not correct (error covariances are important part of the error functional and have a strong influence on whether the double penalty problem occurs or not).*

In the light of your comments, we have decided to renounce calling the main issue (double penalty error) a weakness of classical DA since it is instead a weakness in the choice of the metrics which, as a result, impacts data assimilation. We hope the revision will be clearer in that regard.

However, we would like to stress that the double-penalty issue, which comes from a (model) location error of fields, does not disappear by a smart choice of a weighted Euclidean distance (Mahalanobis distance). It is true that inflating the correlation length of the covariance matrices is a known trick to mitigate the double penalty error but it does not solve it, since it cannot correct for the location of fields (e.g., Hoffman et al., 1995). The weighted Euclidean distance only measures amplitude and smoothness mismatches, not the full *distortion*. This was a key incentive for the stream of work mentioned in the introduction (e.g.. Hoffman et al., 1995; Hoffman and Grassotti, 1996; Ravela et al., 2007; Ning et al., 2014; Feyeux et al., 2018). Here is a thought experiment to stress this: one can imagine inflating the correlation length of the covariance matrix of a weighted Euclidean distance of two mislocated peaked fields which are far apart. But it could become a blunt tool with poor separation power. Indeed, it is not difficult to check that in the very large correlation length limit, such norm only compares the mean of the two fields, i.e. only one degree of freedom.

This is now clarified/further discussed in the motivational section 1.1 of the introduction:

"It has been recognised that while the weighted Euclidean (Mahalanobis) distance can handle amplitude and smoothness mismatch, it cannot cope with mislocation error and hence account for the full *distortion* between mismatched fields (Hoffman et al., 1995). Hence, even though tuning covariances of Gaussian error distributions as in classical DA, such as increasing the correlation length, might help mitigate the double-penalty error, it is insufficient. In Fig. 1, one might replace the Euclidean norm by a weighted Euclidean one with a large correlation length. This would yield similar norm values for both cases. Unfortunately, it is not difficult to show that in this limit this (almost singular) norm can only distinguish between the spatial mean of both fields; it became blunt with no discriminating power."

*(3) Related to the previous question, I do NOT understand what the authors label as the second weakness of classical DA. In fact, the last paragraph of Sec. 1.1 hardly makes any sense when it talks about "overlap in space and time" between background and observations. The material seems to draw on intuition coming from the Bayes rule but that applies to probability densities; the 3D-Var analysis is an operator-convex combination of meteorological fields which is something completely different.*

We agree that the motivational argument was too confusing, especially at the very beginning of the manuscript. Hence, in the light of your comments, we have decided to withdraw it. Hence section 1.1, now entitled *Data assimilation and the double-penalty issue* has been rewritten and is now mainly focused on the double-penalty error. Thank you for pointing out the lack of clarity of the original section 1.1.

*(4) As far as I can see, the technique can only assimilate meteorological fields that essentially behave like distributions, which is clearly a major restriction. I believe the assumption that the fields be positive is not enough (after all, by choice of origin any meteorological field can be assumed to have non-negative values). All the examples mentioned by the authors are extensive quantities. Is there an issue with applying the approach to an intensive quantity such as, say, the temperature?*

Yes, the technique has a major limitation in that it is meant for non-negative fields. We have

been very upfront about this. Note, however, that we have introduced a way to address the unbalanced case where the mass of the origin and target fields are not equal. That is why we speak of non-negative fields, not distributions (which is too restrictive).

Yes, the assumption that the fields are non-negative is sufficient, at least mathematically. If one wishes, the origin of fields could be redefined to make them non-negative and our theory could be applied. But of course such trick would introduce major biases that would force to reconsider how to apply data assimilation. And what would be the origin of fields for wind components? That is why we prefer to emphasise potential applications to chemical species fields, biogeochemical fields, water vapour, etc., where location errors can be significant.

No, there is no issue applying OTDA to temperature. But of course, temperature fields are known to be much smoother; they do not exhibit strong location errors, compared to, say, the relative or specific humidity (which are intensive fields as well). Hence, it is a priori more interesting to apply OTDA to, e.g., humidity fields where, for this reason, the double-penalty error is more prominent, and perhaps for fields with exact or approximate conservation laws. But these are not necessary conditions for the applicability of the tool.

*(a) It is not clear how the authors deal with comparing fields that do not have the same mass. Also, it is not clear what the cumbersome result in Fig. 5 has to do with the assumption whether or not the two fields have the same mass.*

The reviewer refers to Fig. 5 (Fig. 4 in the revised manuscript), where the solution that we proposed has not been introduced yet. At this stage in the paper, we only state the nature of the problem, and explain that it was deemed so far unclear how to compare fields that do not have the same mass. Our solution is only introduced later on, starting from section 2.3. We stress this in the revised manuscript.

As for the second comment, the assumption that $\mathbf{y}^{\mathrm{b}}$ and $\mathbf{y}^{\mathrm{o}}$ have the same mass was merely for convenience; this is not crucial to the argument. We have clarified the corresponding statement and sentence. (But the fact that OT – not OTDA –conserves mass by definition is of course critical to the derivation we provide.)

*(b) The concept of the entropy regularisation is not clear. It is not even clear why this renders the problem convex or at least uniquely solvable.*

We follow a classical technique used by the applied mathematicians in the field of numerical optimal transport. As clearly mentioned when introducing the technique, we refer to the nice and accessible textbook of Peyré and Cuturi (2019):
*Hence, entropic regularisation is used to render the problem strictly convex. A comprehensive justification is given by Peyré and Cuturi (2019).*

We have extended the sentence referencing their book and now precisely points to where the proof of convexity is given if it can help. That said, since the proof is simple and enlightening, we can give it in this response and in the revised manuscript. Indeed, the regularised problem has a cost function of the form

$$\varepsilon \mathcal{K}(\mathbf{p}|\mathbf{q}) + \mathbf{c}^{\mathsf{T}}\mathbf{p} = \varepsilon \sum_i p_i \ln\left(\frac{p_i}{q_i} - p_i\right) + q_i + \sum_i c_i p_i \tag{1}$$

to be minimised over the $p_i$, and where $\varepsilon > 0$. The Hessian of this cost function is readily computed and it is diagonal with entries

$$\varepsilon/p_i > 0, \tag{2}$$

which, since $0 \leq p_i \leq 1$, shows that the optimisation problem is $\varepsilon$–strongly convex and in particular strictly convex as long as $\varepsilon > 0$.

We have added the following sentence:

It can be checked that the Hessian of the regularised cost function Eq. (15) is a diagonal matrix of coefficients $\varepsilon/P_{ij}^{\mathrm{bo}} \geq \varepsilon$ since $0 \leq P_{ij}^{\mathrm{bo}} \leq 1$, making the problem $\varepsilon$–strongly convex.

*(c) There are other ways to measure distances between meteorological fields that avoid or alleviate the double penalty error (depending on the smoothness of the fields). Have the authors compared their Wasserstein approach with other metrics, also given that the entropy regularised Wasserstein distances are not easy to calculate and optimise?*

We did not, since the main focus of the paper was to show how to merge classical data assimilation and the optimal transport. One of the reason why we chose optimal transport over other approaches is that there is a large community of applied mathematicians that built solid foundations for this tool, and recently made progress on the numerical implementation (Peyré and Cuturi, 2019; Flamary et al., 2021), so that computing the Wasserstein distance is not as challenging as it used to be. Further, in order to make a meaningful comparison, we would need a concrete, practical case with real data, which is beyond the scope of this paper.

*(d) On pg 6 the authors claim that "our problem is not subject to the curse of dimensionality". Although this is true, the curse of dimensionality (in the sense implied by the discussion here) is not actually a concern in 3D-Var either which is what the method should be compared with. So this remark is misleading.*

The remark was only intended for readers that use optimal transport theory and apply it to probability distributions in the field of geosciences (which includes some of us) and may have thought that the present theory is impacted by the curse of dimensionality as well. Nothing more was intended, especially in the scope of this introduction.

**References**

Feyeux, N.: Transport optimal pour l'assimilation de données images, Ph.D. thesis, Université Grenoble Alpes, `https://inria.hal.science/tel-01480695`, 2016.

Feyeux, N., Vidard, A., and Nodet, M.: Optimal transport for variational data assimilation, Nonlin. Processes Geophys., 25, 55–66, https://doi.org/10.5194/npg-25-55-2018, 2018.

Flamary, R., Courty, N., Gramfort, A., Alaya, M. Z., Boisbunon, A., Chambon, S., Chapel, L., Corenflos, A., K., F., Fournier, N., Gautheron, L., Gayraud, N. T. H., Janati, H., Rakotomamonjy, A., Redko, I., Rolet, A., Schutz, A., Seguy, V., Sutherland, D. J., Tavenard, R., Tong, A., and Vayer, T.: POT: Python Optimal Transport, Journal of Machine Learning Research, 22, 1–8, `http://jmlr.org/papers/v22/20-451.html`, 2021.

Hoffman, R. N. and Grassotti, C.: A Technique for Assimilating SSM/I Observations of Marine Atmospheric Storms: Tests with ECMWF Analyses, Journal of Applied Meteorology and Climatology, 35, 1177–1188, https://doi.org/10.1175/1520-0450(1996)035¡1177:ATFASO¿2.0.CO;2, 1996.

Hoffman, R. N., Liu, Z., Louis, J.-F., and Grassoti, C.: Distortion representation of forecast errors, Mon. Wea. Rev., 123, 2758–2770, https://doi.org/10.1175/1520-0493(1995)123¡2758:DROFE¿2.0.CO;2, 1995.

Ning, L.and Carli, F. P., Ebtehaj, A. M., Foufoula-Georgiou, E., and Georgiou, T. T.: Coping with model error in variational data assimilation using optimal mass transport, Water Resources Research, 50, 5817–5830, https://doi.org/https://doi.org/10.1002/2013WR014966, 2014.

Peyré, G. and Cuturi, M.: Computational Optimal Transport: With Applications to Data Science, Foundations and Trends® in Machine Learning, 11, 355–607, https://doi.org/10.1561/2200000073, 2019.

Ravela, S., Emanuel, K., and McLaughlin, D.: Data assimilation by field alignement, Physica D, 230, 127–145, https://doi.org/10.1016/j.physd.2006.09.035, 2007.

**Extended authors' response to Reviewer 2' s comments for "Bridging classical data assimilation and optimal transport"**

Preprint egusphere-2023-2755

M. Bocquet , P. J. Vanderbecken, A. Farchi, J. Dumont Le Brazidec, and Y. Roustan

22 March 2024

*The paper proposed a different objective function for the data assimilation problem, which is a mixed sum between the classic cost function and the Wasserstein metrics. With assumptions on strictly convexity and linearity, the problem is convex. The paper then derives the optimization conditions for the new objective function by combining convex optimization analysis and the duality concerned about the Wasserstein metric with entropy regularization. It is further supported by several 1D and 2D data assimilation problems.*

We appreciate the reviewer's comments and suggestions. In the following, we discuss the raised concerns and what we have changed or will revise in the manuscript. The pdf document highlighting the differences between the original and the revised manuscript is provided.

*(1) My biggest complaint is the paper title. I think the title does not match the content of the paper. Based on the title, it sounds like a providing a theoretical connection between classical data assimilation and OT. However, it is more like introducing an application of OT theory for data assimilation problems. I suggest the authors change the title of this paper to a more descriptive one.*

We have presented this work to many colleagues (in geophysical data assimilation with some knowledge of optimal transport) who did not feel this issue. But we can certainly make the title of the paper more specific. We have opted for "**Bridging classical data assimilation and optimal transport: The 3D-Var case**" since we are proposing the OTDA 3D-Var and a way to carry out its analysis for both first-order and second-order moments.

*(2) The assumption in equation (11) is rather strong. H is linear, and all the costs are convex. That means the paper only deals with log-concave distributions, which does not apply to the multi-modal distributions that are challenging to handle. The baseline DA (17) is a strictly convex problem that can be solved easily. Even if one is worried about noise overfitting, many existing good methods exist to handle this.*

The fact that $\mathbf{H}$ is linear is for convenience because it can make the dual optimal problem formally equivalent. This could be alleviated, by for instance sticking to a primal formulation. In any case, it must be kept in mind that assuming $\mathbf{H}$ linear is a common practice (with good reasons) in classical data assimilation through explicit linearisation, or implicitly as in the EnKF. For instance in one of the seminal paper of data assimilation (Courtier, 1997) about the dual formulation of 4D-Var (relevant to and cited in the manuscript), the authors assume linearisation of $\mathbf{H}$ from the very beginning of the paper owing to the so-called *incremental* formulation of 4D-Var.

So this assumption is actually not as shocking as it may seem. But we agree that as such, this is a strong assumption, out of the data assimilation context. Following your concern, we have added a bit a context in the revised manuscript to support the assumption.

"Secondly, the observation operator $\mathbf{H}$ is assumed to be *linear*. This is only meant for convenience and to obtain a rigorous correspondence between the primal and dual cost functions of the 3D–Var. Making this assumption is very common in geophysical DA: $\mathbf{H}$ can indeed be seen as the tangent linear of a nonlinear observation operator within the inner loop of a 3D–Var or a 4D–Var (see for instance Courtier, 1997)."

However, the fact that the problem is convex, or can be considered so, is a completely standard assumption to classical geophysical data assimilation. Even though the standard methods DA applied to high-dimensional systems (3D-Var, 4D-Var, EnKF) are not rigorously convex, they are in practice assumed to be so with one single outstanding minimum. Almost all the methods in operation rely on a linearisation and implicit Gaussian assumptions (see, e.g., Carrassi et al., 2018, and references within). Moreover, all the incremental variational formulations of 3D–Var and 4D–Var use a truly strongly convex cost function in their inner loop so as to leverage a conjugate gradient optimisation scheme. They would fail in presence of true multi-modality. With the exception of the linearity of $\mathbf{H}$, the statistical assumptions in Eq. (11) are actually more general that those used in classical 3D-Var (Daley, 1991).

We note that the problem of strong nonlinearity, non-Gaussianity, and multi-modal distributions, a problem the authors are not unfamiliar with (e.g., Bocquet et al., 2010; Farchi and Bocquet, 2018), is really far apart from the goal of this paper which is to replace local metrics with nonlocal ones in classical DA, which is the only focus of the paper.

Finally, to answer the end of the comment, the objective of the paper is not to come back to the well-known (numerical) solution of the classical 3D-Var, but to address the extended hybrid OTDA problem.

*(3) Between (17) and (18), there are new introductions of $\mathbf{x}^{\mathrm{b}}$ and $\mathbf{x}^{\mathrm{o}}$ with Wasserstein metrics, turning the problem from strongly convex to a mixed problem. The motivation is not very clearly stated. The significantly increased computational cost associated with (18) has to be supported by very strong reasons. For example, what properties can we achieve by combining these two different cost functions?*

This *strongly convex problem* is the basic 3D-Var cost function of classical data assimilation, which is very common in geophysical data assimilation, which we indeed turn into a mixed problem to tackle the double-penalty, location error issues.

The main motivation for introducing non-local metrics into classical data assimilation is that of model error, and especially model error based on location errors, which is ubiquitous in the geosciences and leads to the so-called *double-penalty error*. Following your comment and that of reviewer 1, we have clarified the introduction (which is key to understanding why one would move from Eq. (17) to Eq. (18)), especially its beginning which could have been misleading in that regard. Note that this motivation was put forward in a long stream of works mentioned in the introduction, e.g., Hoffman et al. (1995); Ravela et al. (2007); Ning et al. (2014); Feyeux (2016) who also motivate the introduction of non-local metrics in the fields of verification and data assimilation. We are to a very large extent making this motivation ours.

Also, please be aware that Sect. 1.4 of the (original and revised) introduction was precisely building arguments in favour of Eq. (18).

To further address your concern, that should be partially alleviated by the revised introduction, we have added the following text when emphasising the linearity of $\mathbf{H}$:

Moving from Eq. (17) to Eq. (18) following the principles and guidance of the introductory Sect. 1.4 is empirical, but no more than in Ning et al. (2014); Feyeux (2016). Showing its merits is the goal of the present paper. As opposed to Feyeux et al. (2018), it can deal with sparse and noisy observations, i.e. non-trivial $\mathbf{H}$. We will show that classical DA is embedded in this generalisation. Moreover, the merits of the new cost function will be a posteriori qualitatively supported by the outcome of the numerical experiments (to the expert's eyes), which improve over previous formalism's outcomes. We would like to point out that we have also developed a consistent probabilistic and Bayesian formalism fully supporting the introduction of Eq. (18). However we felt that the derivation is too long and technical for this paper, and would not be helpful in the exploration of the direct consequences of Eq. (18).

*(4) Section 2.4 has too many details about the derivation that are standard steps in convex optimization. I suggest putting many in an appendix and only stating the main formula.*

The reviewer may have a strong mathematical/computer science background, which is not very common to many experts in geophysical data assimilation. But we agree that the manuscript could benefit from moving a few derivations in an appendix, which we did in the revised manuscript following your advice. Here is what we have done:

1. we moved the lengthy derivation from the full primal problem to the dual problem in Sect. 2.4.1 to Appendix A,

2. we moved the 3-page long two-step approach derivation of Sect. 2.4.2 and Sect. 2.4.3 to Appendix B.

The rest of Sect. 2 is important to the paper and hopefully instructive.

*(5) The numerical examples in Sections 3 & 4 are a bit too simple. Of course, 1D and 2D OT are not so costly. However, when the dimension becomes large, the extra two terms in (18) become increasingly cumbersome, and computational cost is forbidden. This work is for geoscience applications with often high-dimensional state space.*

We disagree with the reviewer. Firstly, the example that we provide are numerically significantly more challenging than those provided earlier on the same attempt to bridge classical data assimilation with optimal transport (Feyeux, 2016; Feyeux et al., 2018). Secondly and most importantly, the reviewer is mistaking the dimension of the vector space stemming from the discretised physical space for the dimension of the physical space. **Although the physical space is here of dimensions 1 or 2, the discretised state space dimension goes here up to $10^4$ (to be multiplied by the number of control variable types, which is $3$ here).** Note that in geophysical data assimilation, *high-dimensional* refers to the dimension of the discretised vector space, not that of the underlying physical space (which is 1, 2, or 3, possibly but rarely 4).

*Overall, I feel the paper title is too big of a summary for the paper, and the numerical examples, on the other hand, are elementary. While I can relate the formulation from (6) to (7), which is very neat and has a clear mathematical intuition, the hybrid sum in (18) seems to be a "cocktail" of two different metrics. Further understanding is necessary even if the authors don't plan on proving any mathematical properties.*

Following your recommendation, we have changed the title. We believe that it is now really faithful to what is developed in the paper.

We understand your feeling as to the *cocktail of metrics*. But, please understand that our formalism is rather rigorous for the geophysical data assimilation and is actually shown to significantly improve upon the first attempts by Feyeux (2016). It also has the very pleasant property that classical data assimilation was shown to be embedded in the OTDA formalism, as opposed to the theories previously advocated. To us, this already makes a strong case for this cocktail.

Finally, to strengthen this approach, we have developed a consistent Bayesian and probabilistic view on the OTDA formalism, a view often supported in classical data assimilation to back the methods (Lorenc, 1986). However, we had the feeling it would have been too long and technical to be reported here, and only marginally relevant to *Nonlinear Processes in Geophysics* so that is was not included in the manuscript.

**References**

Bocquet, M., Pires, C. A., and Wu, L.: Beyond Gaussian statistical modeling in geophysical data assimilation, Mon. Wea. Rev., 138, 2997–3023, https://doi.org/10.1175/2010MWR3164.1, 2010.

Carrassi, A., Bocquet, M., Bertino, L., and Evensen, G.: Data Assimilation in the Geosciences: An overview on methods, issues, and perspectives, WIREs Climate Change, 9, e535, https://doi.org/10.1002/wcc.535, 2018.

Courtier, P.: Dual formulation of four-dimensional variational assimilation, Q. J. R. Meteorol. Soc., 123, 2449–2461, https://doi.org/10.1002/qj.49712354414, 1997.

Daley, R.: Atmospheric Data Analysis, Cambridge University Press, New-York, 1991.

Farchi, A. and Bocquet, M.: Review article: Comparison of local particle filters and new implementations, Nonlin. Processes Geophys., 25, 765–807, https://doi.org/10.5194/npg-25-765-2018, 2018.

Feyeux, N.: Transport optimal pour l'assimilation de données images, Ph.D. thesis, Université Grenoble Alpes, `https://inria.hal.science/tel-01480695`, 2016.

Feyeux, N., Vidard, A., and Nodet, M.: Optimal transport for variational data assimilation, Nonlin. Processes Geophys., 25, 55–66, https://doi.org/10.5194/npg-25-55-2018, 2018.

Hoffman, R. N., Liu, Z., Louis, J.-F., and Grassoti, C.: Distortion representation of forecast errors, Mon. Wea. Rev., 123, 2758–2770, https://doi.org/10.1175/1520-0493(1995)123¡2758:DROFE¿2.0.CO;2, 1995.

Lorenc, A. C.: Analysis methods for numerical weather prediction, Q. J. R. Meteorol. Soc., 112, 1177–1194, https://doi.org/10.1002/qj.49711247414, 1986.

Ning, L.and Carli, F. P., Ebtehaj, A. M., Foufoula-Georgiou, E., and Georgiou, T. T.: Coping with model error in variational data assimilation using optimal mass transport, Water Resources Research, 50, 5817–5830, https://doi.org/https://doi.org/10.1002/2013WR014966, 2014.

Ravela, S., Emanuel, K., and McLaughlin, D.: Data assimilation by field alignement, Physica D, 230, 127–145, https://doi.org/10.1016/j.physd.2006.09.035, 2007.

---

## Editor Decision (ED1)

Two referees have sent their evaluations. They are the same as the referees of version 1.

Referee 2 recommends acceptance of the paper as it stands.

Referee 1 is more critical. He/she answers *No* to the question *Is the presentation clear and concise* ?, and writes *The authors have addressed most of the comments, but from my point of view several issues remain*. He/she mentions a few specific points which have to do with basic clarification of the approach taken by the authors, and not on the results nor on the conclusions the authors draw from those results. He/she recommends acceptance subject to minor revisions, but does not rule out a possible further review. It is for keeping that possibility open that I ask as Editor for major revision.

In addition, I have as Editor a number of suggestions for corrections that are mostly independent of the scientific content of the paper, but could in my opinion improve its readability. I put them here in approximate order of decreasing importance.

1 Figure 14. Contrary to what the authors write, no error bars are visible on panel (a). The panel is actually identical with panel (a) of Fig. 11.

2 The authors write in subsection 2.5 (ll. 362-363) … *entropic regularisation is enforced via* …, but say nothing about entropic regularization in subsection 2.4. Does that mean that entropic regularization is not implemented in the algorithm described in subsection 2.4 ?

3 Ll. 335 and 597, what is a *Monge map* (a scheme of the form of Fig. 3) ?

4 Ll. 469- 471, I find the sentence starting *If $y^b$ and $y^o$ were* … rather vague. What does it mean that *the classical analysis would be as good as can be, while the OTDA solution may be too safe* ? That the sharp transitions visible on panel 12 (b) would not be there, and that the OTDA solution would be too smooth ? Or something else ?

5 L. 478, What does it mean that the first guess is *weak* ? That it is numerically small so as to be negligible, or erroneous, unreliable, or what ?

6 Caption of Fig. 12, what is a *heatmap* (I suspect a typo) ?

7 L. 155, *the kernel of* **H** *is non-trivial*. May be confusing. I suggest you simply write *The main caveat of Eq. (7) comes from the fact that the system is only partially observed.*

8 L. 537, … *conformally to the spaces* … $\rightarrow$ … *onto the spaces* …

9 L. 285 … *the same tensor index **is** present* …

10 From what I can judge, the English is correct and perfectly understandable. A few corrections will however have to be made here and there. The paper will be copy-edited anyway, but I mention one point. The authors repeatedly write *associated to* (l. 288 for instance). That should be *associated with* (incidentally, that is a great classic of French authors when writing English).

I look forward to receiving a new version of the paper, taking into account the comments and suggestions of referee 1, as well as my own. Should the authors disagree with a particular comment or decide not to follow a particular suggestion, they must state their reasons for that. I may submit the revised version to a new review by Referee 1.

---

## Author Response (AR2)

**Authors' response to the Editor's comments on the revised manuscript "Bridging classical data assimilation and optimal transport: The 3D-Var case"**

Preprint egusphere-2023-2755.R1

M. Bocquet , P. J. Vanderbecken, A. Farchi, J. Dumont Le Brazidec, and Y. Roustan

17 May 2024

We appreciate the Editor's commitment in reviewing the manuscript, his comments and suggestions. In the following, we provide answers to the questions and suggestions and describe what has been changed accordingly in the manuscript. The pdf document highlighting the differences between the original and the revised manuscript is provided.

*(1) Figure 14. Contrary to what the authors write, no error bars are visible on panel (a). The panel is actually identical with panel (a) of Fig. 11.*

By error bars we meant the shades encompassing each curve and whose boundaries correspond to plus and minus the standard deviations about each curve. These are the key additions compared to Fig. 11 (they are based on the same test configuration but are not identical). We modified the caption to be clearer:

... but with the addition of shaded regions delineated by plus and minus the standard deviations about the estimates for $\mathbf{x}^{\mathrm{a}}, \mathbf{x}^{\mathrm{a}}_{\mathrm{cl}}, \mathbf{x}^{\mathrm{b}}, \mathbf{x}^{\mathrm{o}}$. These standard deviations are computed from the diagonal of the diagnosed posterior error covariance matrices associated with $\mathbf{x}^{\mathrm{a}}, \mathbf{x}^{\mathrm{a}}_{\mathrm{cl}}, \mathbf{x}^{\mathrm{b}}$, and $\mathbf{x}^{\mathrm{o}}$.

*(2) The authors write in subsection 2.5 (ll. 362-363) ... entropic regularisation is enforced via ..., but say nothing about entropic regularization in subsection 2.4. Does that mean that entropic regularization is not implemented in the algorithm described in subsection 2.4 ?*

You are absolutely right. We did not mention entropic regularisation in Section 2.4 because it would have made the equations lengthier, and more obscure. However, in practice in numerics, we do implement entropic regularisation. By contrast, it is critical to account for the entropic regularisation in Section 2.5, since the section provides the final formulas used in the numerical experiments and because the few tricks used in the derivation deal with the entropic regularisation. We have added a sentence at the end of Subsection 2.4.1 to clarify this point. Thank you for spotting this.

Note that in Section 2.4, we did not add the entropic regularisation to the cost functions for the sake of conciseness and because it does not play a role in the key ideas developed in this section; it would however be added and employed in numerical applications.

*(3) Ll. 335 and 597, what is a Monge map (a scheme of the form of Fig. 3) ?*

The Monge map is the optimal deterministic map $T$ introduced in the deterministic formalism of optimal transport by Monge, and defined by Eqs. (2,3) of the manuscript. This has been made explicit in the revised manuscript, just after Eq. (3):

... whose purpose is to minimise the total transport cost between $\rho_{\mathrm{o}}$ and $\rho_{\mathrm{b}}$, and whose optimal map $T$ is often referred to as the *Monge map*.

*(4) Ll. 469- 471, I find the sentence starting If $\mathbf{y}^b$ and $\mathbf{y}^o$ were...rather vague. What does it mean that the classical analysis would be as good as can be, while the OTDA solution may be too safe ? That the sharp transitions visible on panel 12 (b) would not be there, and that the OTDA solution would be too smooth ? Or something else ?*

Yes, exactly. Following your questions, we elaborated on the sentence in the revised manuscript:

If $\mathbf{y}^b$ and $\mathbf{y}^o$ were consistently obtained from a truth perturbed with errors with short-range correlation, i.e. if they were drawn from the true prior distribution and in the absence of mislocation errors, then the classical analysis would be as good as can be, while the OTDA solution may be too safe, i.e. too smooth.

*(5) L. 478, What does it mean that the first guess is weak ? That it is numerically small so as to be negligible, or erroneous, unreliable, or what ?*

We meant, using rigorous terminology, uncertain (unreliable) as compared to the observations uncertainty. We modified the text to clarify this point. Thank you for the suggestion.

... because the first guess in that region is very uncertain.

*(6) Caption of Fig. 12, what is a heatmap (I suspect a typo) ?*

*Heatmap* is another popular name for *density plot*. To avoid introducing new names for the same object, we replaced *heatmaps* in the revised manuscript with *concentration maps* as in the first sentence of the caption.

*(7) L. 155, the kernel of $\mathbf{H}$ is non-trivial. May be confusing. I suggest you simply write The main caveat of Eq. (7) comes from the fact that the system is only partially observed.*

We followed your suggestion and changed the sentence accordingly. Thank you for the suggestion.

*(8) L. 537, ... conformally to the spaces ... $\longrightarrow$ ... onto the spaces ...*

Corrected. Thank you for the suggestion.

*(9) L. 285 ... the same tensor index is present ...*

Corrected.

*(10) From what I can judge, the English is correct and perfectly understandable. A few corrections will however have to be made here and there. The paper will be copy-edited anyway, but I mention one point. The authors repeatedly write associated to (l. 288 for instance). That should be associated with (incidentally, that is a great classic of French authors when writing English).*

All the occurrences of *associated to* have been corrected. Thank you for the suggestion.

**Authors' response to Reviewer 1's comments**
**on the revised manuscript "Bridging classical data assimilation and optimal transport: The 3D-Var case"**

Preprint egusphere-2023-2755.R1

M. Bocquet , P. J. Vanderbecken, A. Farchi, J. Dumont Le Brazidec, and Y. Roustan

17 May 2024

We appreciate the reviewer's comments and suggestions. In the following, we discuss the raised concerns and what we have changed in the manuscript. The pdf document highlighting the differences between the original and the revised manuscript is provided.

*(1) Related to my previous comment 1 and the authors' response, I agree that the manuscript ultimately clarifies that the theory applies to fields and not probability distributions. But in the original version at least, this is not stated until Sec. 1.4 on pg 5. The new MS makes this somewhat clearer although I personally believe it should be said earlier and more prominently.*

We indeed had already modified the text to clarify this point following your initial comment. We do not believe that the *revised* manuscript is confusing or misleading in that regard. The title and the abstract of the revised manuscript were quite clear on the topic: (i) 3D-Var (in the revised title) is mostly focused on states and not pdfs, (ii) in the revised abstract, we wrote: *As such it provides a very attractive metric for non-negative, sharp fields comparison — the Wasserstein distance — which could further be used in data assimilation for the geosciences.* And then later on in the revised abstract: *The resulting OTDA framework accounts for both the classical source of prior errors, background and observation, together with a Wasserstein barycentre in between states that stand for these background and observation.* Note that, in the introduction specifically, we cannot alert the reader about this before section 1.4, because we did not introduce the specific problem of the paper yet. Adding a warning on a subject not yet introduced would be confusing.

We would like to mention that, closely connected to this point, we have slightly improved the paragraph where the curse of dimensionality is mentioned.

In the context of this paper, it is *critical* to be aware that the use of OT in practical DA focused so far on applying OT independently to the pdf of each single scalar variable. Quite often, OT is applied to the pdf of a single random variable because

- OT in one dimension (the space of the values taken by this random variable) together with the quadratic cost has a very simple solution that only relies on the cumulative distribution functions of the origin and target measures (see e.g., Remark 2.30 in Peyré and Cuturi, 2019), a technique known in statistics as quantile matching,

- increasing the number of random variables is subject to the curse of dimensionality, necessitating an exponential increase in computational resources, when increasing the resolution of the discretised fields.

This is very different from our context and objective where the objects dealt with by OT are (non-negative) physical field states, not the pdf of one of their scalar variables.

Note that we have completely removed the last sentence of the former version of this paragraph about OTDA not being subject to the curse of dimensionality, which is was unnecessary and indeed potentially confusing.

*(2) Related to my previous comments 2 and 4c and what types of distances are able to "cope" with "distortions", I believe it depends on what one means with "to cope" and with "distortions",*

*and I also believe that there is no simple binary answer. The authors do not analyse this important issue in any depth which I feel is the main shortcoming of this paper.*

We do not make claim of novelty on this topic; we do not add more in the revised manuscript and rely entirely on the literature where the need for non-local metrics was advocated. The novelty is in a consistent and practical set of techniques to embed OT into classical DA. The justification of why we would need that set of methods, what *distorsion* is and why classical methods have poor discriminating power (to cope with) also very much depend on the application of the potential user of the method. For us, the authors, our physical motivation is rooted in atmospheric chemistry and dispersion and is discussed at length in Farchi et al. (2016); Vanderbecken et al. (2023). We also found other references in the field of meteorological verification where instead of *distorsion*, the authors speak of *amplitude*, *structure* and *location*, (e.g., Wernli et al., 2008), pointing again to the fact that traditional local metrics are non-discriminative for fields such as precipitation. This reference was added to the newly revised manuscript.

*(3) Related to previous comment (4a), it is still not clear to me how the authors deal with comparing fields that do not have the same mass.*

It is likely that we did not understand your question, and in particular on which part of the manuscript it is related to. If your question is related to the general idea of the method, the unbalance is addressed in the cost function by the error terms (hence those of classical DA), which do not require the fields $\mathbf{y}^{\mathrm{b}}$ and $\mathbf{x}^{\mathrm{b}}$ on the one hand, and $\mathbf{y}^{\mathrm{o}}$ and $\mathbf{x}^{\mathrm{o}}$ on the other hand to have the same mass, as opposed to the OT terms where the origin and target fields must have the same mass. If your question is related to the thought experiment where we showed that the previous proposal on combining OT and DA is flawed and where we assumed that $\mathbf{y}^{\mathrm{b}}$ and $\mathbf{y}^{\mathrm{o}}$ have the same mass, then we merely proved that the previous proposal is flawed in the specific case where $\mathfrak{m}(\mathbf{y}^{\mathrm{b}}) = \mathfrak{m}(\mathbf{y}^{\mathrm{o}})$. It is hence flawed in general.

*(4) Related to previous comment (4b), I now see that the regularised problem is convex but as a matter of fact, the original problem is a linear program and thus convex, albeit potentially not strictly convex. This should be clarified. In addition, it cannot hurt to comment on why the KL approach is appropriate, given that it strongly penalises fields that are rather localised with respect to the measure $\nu$. Dealing with strongly localised fields was, as far as I understand, a motivation for the authors to propose the optimal transport methodology in the first place.*

You are absolutely right. We were wrong in assuming that the original problem was not necessarily convex. Further, as you explained, it is often not strictly convex (which we know from the theory and by our own experience). Thank you for insisting on the topic. We have amended the manuscript accordingly. So the KL regularisation makes the problem strictly convex whenever $\varepsilon > 0$. It also has the key property that it lifts the positivity constraint of the fields. It is so obvious that it is easy to forget insisting on this key virtue.

The optimisation problem Eq. (14) is a *linear program* which is convex (Peyré and Cuturi, 2019, and references therein). Yet, it is not generally strictly convex, and hence does not necessarily exhibit a single minimum. Adding to the difficulty, its cost function Eq. (14a) is constrained. *Entropic regularisation* addresses these issues and is used here to lift the constraints and to render the problem strictly convex. It will in particular force any state vector which is solution of the problem to be positive.

The KL term is scaled by $\varepsilon > 0$ which can nonetheless be small. In practice, even with rather strongly localised fields, we did not see the issue with the 1D and 2D examples (many more than those reported in the manuscript). However, we agree that with fields with multiple relevant scales with, e.g., a large scale together with highly localised features, getting the optimal $\varepsilon$ could be a real subject of investigation.

**References**

Farchi, A., Bocquet, M., Roustan, Y., Mathieu, A., and Quérel, A.: Using the Wasserstein distance to compare fields of pollutants: application to the radionuclide atmospheric dispersion of the Fukushima-Daiichi accident, Tellus B, 68, 31 682, https://doi.org/10.3402/tellusb.v68.31682, 2016.

Peyré, G. and Cuturi, M.: Computational Optimal Transport: With Applications to Data Science, Foundations and Trends® in Machine Learning, 11, 355–607, https://doi.org/10.1561/2200000073, 2019.

Vanderbecken, P. J., Dumont Le Brazidec, J., Farchi, A., Bocquet, M., Roustan, Y., Potier, E., and Broquet, G.: Accounting for meteorological biases in simulated plumes using smarter metrics, Atmos. Meas. Tech., 16, 1745–1766, https://doi.org/10.5194/amt-16-1745-2023, 2023.

Wernli, H., Paulat, M., Hagen, M., and Frei, C.: SAL–A Novel Quality Measure for the Verification of Quantitative Precipitation Forecasts, Mon. Wea. Rev., 136, 4470–4487, https://doi.org/10.1175/2008MWR2415.1, 2008.

---

## Author Response (AR3)

**Authors' response to the Editor's comments**
**on the revised manuscript "Bridging classical data**
**assimilation and optimal transport: The 3D-Var case"**

Preprint egusphere-2023-2755.R2

M. Bocquet , P. J. Vanderbecken, A. Farchi, J. Dumont Le Brazidec, and Y. Roustan

21 May 2024

We appreciate the Editor's commitment in reviewing the manuscript, his comments and suggestions. The three technical required corrections have been implemented in the revised manuscript. Thank you for your suggestions.